# One-form symmetries and the 3d $\mathcal{N} = 2$ $A$-model: Topologically twisted indices and CS theories

**Cyril Closset, Elias Furrer and Osama Khlaif**

School of Mathematics, University of Birmingham,
Watson Building, Edgbaston, Birmingham B15 2TT, UK

## Abstract

We study three-dimensional $\mathcal{N} = 2$ supersymmetric Chern–Simons-matter gauge theories with a one-form symmetry in the $A$-model formalism on $\Sigma_g \times S^1$. We explicitly compute expectation values of topological line operators that implement the one-form symmetry. This allows us to compute the topologically twisted index on the closed Riemann surface $\Sigma_g$ for any real compact gauge group $G$ as long as the ground states are all bosonic. All computations are carried out in the effective $A$-model on $\Sigma_g$, whose $S^1$ ground states are the so-called Bethe vacua. We discuss how the 3d one-form symmetry acts on the Bethe vacua, and also how its 't Hooft anomaly constrains the vacuum structure. In the special case of the $SU(N)_K$ $\mathcal{N} = 2$ Chern–Simons theory, we obtain results for the $(SU(N)/\mathbb{Z}_r)_K^\theta$ $\mathcal{N} = 2$ Chern–Simons theories, for all non-anomalous $\mathbb{Z}_r \subseteq \mathbb{Z}_N$ subgroups of the centre of the gauge group, and with a $\mathbb{Z}_r$ $\theta$-angle turned on. In the special cases with $N$ even, $\frac{N}{r}$ odd and $\frac{K}{r}$ even, we find a mixed 't Hooft anomaly between gravity and the $\mathbb{Z}_r^{(1)}$ one-form symmetry of the $SU(N)_K$ theory, and the infrared 3d TQFT after gauging is spin. In all cases, we count the Bethe states and the higher-genus states in terms of refinements of Jordan's totient function. This counting gives us the twisted indices if and only if the infrared 3d TQFT is bosonic. Our results lead to precise conjectures about integrality of indices, which appear to have a strong number-theoretic flavour. *Note:* this paper directly builds upon unpublished notes by Brian Willett from 2020.

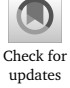

# 1 Introduction

Three-dimensional $\mathcal{N} = 2$ supersymmetric gauge theories are free theories in the ultraviolet (UV) but flow to strong coupling in the infrared (IR). Over the past 15 years, a huge amount of non-perturbative results were obtained for such 3d theories that also have a $U(1)_R$ $R$-symmetry, using powerful supersymmetric localisation techniques, starting with the exact computations of the $S^2 \times S^1$ (so-called) superconformal index [1, 2] and of the three-sphere partition function [3, 4] – see *e.g.* [5–7] for reviews.

Almost all of these 3d $\mathcal{N} = 2$ supersymmetric partition functions can be understood in the framework of the 3d $A$-model, which allows us to evaluate supersymmetric path integrals on any Seifert 3-manifold $\mathcal{M}_3$ [7–9]. The 3d $A$-model is a two-dimensional cohomological topological field theory (Coh-TQFT) obtained as the topological $A$-twist of the 2d $\mathcal{N} = (2, 2)$ supersymmetric effective field theory description of the 3d $\mathcal{N} = 2$ gauge theory compactified on a circle. In the simplest instance, the 3d $A$-model computes the twisted index on a genus-$g$ Riemann surface, $\Sigma_g$ [10–13]:

$$Z_{\Sigma_g \times S^1} = \text{Tr}_{\Sigma_g} \left( (-1)^{\text{F}} y^{Q_F} \right) = \sum_{\hat{u} \in \mathcal{S}_{\text{BE}}} \mathcal{H}(\hat{u}, \nu)^{g-1}. \tag{1}$$

Here the trace in the first equality is over the supersymmetric ground states on $\Sigma_g$ with a topological twist, where $(-1)^{\text{F}}$ is the fermion number and $y = e^{2\pi i \nu}$ denote fugacities for some flavour charges $Q_F$. The second equality of (1) computes this twisted flavoured index as a sum over the so-called Bethe vacua $\{\hat{u}\}$. These are the ground states of the 3d theory compactified on $T^2$. More generally, for any Seifert-fibered three-manifold $\mathcal{M}_3$, there exists a similar formula for the supersymmetric partition function [9]. Let

$$\mathcal{H}_{S^1} \cong \text{Span}_{\mathbb{C}} \{ |\hat{u}\rangle \}, \tag{2}$$

denote the Hilbert space of $A$-model ground states, spanned by the Bethe vacua, where the states $|\hat{u}\rangle$ are orthonormal. We then have:

$$Z_{\mathcal{M}_3} = \text{Tr}_{\mathcal{H}_{S^1}} \left( \mathcal{H}^{g-1} \mathcal{G}_{\mathcal{M}_3} \right) = \sum_{\hat{u} \in \mathcal{S}_{\text{BE}}} \langle \hat{u} | \mathcal{H}^{g-1} \mathcal{G}_{\mathcal{M}_3} | \hat{u} \rangle = \sum_{\hat{u} \in \mathcal{S}_{\text{BE}}} \mathcal{H}(\hat{u}, \nu)^{g-1} \mathcal{G}_{\mathcal{M}_3}(\hat{u}, \nu). \tag{3}$$

Here, $\mathcal{H}$ and $\mathcal{G}_{\mathcal{M}_3}$ are called the handle-gluing operator and the Seifert fibering operator, respectively. They are both diagonalised by the Bethe vacua (with $\mathcal{O}(\hat{u})$ denoting the eigenvalue of $\mathcal{O}$). The fibering operator $\mathcal{G}_{\mathcal{M}_3}$ induces a non-trivial fibration of $S^1$ over $\Sigma_g$ [8, 9].

While this 3d $A$-model formalism is very powerful, it was originally developed under the assumption that the gauge group $\widetilde{G}$ be a product of simply-connected and of unitary compact gauge groups – that is, the compact Lie group $\widetilde{G}$ is such that $\pi_1(\widetilde{G})$ is a free abelian group. In this paper, we are interested in more general cases with a gauge group of the form:

$$G = \widetilde{G}/\widetilde{\Gamma}, \qquad \widetilde{\Gamma} \subseteq Z(\widetilde{G}), \tag{4}$$

where the discrete abelian group $\widetilde{\Gamma}$ is a subgroup of the centre of $\widetilde{G}$. This group is also a subgroup of the one-form symmetry group $\Gamma_{3d}^{(1)}$ of the $\widetilde{G}$ gauge theory:

$$\Gamma_{3d}^{(1)} \cong \Gamma, \qquad \widetilde{\Gamma} \subseteq \Gamma, \tag{5}$$

which is the group under which Wilson lines can be charged [14, 15]. Supersymmetric observables in the gauge theory with gauge group $G$ can be obtained from the original $\widetilde{G}$ gauge theory by gauging a non-anomalous subgroup $\widetilde{\Gamma}$ of the one-form symmetry [16]. Schematically, this corresponds to summing over all background gauge fields $B$ for the one-form symmetry $\widetilde{\Gamma}_{3d}^{(1)} \cong \widetilde{\Gamma}$, namely:

$$Z_{\mathcal{M}_3} \left[ \mathcal{T}/\widetilde{\Gamma}_{3d}^{(1)} \right] = \sum_B Z_{\mathcal{M}_3}[\mathcal{T}](B). \tag{6}$$

Here, $Z_{\mathcal{M}_3}[\mathcal{T}](B)$ denotes the path integral of the original $\widetilde{G}$ gauge theory, $\mathcal{T}$, in the presence of a background gauge field $B \in H^2(\mathcal{M}_3, \widetilde{\Gamma})$.

Turning on a background gauge field $B$ for $\Gamma_{3d}^{(1)}$ is equivalent to inserting a web of topological line operators $\mathcal{U}^\gamma$ [16]. In this paper, we discuss such insertions in the 3d $A$-model

formalism. More generally, we discuss the action of the one-form symmetry on the Bethe vacua. In particular, we explain how 't Hooft anomalies for the one-form symmetry $\Gamma_{3d}^{(1)}$ usefully constrain the structure of the Bethe vacua. We then discuss the gauging of non-anomalous subgroups of $\Gamma_{3d}^{(1)}$ as explicitly as possible.

The core ideas explored in this work were first discussed by Brian Willett in unpublished notes [17]; they were also briefly explained in [18]. The gauging of non-anomalous one-form and zero-form symmetries in 2d TQFTs was also discussed in detail by Gukov, Pei, Reid and Shehper [19], and we will recover many of their results. In this paper, we give a detailed account of the physics of 3d $\mathcal{N} = 2$ supersymmetric gauge theories with one-form symmetries when compactified on $S^1$, including the dynamical consequences of 't Hooft anomalies. We then study topologically-twisted supersymmetric indices for 3d $\mathcal{N} = 2$ gauge theories with general gauge groups $G$. For concreteness, after discussing the formalism in general terms, our main computations will focus on the case of theories with $\widetilde{G} = SU(N)$ and a $\mathbb{Z}_N$ one-form symmetry. As we will see, the story can become particularly intricate depending on the arithmetic properties of $N$ – here, $N$ prime will be the easy case, but for general $N$ a number of interesting phenomena will arise, including a subtle 3d modular anomaly.

We plan to discuss further aspects of generalised symmetries of 3d $\mathcal{N} = 2$ supersymmetric field theories, including a more thorough discussion of supersymmetric partition functions for general Seifert three-manifolds, as well as what happens in the presence of fermionic Bethe states, in future works [20, 21]. The present work is also related in various ways to many recent papers on 3d field theory, including *e.g.* [22–33].

## 1.1 Warm-up example: The $SU(2)_K$ supersymmetric Chern–Simons theory

Before discussing our general result, let us first discuss the example of the 3d $\mathcal{N} = 2$ supersymmetric Chern–Simons (CS) theory $SU(2)_K$. Gauging the $\mathbb{Z}_2$ one-form symmetry, one obtains the $SO(3)_K$ CS theory. This is an interesting example which already involves many of the intricacies of the general $SU(N)_K$ case to be discussed below.

Upon integrating out the gauginos in the vector multiplet, the $\mathcal{N} = 2$ $SU(2)_K$ CS theory is equivalent to the bosonic CS theory $SU(2)_k$ with $k = K - 2$. The theory has $K - 1$ Bethe vacua, corresponding to the allowable solutions to the Bethe equations:

$$\Pi(u) \equiv x^{2K} = 1, \qquad x \equiv e^{2\pi i u}, \qquad u \to u + 1. \tag{7}$$

Here, $u$ is the 2d Coulomb branch parameter, with the $\mathbb{Z}_2$ Weyl group acting as $u \to -u$. The operator $\Pi$ is called the gauge flux operator, and it is given in terms of the effective twisted superpotential $\mathcal{W}$ of the $A$-model:

$$\Pi(u) = \exp\left(2\pi i \frac{\partial \mathcal{W}}{\partial u}\right), \qquad \mathcal{W} = K u^2. \tag{8}$$

The Bethe vacua corresponds to pairs of solutions $\pm \hat{u}_l$ related by the Weyl symmetry, with:

$$\hat{u}_l = \frac{l}{2K}, \qquad l = 1, \ldots, K - 1. \tag{9}$$

The handle-gluing operator of this theory reads:

$$\mathcal{H}(u) = \frac{K}{2 \sin^2(2\pi u)}, \tag{10}$$

and therefore the $\Sigma_g$ twisted index (1) is given by:

$$Z_{\Sigma_g \times S^1}[SU(2)_K] \equiv \langle 1 \rangle_{\Sigma_g} = \sum_{l=1}^{K-1} \left[\sqrt{\frac{2}{K}} \sin\left(\frac{\pi l}{K}\right)\right]^{2-2g}. \tag{11}$$

Setting $K = k + 2$, this is the well-known Verlinde formula for the pure CS theory $SU(2)_k$ – equivalently, it gives the number of conformal blocks of the $SU(2)_k$ WZW model [34] (see also [35–39] and references therein).

**Higher-form symmetries and their anomalies**   The 1-form symmetry $\Gamma_{3d}^{(1)} = \mathbb{Z}_2$ of the 3d CS theory is realised as $\mathbb{Z}_2^{(1)} \oplus \mathbb{Z}_2^{(0)}$ in the 2d description. Let us now explain how these two symmetries are manifested in the $A$-model. First of all, the non-trivial charge operator for $\mathbb{Z}_2^{(1)}$ is a local topological operator. As we will explain later on, it is given by a square root of the gauge flux operator, which we thus denote as:

$$\Pi^{\frac{1}{2}}(u) = -x^K \,. \tag{12}$$

Here, the sign of the square root is chosen to match known results. Inserting this operator on $\Sigma_g$, we obtain:

$$\langle \Pi^{\frac{1}{2}} \rangle_{\Sigma_g} = \sum_{l=1}^{K-1} (-1)^{l+1} \left[ \sqrt{\frac{2}{K}} \sin\left(\frac{\pi l}{K}\right) \right]^{2-2g} \,. \tag{13}$$

Note that this vanishes if $K$ is odd. In particular, the insertion on the torus gives us:

$$\langle \Pi^{\frac{1}{2}} \rangle_{T^2} = \begin{cases} 1 \,, & \text{for } K \text{ even,} \\ 0 \,, & \text{for } K \text{ odd.} \end{cases} \tag{14}$$

Let $\mathcal{T}$ denote the $A$-model for this theory. The gauging of $\mathbb{Z}_2^{(1)}$ consists simply of summing over insertions of the flux operator:

$$Z_{\Sigma_g}\left[SU(2)_K / \mathbb{Z}_2^{(1)}\right] = \frac{1}{2}\left(\langle 1 \rangle_{\Sigma_g} + \langle \Pi^{\frac{1}{2}} \rangle_{\Sigma_g}\right) = \sum_{j=0}^{\lfloor \frac{K-2}{2} \rfloor} \left[ \sqrt{\frac{2}{K}} \sin\left(\frac{\pi(2j+1)}{K}\right) \right]^{2-2g} \,. \tag{15}$$

When gauging, we have the freedom of introducing a coupling of the $\mathbb{Z}_2^{(1)}$ gauge field $B$ to a background gauge field $\theta$ for the so-called quantum $(-1)$-form symmetry, as we will review; here we chose $\theta = 0$ for simplicity of presentation.

The zero-form symmetry $\mathbb{Z}_2^{(0)}$ is generated by topological line operators $\mathcal{U}^\gamma$, for $\gamma \in \mathbb{Z}_2^{(0)}$. To study the action of the 0-form symmetry on $\mathcal{H}_{S^1}$, we first consider the $A$-model on a cylinder. We thus have a single generator $\mathcal{U}^\gamma(\mathcal{C})$ wrapping the cylinder, with the fusion $\mathcal{U}^2 = \mathbf{1}$, and we find that:

$$\mathcal{U}(\mathcal{C})|\hat{u}_l\rangle = |\hat{u}_{K-l}\rangle \,. \tag{16}$$

Note that $\mathcal{U}(\mathcal{C})$ has a single fixed point, $|\hat{u}_{\frac{K}{2}}\rangle$, if and only if $K$ is even. The $SU(2)_K$ 3d $\mathcal{N} = 2$ CS theory has a 't Hooft anomaly $\frac{K}{2}$ (mod 1) for $\Gamma_{3d}^{(1)} = \mathbb{Z}_2$ – that is, the $\mathbb{Z}_2$ one-form symmetry is anomalous if $K$ is odd [16]. In the 3d $A$-model, this manifests itself as a mixed $\mathbb{Z}_2^{(1)}$-$\mathbb{Z}_2^{(0)}$ anomaly. The topological operators acting on $\mathcal{H}_{S^1}$ satisfy the twisted commutation relation:

$$\mathcal{U}(\mathcal{C}) \Pi^{\frac{1}{2}} = (-1)^K \Pi^{\frac{1}{2}} \mathcal{U}(\mathcal{C}) \,, \tag{17}$$

which follows from (16) and the fact that $\Pi^{\frac{1}{2}}|\hat{u}\rangle = (-1)^{l+1}|\hat{u}\rangle$.

Let us now set $g = 1$, with the $\mathcal{C}$ being one generator of $H_1(T^2, \mathbb{Z}) \cong \mathbb{Z}^2$ and $\widetilde{\mathcal{C}}$ denoting the other generator (the Euclidean time direction). Inserting a topological line along $\mathcal{C}$, we find:

$$\langle \mathcal{U}(\mathcal{C}) \rangle_{T^2} = \sum_{l=1}^{K-1} \langle \hat{u}_l | \mathcal{U}(\mathcal{C}) | \hat{u}_l \rangle = \sum_{l=1}^{K-1} \langle \hat{u}_l | \hat{u}_{K-l} \rangle = \begin{cases} 1 \,, & \text{if } K \text{ is even,} \\ 0 \,, & \text{if } K \text{ is odd.} \end{cases} \tag{18}$$

Finally, the insertion of a topological line operator along $\widetilde{\mathcal{C}} \subset T^2$ corresponds to a trace over the non-trivial twisted-sector Hilbert space, which arises from the Bethe vacuum $\hat{u}_{\frac{K}{2}}$ that preserves $\mathbb{Z}_2^{(0)}$ – such a twisted sector only exists for $K$ even. We can compute:

$$\langle \mathcal{U}(\widetilde{\mathcal{C}}) \rangle_{T^2} = \langle \mathcal{U}(\mathcal{C}) \rangle_{T^2} = \langle \Pi^{\frac{1}{2}} \rangle_{T^2}. \tag{19}$$

The first equality is expected from modular invariance in 2d. Moreover, the second equality is also a consequence of modular invariance for the 3d Chern–Simons theory on $T^3$, and it is one way to fix the sign in (12). Similarly, if we insert both the $\mathbb{Z}_2^{(1)}$ operator $\Pi^{\frac{1}{2}}$ and $\mathbb{Z}_2^{(0)}$ operator $\mathcal{U}(\mathcal{C})$ on $T^2$, we find:

$$\langle \Pi^{\frac{1}{2}} \mathcal{U}(\mathcal{C}) \rangle_{T^2} = \sum_l \Pi^{\frac{1}{2}}(\hat{u}_l) \langle \hat{u}_l | \hat{u}_l + \tfrac{1}{2} \rangle = \begin{cases} (-1)^{\frac{K-2}{2}}, & \text{if } K \text{ is even,} \\ 0, & \text{if } K \text{ is odd.} \end{cases} \tag{20}$$

Note that this vanishes if and only if the theory is anomalous, which is the expected result given the commutator (17). Finally, we note that:

$$\langle \Pi^{\frac{1}{2}} \mathcal{U}(\mathcal{C}) \rangle_{T^2} = (-1)^{\frac{K-2}{2}} \langle \mathcal{U}(\mathcal{C}) \rangle_{T^2}, \tag{21}$$

for $K$ even. From the point of view of the 3d CS theory on $T^3$, we would naively have expected that relation to hold with a trivial sign, but this only happens if $\frac{K}{2}$ is odd. For $\frac{K}{2}$ even, we have a sort of modular anomaly on $T^3$, which is a consequence of a mixed anomaly between the 3d one-form symmetry and gravity. We will discuss this subtle and important point in section 3.2, in the context of the $SU(N)_K$ $\mathcal{N}=2$ CS theory.

**The $SO(3)_K$ $T^2$ partition function**   Let us now assume that $K$ is even, so that we can gauge both $\mathbb{Z}_2^{(1)}$ and $\mathbb{Z}_2^{(0)}$. Let us first consider $\Sigma_g = T^2$. Naively, this should give us the Witten index of the $SO(3)_K$ $\mathcal{N}=2$ supersymmetric CS theory. The gauging corresponds to inserting all possible topological operators for $\mathbb{Z}_2^{(1)} \times \mathbb{Z}_2^{(0)}$ on $T^2$. We thus find:

$$
\begin{aligned}
Z_{T^2}[SO(3)_K] &= \frac{1}{4} \sum_{n,n',n'' \in \mathbb{Z}_2} \left\langle \mathcal{U}(\mathcal{C})^n \mathcal{U}(\widetilde{\mathcal{C}})^{n'} \Pi^{\frac{n''}{2}} \right\rangle_{T^2} \\
&= \frac{1}{4}\left( K - 1 + 3 + 1 + 3(-1)^{\frac{K-2}{2}} \right) = \begin{cases} \frac{K}{4} + \frac{3}{2}, & \text{for } \frac{K}{2} \text{ odd,} \\ \frac{K}{4}, & \text{for } \frac{K}{2} \text{ even.} \end{cases}
\end{aligned} \tag{22}
$$

It is illuminating to compare this result to the non-supersymmetric $SO(3)_k$ CS theory (with $k = K-2$). The above result should give us the number of states of the $SO(3)_k$ theory on $T^2$, which then reads:

$$Z_{T^3}[\mathcal{N}=0 \ SO(3)_k] = \begin{cases} \frac{k}{4} + 2, & \text{for } \frac{k}{2} \text{ even,} \\ \frac{k}{4} + \frac{1}{2}, & \text{for } \frac{k}{2} \text{ odd.} \end{cases} \tag{23}$$

We have a bosonic 3d TQFT for $k \in 4\mathbb{Z}$, while $SO(3)_k$ with $k + 2 \in 4\mathbb{Z}$ is a spin-TQFT [40]. In either case, one can check that (23) corresponds to the number of $SO(3)_k$ Wilson lines. The 3d one-form gauging $SU(2)_k \to SO(3)_k = SU(2)_k/\mathbb{Z}_2$ can be performed directly as follows. There are $k+1$ $SU(2)_k$ Wilson lines, or anyons, for the $SU(2)$ representations of spin $j = 0, \ldots, \frac{k}{2}$, namely:

$$\mathcal{N}=0 \ SU(2)_k \quad : \quad \{W_j\} = \left\{ \mathbf{1}, W_{\frac{1}{2}}, W_1, \ldots, W_{\frac{k}{2}} \right\}, \tag{24}$$

with the fusion rules [41]:

$$W_{j_1} W_{j_2} = \sum_{j=|j_1-j_2|}^{\min(j_1+j_2, k-j_1-j_2)} W_j. \tag{25}$$

The lines (24) include exactly two abelian anyons, $\mathbf{1}$ and $a \equiv W_{\frac{k}{2}}$, with fusion $a^2 = \mathbf{1}$, which are therefore the $\mathbb{Z}_2^{(1)}$ topological lines in 3d. We have:

$$\left(\mathbb{Z}_2^{(1)}\right)_{3d} : W_j \to (-1)^{2j} W_j, \qquad a W_j = W_{\frac{k}{2}-j}, \tag{26}$$

for the action of $a$ on $W_j$ by linking and for the fusion, respectively. The gauging of $(\mathbb{Z}_2^{(1)})_{3d}$ consists of three steps [42]. First, we discard all lines which are not invariant under $(\mathbb{Z}_2^{(1)})_{3d}$, leaving us with $\{W_j\}$, $j \in \mathbb{Z}$. Secondly, we identify the lines $W$ and $aW$. Thirdly, any line that is a fixed point of the fusion with $a$ gives rise to two distinct lines in the gauged theory. The fixed-point line is $W_{\frac{k}{4}}$, and thus only survives the first step of gauging for $\frac{k}{2}$ even. We thus find the $SO(3)_k$ lines:

$$\mathcal{N} = 0 \ SO(3)_k \quad : \quad \begin{cases} \left\{\mathbf{1}, W_1, W_2, \ldots, W_{\frac{k}{4}-1}, W_{\frac{k}{4};(1)}, W_{\frac{k}{4};(2)}\right\}, & \text{for } \frac{k}{2} \text{ even,} \\ \left\{\mathbf{1}, W_1, W_2, \ldots, W_{\frac{k}{4}-\frac{1}{2}}\right\}, & \text{for } \frac{k}{2} \text{ odd.} \end{cases} \tag{27}$$

This precisely reproduces (23). It is important to note, however, that the above result is the correct Witten index of the $SO(3)_K$ $\mathcal{N} = 2$ supersymmetric theory if and only if $\frac{K}{2}$ is odd (hence if $\frac{k}{2}$ is even), so that the 'modular anomaly' in (21) disappears. When $\frac{K}{2}$ is even, the infrared 3d TQFT $SO(3)_k$ is actually a spin-TQFT because the abelian anyon has spin $h[a] = \frac{1}{2}$, and additional care must be taken in interpreting our result. In that case, we should have explicitly chosen a spin structure on $\Sigma_g$ and the index will depend on that choice. One can show that the true Witten index of $SO(3)_K$ for $\frac{K}{2}$ even and in the RR sector on $T^2$ is equal to $\frac{K}{4} - 2$ [43]; the extension of the $A$-model formalism to include this case will be discussed elsewhere [21].

**The higher-genus twisted index for $SO(3)_K$** We can generalise the computation (22) to obtain the $SO(3)_K$ twisted index for any $\Sigma_g$, for $K$ even. First, we find that the insertion of the non-trivial $\mathbb{Z}_2^{(0)}$ line along any generator $\mathcal{C}_i$ of $H_1(\Sigma_g, \mathbb{Z}) \cong \mathbb{Z}^{2g}$ gives us:

$$\langle \mathcal{U}(\mathcal{C}_i) \rangle_{\Sigma_g \times S^1} = \sum_{l=1}^{K-1} \langle \hat{u}_l | \mathcal{U}(\mathcal{C}_i) \mathcal{H}^{g-1} | \hat{u}_l \rangle = \mathcal{H}\left(\hat{u}_{\frac{K}{2}}\right)^{g-1} = \left(\frac{K}{2}\right)^{g-1}. \tag{28}$$

Moreover, the insertion of any non-trivial set of $\mathbb{Z}_2^{(0)}$ lines on $\Sigma_g$ is equal to (28), due to invariance under large diffeomorphism. We similarly find that:

$$\langle \mathcal{U}(\mathcal{C}_i) \Pi^{\frac{1}{2}} \rangle_{\Sigma_g \times S^1} = (-1)^{\frac{K-2}{2}} \left(\frac{K}{2}\right)^{g-1}. \tag{29}$$

The $\Sigma_g \times S^1$ partition function for $SO(3)_K$ is obtained as:

$$Z_{\Sigma_g}[SO(3)_K] = \frac{1}{2^{2g}} \sum_{n' \in \mathbb{Z}_2} \sum_{(n_i) \in \mathbb{Z}_2^{2g}} \left\langle \Pi^{\frac{n'}{2}} \prod_{i=1}^{2g} \mathcal{U}(\mathcal{C}_i)^{n_i} \right\rangle_{\Sigma_g \times S^1}, \tag{30}$$

which gives us:

$$Z_{\Sigma_g}[SO(3)_K] = \frac{K^{g-1}}{2^{3g-1}} \left( 2 \sum_{j=0}^{\frac{K}{2}-1} \left[ \sin\left(\frac{\pi(2j+1)}{K}\right) \right]^{2-2g} + \left(2^{2g}-1\right)\left(1+(-1)^{\frac{K-2}{2}}\right) \right). \tag{31}$$

This matches known results. In the case where $\frac{K}{2}$ is odd, so that $k = K - 2 \in 4\mathbb{Z}$, (30) is in perfect agreement with the number of conformal blocks for the $SO(3)_k$ WZW model –

see *e.g.* [44] for a mathematical derivation. In the spin-TQFT case, $k + 2 \in 4\mathbb{Z}$, our result can also be recovered from the corresponding fermionic WZW model, but the full story is much more subtle [43], as already mentioned. As a special case, we note that WZW$[SO(3)_2]$ is a free CFT [40] (equivalent to three Majorana fermions), which explains why we find $Z_{\Sigma_g \times S^1} = 1$ for any $g$ when setting $K = 4$ in (31); however, the true (RR-sector) Witten index would be equal to $-1$ in this case [21].

## 1.2 Results for general $G$ and for the $SU(N)_K$ gauge theory

In the rest of this introduction, we summarise the main results of this paper, essentially in the order in which they will be presented.

**One-form symmetries in the 3d $A$-model**   Given a 3d $\mathcal{N} = 2$ gauge theory with gauge group $\widetilde{G}$ compactified on $S^1$, as discussed above, its one-form symmetry $\Gamma_{3d}^{(1)}$ descends to both a one-form symmetry and a zero-form symmetry in 2d, denoted by $\Gamma^{(1)}$ and $\Gamma^{(0)}$, respectively. Thus, we must first understand how these two distinct symmetries act on the $A$-model Hilbert space. For any 2d TQFT, this question was recently addressed in [19]. The one-form symmetry $\Gamma^{(1)}$ acts on the ground states by a phase, and the zero-form symmetry $\Gamma^{(0)}$ acts by permutation. Indeed, we have [17,18]:

$$\Gamma^{(1)} : |\hat{u}\rangle \longrightarrow \Pi(\hat{u})^\gamma |\hat{u}\rangle \,, \qquad \Gamma^{(0)} : |\hat{u}\rangle \longrightarrow |\hat{u} + \gamma\rangle \,, \tag{32}$$

where $\gamma \in \Gamma$ is a group element. Here $\Pi(\hat{u})^\gamma \in \mathrm{Hom}(\Gamma^{(1)}, U(1))$ is a phase, and $|\hat{u} + \gamma\rangle$ denotes the new vacuum under the action of $\gamma \in \Gamma^{(0)}$. We will explain these transformations and their consequences in much detail throughout this work.

Focusing on $\mathcal{M}_3 = \Sigma \times S^1$, the 0-form symmetry $\Gamma^{(0)}$ along $\Sigma$ is implemented by topological lines, while the 1-form symmetry $\Gamma^{(1)}$ is implemented by the topological point operators that arise from the 3d topological lines wrapping $S^1$. The Bethe vacua diagonalise the $\Gamma^{(1)}$ topological operators $\Pi^\gamma$, as shown in (32), while the 0-form symmetry $\Gamma^{(0)}$ is (partially) spontaneously broken in most 2d vacua. In particular, the Bethe vacua organise themselves into orbits of $\Gamma^{(0)}$.

**'t Hooft anomalies**   The three-dimensional one-form symmetry can be anomalous, as the topological lines can themselves be charged under $\Gamma_{3d}^{(1)}$. This results in a mixed $\Gamma^{(1)}$-$\Gamma^{(0)}$ anomaly in the $A$-model on $\Sigma$. This 't Hooft anomaly implies general constraints on the structure of the orbits of Bethe vacua under the action of $\Gamma^{(0)}$. For instance, some Bethe vacua can be fixed under the whole of $\Gamma^{(0)}$ if and only if the anomaly vanishes. This follows from the fact that $\Gamma^{(1)}$ cannot be spontaneously broken in the 2d vacuum [16], and thus the mixed anomaly must be matched by distinct 2d symmetry-protected topological (SPT) phases for $\Gamma^{(1)}$ in the $A$-model description.

We will also see that the 3d 't Hooft anomaly is entirely determined by the bare Chern–Simons levels $K$ of the UV gauge theory. We further comment on the structure of the $(d + 1)$-dimensional anomaly theory, in both the $d = 3$ and $d = 2$ descriptions, in appendix C.

In the explicit $\widetilde{G} = SU(N)_K$ example at the core of this paper, a key role will also be played by a mixed 't Hooft anomaly between gravity and the 1-form symmetry in 3d. Such an anomaly can appear in more general gauge theories, and can change the interpretation of our results significantly. The general results of this paper, as presented in section 2, are thus obtained assuming that this gravitational anomaly vanishes – that happens if the 3d topological line that generates $\Gamma_{3d}^{(1)}$ has trivial braiding with the transparent line $\psi$ that captures the dependence on the spin structure of the 3-manifold. The more general case will be discussed in future work [21].

**Background gauge fields** The partition function of the $\widetilde{G}$ gauge theory on $\Sigma_g \times S^1$ can be found by inserting powers of the handle-gluing operator $\mathcal{H}$ on $\Sigma_g = T^2$ [10]. We wish to compute the topologically twisted index in the presence of topological operators for the discrete symmetry $\Gamma$, which is equivalent to turning on some background gauge fields for $\Gamma^{(1)} \times \Gamma^{(0)}$. Schematically, we have:

$$Z_{\Sigma_g \times S^1}(B, C) = \left\langle \Pi^{\gamma^{(1)}} \mathcal{U}^{\gamma^{(0)}} \right\rangle_{\Sigma_g} . \tag{33}$$

Here, $B \in H^2(\Sigma_g, \Gamma^{(1)})$ and $C \in H^1(\Sigma_g, \Gamma^{(0)})$ are the background gauge fields, while $\Pi^{\gamma^{(1)}}$ for $\gamma^{(1)} \in \Gamma^{(1)}$ and $\mathcal{U}^{\gamma^{(0)}}$ for $\gamma^{(0)} \in \Gamma^{(0)}$ denote the point and line operators, respectively, corresponding to the background gauge fields. We explicitly evaluate (33) using the 2d TQFT point of view. This simply corresponds to using the pair-of-pants decomposition of the Riemann surface with appropriate lines inserted. This prescription gives unambiguous results for insertions of the $\Gamma^{(0)}$ lines. The insertion of the $\Gamma^{(1)}$ operator is slightly ambiguous, as the flux operator $\Pi^{\gamma^{(1)}}$ is only defined up a certain root of unity. In all examples we will consider, we will be able to fix this ambiguity by demanding 3d modularity for the expectations values of elementary topological lines at $g = 1$, namely on $\mathcal{M}_3 = T^3$.

**Discrete gauging** Gauging the zero-form and one-form symmetries corresponds to summing over all possible background gauge fields [16]. Note that, in the *A*-model description, we can gauge $\Gamma^{(1)}$ and $\Gamma^{(0)}$ separately. We also keep track of the background gauge fields for the dual $(-1)$-form and 0-form symmetries, respectively. We are careful to fix the overall normalisation of the partition functions for the gauged theories in order to preserve the 2d Hilbert-space interpretation in the gauged theories.

The $\Gamma^{(1)}$ symmetry induces a so-called decomposition [45, 46] of the *A*-model into disconnected sectors, also called 'universes', and the gauging of the 1-form symmetry projects us onto one particular sector. These sectors are indexed by $\theta$-angles which are the background gauge fields for the dual $(-1)$-form symmetry. The gauging of the $\Gamma^{(0)}$ symmetry on $\Sigma_g$ is a standard orbifolding procedure familiar from string theory: in canonical quantisation on $T^2$, we identify the states related by $\Gamma^{(0)}$ while also adding in the twisted sectors states induced by non-trivial stabiliser subgroups. Overall, our general analysis closely follows previous discussions obtained in the 2d TQFT framework [17, 19]; the main improvement in our analysis is that we carefully study the Hilbert spaces spanned by the Bethe states before and after the gauging, and that we elucidate how 't Hooft anomalies constrain the structure of 2d ground states. We also keep track of all the dual symmetries that arise upon gauging.

**Symmetries of the $\mathcal{N} = 2$ $SU(N)_K$ CS theory** We apply the general descriptions of the higher-form symmetries of the 3d *A*-model to the pure $\mathcal{N} = 2$ supersymmetric CS theory with gauge group $SU(N)$ at CS level $K$. We study this theory in depth in section 3. In this example, we can compare the results obtained using our formalism to many previous results in the literature, and we find perfect agreement.

The Bethe equations for $SU(N)_K$ have $\binom{K-1}{N-1}$ solutions, which gives us the Witten index $\mathbf{I}_W[SU(N)_K]$ [47]. This 3d CS theory has a $\mathbb{Z}_N^{(1)}$ one-form symmetry, the centre symmetry, with a 't Hooft anomaly given by $K$ modulo $N$. The 3d 1-form symmetry descends to $\Gamma^{(1)} = \mathbb{Z}_N^{(1)}$ and $\Gamma^{(0)} = \mathbb{Z}_N^{(0)}$ in the *A*-model. While the $\mathbb{Z}_N^{(1)}$ symmetry acts on the Hilbert space by $N$-th roots of unity, the $\mathbb{Z}_N^{(0)}$ symmetry organises the Bethe vacua into orbits whose lengths are divisors of $N$. There exists a unique fixed point if and only if the t' Hooft anomaly vanishes.

The gauging pattern of the $\mathbb{Z}_N$ centre symmetry is determined by the subgroups of $\mathbb{Z}_N$, which are labelled by the divisors $d$ of $N$. For instance, for $N$ a prime number and with

$K \in N\mathbb{Z}$, so that the non-anomalous $\mathbb{Z}_N$ has no non-trivial subgroups, we easily determine the 3d Witten index of the fully gauged theory $PSU(N)_K \equiv SU(N)_K/\mathbb{Z}_N$ to be:

$$\mathbf{I}_{\mathrm{W}}[PSU(N)_K] = \frac{1}{N^2}\left[\binom{K-1}{N-1} - 1 + N^3\right]. \tag{34}$$

This index is always an integer, since it can be written as a trace over the $PSU(N)_K$ Hilbert space on $T^2$, and we can prove this fact using number-theoretic identities. We find a much richer structure for general values of $N$, since we can consider the discrete gauging of any non-anomalous subgroup $\mathbb{Z}_r \subset \mathbb{Z}_N$. We study in detail the insertion of all topological operators on $\Sigma_g \times S^1$, for all values of $N$ and $K$.

**$SU(N)_K$ at genus 1**    The Witten index for the gauged theory can be obtained by enumerating the fixed points under all non-anomalous subgroups $\mathbb{Z}_d \subseteq \mathbb{Z}_N$. Three-dimensional modularity implies many relations amongst the expectation values of topological lines inserted on 1-cycles of $T^3$. These relations can be affected by a 3d modular anomaly, which we determine explicitly. This anomaly is a sign that only depends on the arithmetic properties of $N$, $K$ and $d$, we discuss its the relation to spin structures, quantisation conditions of WZW models and mixed 1-form/gravity anomalies. For instance, it allows us to determine explicitly the number of Bethe states for any $N$ and for $K \in N\mathbb{Z}$, as a sum over divisors of $N$:

$$\mathbf{I}_{\mathrm{W}}[PSU(N)_K] = \frac{1}{N^2}\sum_{d|N}\mathscr{J}_3^{N,K}(d)\binom{\frac{K}{d}-1}{\frac{N}{d}-1}. \tag{35}$$

Here, $\mathscr{J}_3^{N,K}$ is a refinement of Jordan's totient function $J_3$ which takes into account the 3d 'modular anomaly'.[1] When that 'modular anomaly' vanishes, $\mathscr{J}_3^{N,K} = J_3$ and then (35) is the proper Witten index of the $PSU(N)_K$ theory, while in the presence of the modular anomaly (that is, when the infrared $PSU(N)_k$ pure CS theory, with $k = K-N$, is a spin-TQFT) the actual Witten index needs to be computed more carefully [21].

A generalisation to the discrete gauging of a non-anomalous subgroup $\mathbb{Z}_r \subset \mathbb{Z}_N$ is straightforward, by truncating the sum (35) to the corresponding divisors. This thus computes the Witten index of the 3d $\mathcal{N} = 2$ CS theories for all groups $SU(N)/\mathbb{Z}_r$ whose simply-connected cover is $SU(N)$, at least when the infrared TQFT is bosonic. It can be checked that the index $\mathbf{I}_{\mathrm{W}}[SU(N)_K/\mathbb{Z}_r]$ is indeed an integer, as expected. Including a $\theta$-angle for $\Gamma^{(-1)} \cong \mathbb{Z}_r$, which corresponds to an holonomy for the dual $\Gamma_{3d}^{(0)} \cong \mathbb{Z}_r$ discrete symmetry in 3d, provides an interesting refinement [15, 16, 48, 49], and probes additional arithmetic properties of the integer $N$. We find that this modifies (35) by a further generalisation of Jordan's totient.

**$SU(N)_K$ at genus $g$**    Riemann surfaces of arbitrary genus $g$ admit $2g$ elementary topological operators associated with the zero-form symmetry $\Gamma^{(0)} \cong \mathbb{Z}_N$. By inserting the handle-gluing operator $\mathcal{H}$, we find that the genus-$g$ partition function for $K \in N\mathbb{Z}$ can be written as:

$$Z_{\Sigma_g \times S^1}\left[PSU(N)_K^{\theta_s}\right] = \frac{1}{N^{2g-1}}\sum_{d|N}J_{2g}(d)\sum_{\hat{u} \in \mathcal{S}_{\mathrm{BE}}^{(\frac{N}{d}\gamma_0)} \cap \mathcal{S}_{\mathrm{BE}}^{\theta_s}}\mathcal{H}(\hat{u})^{g-1}, \tag{36}$$

where $\gamma_0$ is a generator of $\mathbb{Z}_N$, $\mathcal{S}_{\mathrm{BE}}^{(\gamma)}$ is the set of Bethe vacua fixed under the action of $\gamma \in \mathbb{Z}_N$, and $\mathcal{S}_{\mathrm{BE}}^{\theta_s}$ is the set of Bethe vacua that span the smaller 'universe' indexed by the $\theta$-angle $\theta_s$.

---

[1]The Jordan's totient function itself is a generalisation of the more familiar Euler totient $\varphi$, also known as Euler's $\varphi$ function. The functions $J_k$ (for some integer $k$) can be understood as enumeration functions of interactions of $k$ instances of $\mathbb{Z}_N$ groups, similar to how $\varphi(d)$ counts the number of generators of $\mathbb{Z}_d$.

The genus-$g$ Jordan totient $J_{2g}$ has previously appeared in the study of Verlinde bundles over curves [36, 50–53]. Our result (36) generalises various partial results in the literature [19, 36, 50, 54].

**Comparison with abelian anyon condensation**   In the $SU(2)_K$ warm-up example of section 1.1, it was demonstrated that the one-form gauging can be performed directly in three dimensions using the 3d TQFT approach, and that the result on $T^3$ agrees with the $SO(3)_K$ $T^3$ index obtained by gauging both zero-form and one-form symmetries in 2d. In section 4.1, we revisit the calculation of the $PSU(N)_K$ Witten index from one-form gauging in the 3d TQFT description, also known as abelian anyon condensation. The $SU(N)_K$ $\mathcal{N} = 2$ supersymmetric CS theory has a spectrum of Wilson lines $W_{\boldsymbol{\lambda}}$ indexed by Young tableaux $\boldsymbol{\lambda}$. We discuss their fusion rules and identify the abelian anyons (*i.e.* $\mathbb{Z}_N$ topological lines), which allows us to implement the three-step gauging procedure discussed in [42]. For $N$ prime, one can understand each step as a simple modification of the $SU(N)_K$ index, and one recovers (34) as a result. For arbitrary values of $N$, the fusion with the abelian anyons furnishes a group action on the $SU(N)_K$ Wilson lines, and the 3d gauging partitions them into orbits. We express the $PSU(N)_K$ index as a sum over those orbits, and find precise numerical agreement with (35) for small values of $N$ and $K$.[2] This also clarifies the reason why we have a mixed gravitational/one-form symmetry anomaly in some cases, precisely when the condensing anyon is fermionic instead of bosonic. An explicit treatment of anyon condensation in that 'spin-TQFT' case appeared in [42, 43, 55], and we will explore this further in the 3d $A$-model language in future work [21].

**Including matter**   While we focussed on the pure $SU(N)_K$ CS theories in this work, the inclusion of chiral multiplets is straightforward. In general, the one-form symmetry of the $\widetilde{G}$ theory is the discrete subgroup of the centre of $\widetilde{G}$ that is preserved by the matter content. In section 5, we briefly study $U(1)_k$ theories with matter, as well as the $SU(N)_k$ Chern–Simons theory with adjoint matter. Models with an adjoint chiral multiplet are of particular interest since they provide equivariant generalisations of the Verlinde formula [19, 56], as has been studied in the past from various perspectives – see *e.g.* [10, 12, 13, 56–60].

This paper is organised as follows. In section 2, we present the general results on one-form symmetries of 3d $\mathcal{N} = 2$ supersymmetric gauge theories on $\Sigma_g \times S^1$. In sections 3 and 4, we study in much detail the 3d $\mathcal{N} = 2$ $SU(N)_K$ Chern–Simons theory. In section 5, we briefly discuss $U(1)$ and $SU(N)$ $\mathcal{N} = 2$ gauge theories with matter. Section 6 presents our conclusions and outlook. Some useful review material and the more technical computations are relegated to several appendices.

## 2   One-form symmetries in the 3d $A$-model

Consider a 3d $\mathcal{N} = 2$ supersymmetric field theory with a discrete one-form symmetry $\Gamma^{(1)}_{3d}$. In this work, we will specifically study 3d $\mathcal{N} = 2$ gauge theories, and the one-form symmetry will be (part of) the centre symmetry for some gauge group $G$. For instance, a gauge theory with gauge group $SU(N)$ and matter fields transforming in the adjoint representation will have a one-form symmetry $\Gamma^{(1)}_{3d} = \mathbb{Z}_N$.

---

[2]The exact equality between the results obtained using the two distinct methods is left as a conjecture.

## 2.1 Discrete symmetries in the 2d description

The 3d $A$-model (in the terminology of [8]) is a two-dimensional topologically-twisted $\mathcal{N} = (2,2)$ supersymmetric field theory that captures all the information about the twisted chiral ring $\mathcal{R}^{3d}$ of the 3d $\mathcal{N} = 2$ theory compactified on a circle, $S_A^1$, including twisted indices [10–12], correlation functions of half-BPS lines wrapping $S_A^1$ [13,61], and supersymmetric partition functions on Seifert 3-manifolds [7,9,62,63].

Let us start by explaining how the 3d one-form symmetry manifests itself in the two-dimensional description. We start with the 3d theory on $\Sigma \times S_A^1$, with the topological $A$-twist along a two-manifold $\Sigma$. We will mostly consider $\Sigma = \Sigma_g$, a compact closed Riemann surface of genus $g$. Here, translation along the circle $S_A^1$ is part of the 3d $A$-model supersymmetry algebra, and the corresponding momentum is the Kaluza-Klein (KK) charge [8]. In the 2d description on $\Sigma$, it manifests itself as a $U(1)_{KK}$ symmetry. In three dimensions, the operators charged under $\Gamma_{3d}^{(1)}$ are line operators, which are acted on by topological line operators [16].[3] The 3d one-form symmetry $\Gamma_{3d}^{(1)} \cong \Gamma$, for $\Gamma$ a finite abelian group, manifests itself as two distinct symmetries in 2d, namely a one-form and a zero-form symmetry:

$$\Gamma_{3d}^{(1)} \quad \longrightarrow \quad \Gamma^{(1)} \cong \Gamma, \qquad \Gamma^{(0)} \cong \Gamma. \tag{37}$$

Here, the abelian zero-form symmetry $\Gamma^{(0)}$ action is generated by topological lines with support on one-cycles $\mathcal{C} \subset \Sigma$, which act on local operators on $\Sigma$. These 2d local operators correspond to the 3d lines wrapping $S_A^1$. We will be particularly interested in such operators that also commute with the $A$-model supercharge – these are the twisted chiral operators $\mathscr{L} \in \mathcal{R}^{3d}$. The 2d one-form symmetry $\Gamma^{(1)}$, on the other hand, is generated by topological point operators that arise as the 3d topological lines wrapping $S_A^1$. We shall denote the topological line operators that implement $\Gamma^{(0)}$ by:

$$\mathcal{U}^\gamma(\mathcal{C}), \tag{38}$$

and the topological point operators that implement $\Gamma^{(1)}$ by:

$$\Pi^\gamma(p) \equiv \mathcal{U}^\gamma(S_A^1), \tag{39}$$

for every $\gamma \in \Gamma$. These topological operators are depicted in figure 1. The support of $\Pi^\gamma$ at a point $p \in \Sigma$ will be kept implicit in the following.

### 2.1.1 Hilbert spaces and Bethe vacua for $G = \widetilde{G}$

In the $A$-model description, we are mainly concerned with the supersymmetric ground states of the 2d $\mathcal{N} = (2,2)$ effective field theory. These are indexed by the so-called Bethe vacua, $\hat{u}$, which should be viewed as states in a (cohomological) 2d topological quantum field theory (TQFT) that assigns the (ground-state) Hilbert space $\mathcal{H}_{S^1}$ to the spatial circle:

$$\mathcal{H}_{S^1} \cong \text{Span}_{\mathbb{C}}\left\{ |\hat{u}\rangle \, \middle| \, \hat{u} \in \mathcal{S}_{BE} \right\}. \tag{40}$$

In order to discuss the Bethe vacua more explicitly, let us focus on a 3d $A$-model that arises from a 3d $\mathcal{N} = 2$ gauge theory with compact gauge group $G$. For now, let us assume that the fundamental group of $G = \widetilde{G}$ is a free abelian group:

$$\pi_1(\widetilde{G}) \cong \mathbb{Z}^{n_T}, \tag{41}$$

---

[3]Hence, the invertible topological lines that implement the one-form symmetry are a subset of the set of all line operators of the 3d theory, and symmetry operators can themselves be charged, as we will review.

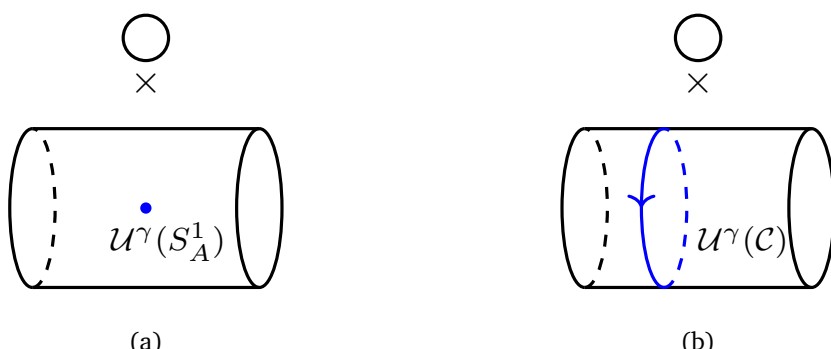

(a)  (b)

Figure 1: Depiction of the $\Gamma^{(1)}$ and $\Gamma^{(0)}$ symmetry operators $\mathcal{U}^{\gamma}(S_A^1) \equiv \Pi^{\gamma}$ and $\mathcal{U}^{\gamma}(\mathcal{C})$, respectively, on $\Sigma \times S_A^1$. Here, $\mathcal{U}^{\gamma}(S_A^1)$ wraps the $S_A^1$ factor in the 3d geometry, depicted here by a small circle, and it is thus a local operator on $\Sigma$. The topological line $\mathcal{U}^{\gamma}(\mathcal{C})$ is supported on a cycle $\mathcal{C}$ on $\Sigma$, in general. Here $\Sigma$ is the cylinder, which is the relevant configuration to discuss the Hilbert space $\mathcal{H}_{S^1}$ of the $A$-model.

for some non-negative integer $n_T$. More concretely, $\widetilde{G}$ is a product of simply-connected groups and of $U(N)$ factors, so that each $U(N)$ factor contributes a $\mathbb{Z}$ factor in (41), and $n_T$ is the number of topological $\mathfrak{u}(1)$ symmetries of the 3d gauge theory. Then the Bethe vacua are essentially given by the critical point of some effective twisted superpotential $\mathcal{W}(u)$ in some Coulomb-branch description, where $u$ denote 2d scalars in abelianised vector multiplets. We are interested in the one-form symmetry of this 3d gauge theory, which is an ordinary centre symmetry:

$$\Gamma \subseteq Z(\widetilde{G}). \tag{42}$$

Namely, $\Gamma$ is the maximal subgroup of the centre of $\widetilde{G}$ which is preserved by the chiral multiplets and by the CS interactions.

The basic construction of the 3d $A$-model with a gauge group $\widetilde{G}$ is summarised in appendix A, which also spells out our conventions in more details. For our present purpose, let us note that:[4]

$$u \in \mathfrak{t}_{\mathbb{C}}, \qquad \mathfrak{m} \in \Lambda_{\mathrm{mw}}^{\widetilde{G}} \subset \mathfrak{t}. \tag{43}$$

Namely, the Coulomb-branch scalar $u$ is valued in the complexified Cartan algebra of $\widetilde{G}$, $\mathfrak{t}_{\mathbb{C}}$, and the magnetic fluxes $\mathfrak{m}$ are valued in the magnetic weight lattice of $\widetilde{G}$, which is a discrete sublattice of the Lie algebra. Large gauge transformations along $S_A^1$ lead to the identifications:

$$u \sim u + \mathfrak{m}, \qquad \forall \mathfrak{m} \in \Lambda_{\mathrm{mw}}^{\widetilde{G}}, \tag{44}$$

hence the classical Coulomb branch is given by $\mathfrak{t}_{\mathbb{C}}/\Lambda_{\mathrm{mw}}^{\widetilde{G}} \cong (\mathbb{C}^*)^{\mathrm{rank}(\widetilde{G})}$. We pick a basis $\{e^a\}$ of $\mathfrak{t}$ such that $u = u_a e^a$ and $u_a \sim u_a + n_a$, $n_a \in \mathbb{Z}$, under any large gauge transformation (44). The *gauge flux operators* are defined in terms of the effective twisted superpotential of the gauge theory on $\mathbb{R}^2 \times S^1$:

$$\Pi_a(u, v) \equiv \exp\left(2\pi i \frac{\partial \mathcal{W}(u, v)}{\partial u_a}\right), \qquad a = 1, \ldots, \mathrm{rank}(\widetilde{G}), \tag{45}$$

with our conventions for $u_a$ as outlined above and in appendix A. Then, the Bethe vacua correspond to elements of the set of allowable Bethe solutions:

$$\mathcal{S}_{\mathrm{BE}} \equiv \left\{ \hat{u} \in \mathfrak{t}/\Lambda_{\mathrm{mw}}^{\widetilde{G}}, \, \middle| \, \Pi_a(\hat{u}, v) = 1 \,, \forall a \quad \text{and} \quad w \cdot \hat{u} \neq \hat{u} \,, \forall w \in W_{\widetilde{G}} \right\} / W_{\widetilde{G}}, \tag{46}$$

---

[4] See appendix B for a definition of the relevant lattices.

where $W_{\widetilde{G}}$ is the Weyl group of $\widetilde{G}$. Here, recall that we need to exclude putative solutions to the Bethe equations, $\{\Pi_a = 1 , \forall a\}$, which are not acted on freely by $W_{\widetilde{G}}$. These Bethe equations can be written more covariantly as:

$$\Pi(u,v)^{\mathfrak{m}} \equiv \exp\left(2\pi i\mathfrak{m}\frac{\partial \mathcal{W}}{\partial u}\right) = 1\,, \qquad \forall \mathfrak{m} \in \Lambda_{\mathrm{mw}}^{\widetilde{G}}\,. \tag{47}$$

We denote the Bethe vacua by $\hat{u}$, corresponding to the allowed solutions to the Bethe equations modulo the action of the Weyl group.

Let us now discuss how the symmetries (37) are realised in the $A$-model description.

**Action of $\Gamma^{(1)}$ on $\mathcal{H}_{S^1}$**  The Bethe vacua diagonalise the topological operator $\mathcal{U}^\gamma(S_A^1)$ defined in (39). Indeed, the insertion of the 3d topological line along $S_A^1$ is equivalent to the insertion of a background flux $\gamma$ along $\Sigma$ – that is, a non-trivial background gauge field for $\Gamma^{(1)}$ corresponds to a non-trivial $\widetilde{G}/\Gamma$ bundle over $\Sigma$ which cannot be lifted to a $\widetilde{G}$ bundle. In the $A$-model description, this operator is given explicitly by a flux operator:

$$\Pi(u,v)^\gamma \equiv \exp\left(2\pi i\gamma\frac{\partial \mathcal{W}}{\partial u}\right)\,, \qquad \gamma \in \Lambda_{\mathrm{mw}}^{\widetilde{G}/\Gamma} \subseteq \Lambda_{\mathrm{mw}}^{\mathfrak{g}}\,, \tag{48}$$

where $\gamma$ corresponds to a magnetic flux valued in the larger GNO lattice $\Lambda_{\mathrm{mw}}^{\widetilde{G}/\Gamma}$, with:

$$\Lambda_{\mathrm{mw}}^{\widetilde{G}/\Gamma} \supset \Lambda_{\mathrm{mw}}^{\widetilde{G}}\,, \qquad \Gamma \cong \Lambda_{\mathrm{mw}}^{\widetilde{G}/\Gamma}/\Lambda_{\mathrm{mw}}^{\widetilde{G}}\,. \tag{49}$$

See also appendix B for a definition of these lattices. On Bethe vacua, the Bethe equations (47) ensure that:

$$\Pi(\hat{u})^{\gamma+\mathfrak{m}} = \Pi(\hat{u})^\gamma\,, \qquad \forall \mathfrak{m} \in \Lambda_{\mathrm{mw}}^{\widetilde{G}}\,. \tag{50}$$

The one-form symmetry operators then act on Bethe vacua as:

$$\Pi^\gamma|\hat{u}\rangle = \Pi(\hat{u})^\gamma|\hat{u}\rangle\,, \qquad \Pi(\hat{u})^\gamma \in \mathbb{C}^*\,, \tag{51}$$

at fixed flavour parameters, for $\gamma \in \Gamma^{(1)}$. In fact, since we also know that, due to the Bethe equations, $\Pi(\hat{u})^{\gamma n} = 1$ for some positive integer $n \le |\Gamma|$, the one-form symmetry acts on Bethe vacua by multiplication by a root of unity. This is in agreement with the general discussion of 2d TQFTs in [19].

It is important to note that the definition for $\Pi^\gamma$ given in (48) is ambiguous, because the twisted superpotential suffers from the ambiguity [8]:

$$\mathcal{W} \to \mathcal{W} + \rho(u)\,, \qquad \rho \in \Lambda_{\mathrm{w}}^{\widetilde{G}}\,. \tag{52}$$

Any such linear shift corresponds to multiplying the flux operator (48) by a phase:

$$\Pi^\gamma \to \chi_\rho(\gamma)\Pi^\gamma\,, \qquad \chi_\rho \equiv e^{2\pi i\rho} \in \hat{\Gamma}^{(1)}\,, \tag{53}$$

where $\hat{\Gamma}^{(1)}$ denotes the Pontryagin dual:

$$\hat{\Gamma}^{(1)} \equiv \mathrm{Hom}(\Gamma^{(1)}, U(1))\,. \tag{54}$$

We will discuss how to fix this ambiguity in section 2.2.

The action of the 1-form symmetry splits the Hilbert space into sectors:

$$\mathcal{H}_{S^1} = \bigoplus_{\chi \in \hat{\Gamma}^{(1)}} \mathcal{H}_{S^1}^\chi\,, \qquad \mathcal{H}_{S^1}^\chi \equiv \mathrm{Span}_{\mathbb{C}}\left\{|\hat{u}\rangle \,\Big|\, \hat{u} \in \mathcal{S}_{\mathrm{BE}}^\chi\right\}\,, \tag{55}$$

where each sector corresponds to the Bethe vacua that return a specific phase $\chi(\gamma)$ for $\Pi^\gamma$ in (51). More precisely, for each $\chi \in \hat{\Gamma}^{(1)}$, we define:

$$\mathcal{S}_{\text{BE}}^\chi \equiv \left\{ \hat{u} \in \mathcal{S}_{\text{BE}} \;\middle|\; \Pi(\hat{u}, v)^\gamma = \chi(\gamma) \, , \; \forall \gamma \in \Gamma^{(1)} \right\}, \tag{56}$$

and we then have that $\mathcal{S}_{\text{BE}} = \oplus_\chi \mathcal{S}_{\text{BE}}^\chi$. This decomposition of the Bethe vacua will be useful below. Let us also note that the existence of a 1-form symmetry in a 2d QFT always implies a so-called decomposition [45, 46] of the theory into disjoint sectors.

**Action of $\Gamma^{(0)}$ on $\mathcal{H}_{S^1}$**    Let $\mathcal{U}^\gamma(\mathcal{C})$ be a topological line operator wrapping the cylinder, as shown in figure 1. To understand its action on the Bethe vacua, recall that $x_a \equiv e^{2\pi i u_a}$ can be seen as complexified holonomies of a maximal torus $\prod_a U(1)_a \subset \widetilde{G}$ along $S_A^1$, and that any Wilson loop in the representation $\mathfrak{R}$ of $\widetilde{G}$ is represented in the $A$-model by:

$$W_{\mathfrak{R}} = \sum_{\rho \in \mathfrak{R}} e^{2\pi i \rho(u)}. \tag{57}$$

Then, the insertion of $\mathcal{U}^\gamma(\mathcal{C})$ acts on Wilson loops wrapping $S_A^1$ as the 3d centre symmetry, which precisely shifts $u$ to $u + \gamma$, for $\gamma \in \Lambda_{\text{mw}}^{\widetilde{G}/\Gamma}$ – this only depends on $\gamma \in \Gamma$ because of (44). The action:

$$\mathcal{U}^\gamma(\mathcal{C}) \,:\, \mathfrak{t}_{\mathbb{C}} \to \mathfrak{t}_{\mathbb{C}} \,:\, u \mapsto u + \gamma, \tag{58}$$

is a symmetry of the Bethe equations for any $\gamma \in \Gamma^{(0)}$, by assumption (since $\Gamma$ is precisely the subgroup of $Z(\widetilde{G})$ that is preserved by the matter content and by the CS levels). Hence we have:

$$\mathcal{U}^\gamma(\mathcal{C})|\hat{u}\rangle = |\hat{u} + \gamma\rangle. \tag{59}$$

Therefore, the one-form symmetry element $\gamma \in \Gamma^{(0)}$ acts as a permutation of the Bethe vacua, as expected on general grounds [19].

**Twisted sector Hilbert spaces $\mathcal{H}_{S^1}^{(\delta)}$**    The twisted sector for any $\delta \in \Gamma^{(0)}$ corresponds to adding a topological line along the time direction on the cylinder $\Sigma \cong \mathbb{R}_\tau \times S^1$. The twisted-sector ground states are Bethe vacua which satisfy the additional twist condition:

$$u + \delta = w \cdot u, \tag{60}$$

for some $w \in W_{\widetilde{G}}$. These states, denoted by $|\hat{u}; \delta\rangle$, form a basis for the twisted Hilbert space:

$$\mathcal{H}_{S^1}^{(\delta)} \cong \text{Span}_{\mathbb{C}} \left\{ |\hat{u}; \delta\rangle \;\middle|\; \hat{u} \in \mathcal{S}_{\text{BE}} \, , \hat{u} + \delta \sim \hat{u} \right\}. \tag{61}$$

Of course, we have $\mathcal{H}_{S^1}^{(0)} = \mathcal{H}_{S^1}$ for $\delta = 0$ the zero element. For future reference, let us define the set of Bethe solutions that enter in (61) as:

$$\mathcal{S}_{\text{BE}}^{(\gamma)} \equiv \left\{ \hat{u} \in \mathcal{S}_{\text{BE}} \;\middle|\; \hat{u} + \gamma \sim \hat{u} \right\}, \qquad \gamma \in \Gamma^{(0)}, \tag{62}$$

where $\hat{u} \sim \hat{u}'$ means that the respective Weyl-group orbits of the solutions $\hat{u}$ and $\hat{u}'$ are equal. Note that (62) should not to be confused with (56).

     The discrete group $\Gamma^{(0)}$ acts on the set of all Bethe vacua, $\mathcal{S}_{\text{BE}}$, as in (59). Let $\text{Orb}(\hat{u}) \subseteq \mathcal{S}_{\text{BE}}$ denote the orbit of the vacuum $\hat{u}$ under the action of $\Gamma^{(0)}$, and let $\text{Stab}(\hat{u}) \subseteq \Gamma^{(0)}$ denote the stabiliser of $\hat{u}$ by $\Gamma^{(0)}$. The orbit-stabiliser theorem gives us the cardinality of $\text{Stab}(\hat{u})$:

$$|\text{Stab}(\hat{u})| = \frac{\left|\Gamma^{(0)}\right|}{|\text{Orb}(\hat{u})|}, \tag{63}$$

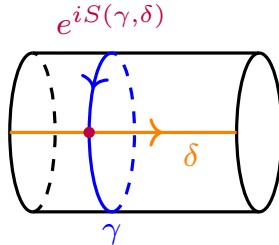

(a) Intersection of two $\Gamma^{(0)}$ lines.

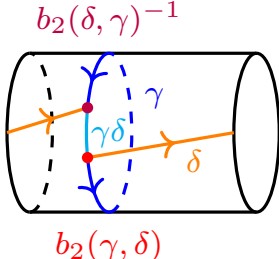

(b) Resolution into trivalent vertices.

Figure 2: The 2d SPT phase arising from the intersection of the topological lines $\mathcal{U}^\gamma$ and $\mathcal{U}^\delta$ on the cylinder, see (66). The phase is obtained by resolving the intersection into two trivalent junctions, where we assign the phase $b_2(\gamma, \delta)$ to each junction (with some customary orientation). That $b_2$ is a group cohomology class follows from the associativity of fusion and from gauge invariance.

which is the number of twisted sectors in which $\hat{u}$ appears:

$$|\hat{u}; \delta\rangle \in \mathcal{H}_{S^1}^{(\delta)}, \qquad \forall \delta \in \text{Stab}(\hat{u}). \tag{64}$$

Finally, let us discuss the action of $\Gamma^{(0)}$ on the twisted sector $\mathcal{H}_{S^1}^{(\delta)}$. In general, the action of $\mathcal{U}^\gamma(\mathcal{C})$ on the twisted-sector states may take the form:

$$\mathcal{U}^\gamma(\mathcal{C})|\hat{u}; \delta\rangle = e^{iS_{\text{SPT}}(\gamma, \delta)}|\hat{u} + \gamma; \delta\rangle, \tag{65}$$

where $S_{\text{SPT}}$ is a 2d SPT phase for $\Gamma^{(0)}$ (see *e.g.* [64,65] for recent discussions). More concretely, the phase this introduces in (65) is given in terms of a $U(1)$-valued group 2-cocycle $b_2$, as follows:

$$e^{iS_{\text{SPT}}(\gamma, \delta)} = \frac{b_2(\gamma, \delta)}{b_2(\delta, \gamma)}, \qquad [b_2] \in H^2\big(\Gamma^{(0)}, U(1)\big). \tag{66}$$

In the two-dimensional description, this phase arises because of the intersection of the topological lines $\mathcal{U}^\gamma$ and $\mathcal{U}^\delta$ on the cylinder, as shown in figure 2; see *e.g.* [66] for a recent discussion. Note that, in the 3d description wherein these lines arise as topological operators for a 3d one-form symmetry, they could be separated along $S_A^1$. In this work, we will only consider examples where the phase (66) is necessarily trivial, $e^{iS_{\text{SPT}}} = 1$; indeed, we will briefly focus on $\Gamma = \mathbb{Z}_N$, in which case $H^2(\mathbb{Z}_N, U(1)) \cong 1$ (see also [67]). We hope to discuss cases where this 2d SPT phase can be non-trivial in future work.

### 2.1.2 't Hooft anomalies and vacuum structure

The one-form symmetry $\Gamma_{3d}^{(1)}$ of the 3d theory can have a non-trivial 't Hooft anomaly captured by a four-dimensional anomaly theory:

$$S_{4d}^{\text{anom}}[B] = 2\pi \int_{\mathfrak{M}_4} \mathcal{P}_{\mathfrak{a}}(B), \tag{67}$$

where $\mathcal{P}_{\mathfrak{a}}(B)$ is an anomaly-dependent modification of the Pontryagin square $\mathcal{P}(B)$ of the one-form background gauge field in 3d extended onto the four-manifold $\mathfrak{M}_4$ with boundary $\Sigma \times S^1$, $B \in H^2(\mathfrak{M}_4, \Gamma)$. The construction of $\mathcal{P}(B)$ is reviewed in appendix C. More concretely, we

pick a set of generator for $\Gamma$ such that $\Gamma \cong \bigoplus_i \mathbb{Z}_{N_i}$ and $B$ decomposes as $B = \sum_i B_i$ with $B_i \in H^2(\mathfrak{M}_4, \mathbb{Z}_i)$, in which case the anomaly theory (67) takes the concrete form [42]:

$$S_{4d}^{anom}[B] = 2\pi \sum_i \frac{\mathfrak{a}_{ii}}{2N_i} \int_{\mathfrak{M}_4} \mathcal{P}(B_i) + 2\pi \sum_{i<j} \frac{\mathfrak{a}_{ij}}{\gcd(N_i, N_j)} \int_{\mathfrak{M}_4} B_i \cup B_j, \tag{68}$$

where the anomaly is captured by the symmetric matrix $\mathfrak{a}_{ij}$ with integer coefficients:

$$\mathfrak{a}_{ij} \in \mathbb{Z}_{\gcd(N_i, N_j)}. \tag{69}$$

Such an anomaly arises because some of the topological lines can themselves be charged under $\Gamma_{3d}^{(1)}$ [16]. Note also that we consider the 3d $\mathcal{N} = 2$ theory on 3-manifolds with an explicit choice of spin structure, which then extends to a choice of spin structure on $\mathfrak{M}_4$ – this is important for (68) to be well-defined with the periodicities (69) [42]. In the two-dimensional description on $\Sigma$, the 't Hooft anomaly is realised as a mixed anomaly between $\Gamma^{(1)}$ and $\Gamma^{(0)}$, corresponding to the 3d anomaly theory on the three-manifold $\mathfrak{M}_3$ with boundary $\Sigma$:

$$S_{3d}^{anom}[B, C] = 2\pi \sum_{i,j} \frac{\mathfrak{a}_{ij}}{\gcd(N_i, N_j)} \int_{\mathfrak{M}_3} C_i \cup B_j, \tag{70}$$

where now $C \in H^1(\mathfrak{M}_3, \Gamma)$, $B \in H^2(\mathfrak{M}_3, \Gamma)$, and we have expanded $C = \sum_i C_i$ and $B = \sum_i B_i$ as above. In appendix C, we obtain this mixed 't Hooft anomaly by considering the continuum version of the anomaly theory [42,68] and dimensionally reducing along $S^1$ on $\mathfrak{M}_4 = \mathfrak{M}_3 \times S^1$, with $\partial \mathfrak{M}_3 = \Sigma_g$.

In two space-time dimensions, the one-form symmetry $\Gamma^{(1)}$ can never be spontaneously broken [16], and it is therefore preserved in each vacuum. On the other hand, the zero-form symmetry $\Gamma^{(0)}$ in the vacuum $\hat{u}$ is spontaneously broken to the subgroup $\text{Stab}(\hat{u}) \subseteq \Gamma^{(0)}$. The 't Hooft anomaly must be matched by the 2d low-energy description, which constrains the structure of the $\Gamma^{(0)}$-orbits of Bethe vacua – we will discuss this momentarily.

The mixed anomaly (70) arises because the one-form symmetry operators $\Pi^{\gamma_{(1)}}$ can be charged under $\Gamma^{(0)}$, and vice versa. (In the 3d description, two topological lines that implement $\Gamma_{3d}^{(1)}$ can link non-trivially.) At the level of the Hilbert space $\mathcal{H}_{S^1}$, the $\Gamma^{(0)}$ and $\Gamma^{(1)}$ charge operators need not commute. Instead, we have a projective representation of $\Gamma^{(0)} \times \Gamma^{(1)}$ on $\mathcal{H}_{S^1}$, with the twisted commutation relations:

$$\Pi^{\gamma_{(1)}} \mathcal{U}^{\gamma_{(0)}} = e^{2\pi i \mathcal{A}(\gamma_{(0)}, \gamma_{(1)})} \mathcal{U}^{\gamma_{(0)}} \Pi^{\gamma_{(1)}}, \tag{71}$$

determined by the anomaly:

$$\mathcal{A} : \Gamma^{(0)} \times \Gamma^{(1)} \to \mathbb{R}/\mathbb{Z}. \tag{72}$$

Given a choice of generators $\gamma_{(0),i}$ and $\gamma_{(1),j}$ for $\Gamma^{(0)} \cong \bigoplus_i \mathbb{Z}_{N_i}^{(0)}$ and of $\Gamma^{(1)} \cong \bigoplus_j \mathbb{Z}_{N_j}^{(j)}$, respectively, we can expand any group element as:

$$\gamma_{(0)} = \sum_i n_i \gamma_{(0),i}, \qquad \gamma_{(1)} = \sum_j m_j \gamma_{(1),j}, \tag{73}$$

for $n_i, m_i \in \mathbb{Z}_{N_i}$. Then the anomaly (72) is related to the coefficients $\mathfrak{a}_{ij}$ appearing in the 3d anomaly theory (70) according to:

$$\mathcal{A}(\gamma_{(0)}, \gamma_{(1)}) = \sum_{i,j} \frac{n_i m_j \mathfrak{a}_{ij}}{\gcd(N_i, N_j)} \mod 1. \tag{74}$$

For a given 3d $\mathcal{N} = 2$ gauge theory, this anomaly is easily computed using the $A$-model description. Indeed, from (51) and (59), we find:[5]

$$[\Pi^{\gamma_{(1)}}, \mathcal{U}^{\gamma_{(0)}}]|\hat{u}\rangle = \frac{\Pi(\hat{u})^{\gamma_{(1)}}}{\Pi(\hat{u})^{\gamma_{(1)}}}|\hat{u}\rangle = e^{2\pi i \mathcal{A}(\gamma_{(0)},\gamma_{(1)})}|\hat{u}\rangle,\tag{75}$$

and therefore:

$$\mathcal{A}(\gamma_{(0)},\gamma_{(1)}) = \gamma_{(1)}\left(\frac{\partial \mathcal{W}(u+\gamma_{(0)})}{\partial u} - \frac{\partial \mathcal{W}(u)}{\partial u}\right) \mod 1,\tag{76}$$

which in turn is independent of $u$ and only depends on the bare CS levels of the UV theory. We will discuss this point further in section 5.2, based on the explicit form of the effective twisted superpotential which we review in appendix A.

Given the anomaly (72), we may also define a homomorphism $\phi_{\mathcal{A}}$ from $\Gamma^{(0)}$ to the Pontryagin dual (54) of $\Gamma^{(1)}$, given by:

$$\phi_{\mathcal{A}} : \Gamma^{(0)} \to \hat{\Gamma}^{(1)} : \gamma \mapsto e^{2\pi i \mathcal{A}(\gamma,-)}.\tag{77}$$

Now, consider coupling the $A$-model to background gauge fields $(C,B)$ for $\Gamma^{(0)} \times \Gamma^{(1)}$, which is equivalent to inserting the topological symmetry operators on $\Sigma$. The action of $\Gamma^{(0)}$ on the 2d vacua comes with a gauge transformation $C \to C + \delta\lambda$, which generates the anomalous term $\lambda\mathfrak{a}\int_{\Sigma} B$, schematically. This anomaly can only be matched if distinct vacua in a given $\Gamma^{(0)}$ orbit are stacked with distinct SPT phases for $\Gamma^{(1)}$. Fixing some vacuum $\hat{u}$ to have the trivial phase, the other vacua $\hat{u} + \gamma$ in $\mathrm{Orb}(\hat{u})$ will have the SPT phases:

$$e^{iS_{\mathrm{SPT}}^{(\gamma)}(B)} = \phi_{\mathcal{A}}(\gamma).\tag{78}$$

Note that which vacua is set to be the trivial SPT phase is immaterial, because we can always shift all vacua by a common SPT phase by adding the corresponding counterterm in the UV. Thus, we explicitly see how the anomaly constrains the vacuum structure.

The homomorphism $\phi_{\mathcal{A}}$ defines a subgroup $\ker\phi_{\mathcal{A}} \leq \Gamma^{(0)}$. The associated quotient group:

$$\Xi^{(0)} \equiv \Gamma^{(0)}/\ker\phi_{\mathcal{A}},\tag{79}$$

labels the distinct SPT phases. In general, $\Gamma^{(0)}$ is a direct sum of cyclic groups, and the distinct SPT phases may not correspond to distinct Bethe vacua in each orbit, which is due to the mixing of the various cyclic subgroups of $\Gamma^{(0)}$. Nevertheless, we can classify all possible orbits. Let $\hat{u}$ be a vacuum with trivial SPT phase, and let $\gamma \in \Gamma^{(0)}$ be in the stabiliser group $\mathrm{Stab}(\hat{u})$, which means in particular that $\hat{u} + \gamma \sim \hat{u}$. The vacuum $\hat{u} + \gamma$ has a SPT phase (78), but since we started with a vacuum $\hat{u}$ that has a trivial phase, so does $\hat{u} + \gamma$. Therefore, $\phi_{\mathcal{A}}(\gamma) = 1$ and thus $\gamma \in \ker\phi_{\mathcal{A}}$. This argument is valid for any element $\gamma \in \mathrm{Stab}(\hat{u})$, and thus every stabiliser of a given Bethe vacuum $\hat{u}$ is a subgroup of the kernel:

$$\mathrm{Stab}(\hat{u}) \leq \ker\phi_{\mathcal{A}}.\tag{80}$$

This restricts the allowed orbit dimensions according to the orbit-stabiliser theorem (63). Given a subgroup $H \leq \ker\phi_{\mathcal{A}}$, there is a corresponding orbit of length $\Gamma^{(0)}/|H|$. Note that any such orbit dimension is an integer multiple of $|\Xi^{(0)}|$.

The extremal cases are the non-anomalous and the maximally anomalous case. The latter case corresponds to $\ker\phi_{\mathcal{A}} = 0$, so that $\mathrm{Stab}(\hat{u})$ is trivial for every vacuum $\hat{u}$. Hence all orbits must have maximal dimension $|\Gamma^{(0)}|$. At the other extreme, if the anomaly $\mathcal{A}$ vanishes

---

[5]Here $[a,b]$ denotes the multiplicative commutator, $[a,b] = a^{-1}b^{-1}ab$.

identically, then $\ker\phi_{\mathcal{A}} = \Gamma^{(0)}$, and $\mathrm{Stab}(\hat{u})$ can be any of the subgroups of $\Gamma^{(0)}$. For the trivial stabiliser, this results in orbits of length $|\Gamma^{(0)}|$. Meanwhile, for $\mathrm{Stab}(\hat{u}) = \Gamma^{(0)}$ we find orbits of length 1 – these are of course the fixed points under $\Gamma^{(0)}$, which can only occur in a theory with trivial 't Hooft anomaly. The generic case occurs when the full $\Gamma^{(0)}$ is anomalous but contains some non-anomalous subgroups.[6] Since $\ker\phi_{\mathcal{A}}$ is a subgroup of $\Gamma^{(0)}$, the allowed orbits are therefore those associated with the non-anomalous stabilisers.

For concreteness, let us consider the parameterisation $\Gamma \cong \bigoplus_i \mathbb{Z}_{N_i}$ as above. Then we have the SPT phases:

$$S_{\mathrm{SPT}}^{(\gamma)}(B_{\gamma_{(1)}}) = 2\pi\left(\sum_{i,j} n_i \mathrm{A}_{ij} m_j \mod 1\right), \qquad \mathrm{A}_{ij} \equiv \frac{\mathfrak{a}_{ij}}{\gcd(N_i, N_j)}, \tag{81}$$

for $\gamma = \sum_i n_i \gamma_{(0),i} \in \Gamma^{(0)}$ and a background gauge field $B_{\gamma_{(1)}}$ corresponding to $\gamma_{(1)} = \sum_j m_j \gamma_{(1),j} \in \Gamma^{(1)}$. Thus we have:

$$\ker\phi_{\mathcal{A}} \cong \left\{ \gamma = \sum_i n_i \gamma_{(0),i} \;\Bigg|\; \sum_i n_i \mathrm{A}_{ij} \in \mathbb{Z} \right\}. \tag{82}$$

This allows for explicit computations in any given example.

**The case $\Gamma = \mathbb{Z}_N$** Let us further discuss the special case of a cyclic group, $\Gamma = \mathbb{Z}_N$, on which we shall focus in the following sections. Denoting by the integers $n \in \mathbb{Z}_N^{(0)}$ and $m \in \mathbb{Z}_N^{(1)}$ the elements $\gamma_{(0)}$ and $\gamma_{(1)}$, respectively, we have the anomaly:

$$\mathcal{A}(\gamma_{(0)}, \gamma_{(1)}) = \frac{\mathfrak{a}nm}{N} \mod 1, \qquad \mathfrak{a} \in \mathbb{Z}_N, \tag{83}$$

determined by the integer $\mathfrak{a}$ (mod $N$). We also have the $\Gamma^{(1)}$ SPT phases:

$$e^{iS_{\mathrm{SPT}}^{(n)}(B)} = \phi_{\mathcal{A}}(n) = e^{2\pi i \frac{\mathfrak{a}n}{N}\int_\Sigma B} = e^{2\pi i \frac{\mathfrak{a}nm}{N}}, \tag{84}$$

for $B$ corresponding to $m \in \mathbb{Z}_N^{(1)}$. We then find that $\ker\phi_{\mathcal{A}} \cong \mathbb{Z}_{\gcd(\mathfrak{a},N)}$ and therefore:

$$\Xi^{(0)} \cong \mathbb{Z}_{d(\mathfrak{a},N)}, \qquad d(\mathfrak{a},N) \equiv \frac{N}{\gcd(\mathfrak{a},N)}. \tag{85}$$

The $\mathbb{Z}_N^{(0)}$-orbits spanned by the Bethe vacua therefore have dimensions that are integer multiples of $d(\mathfrak{a},N)$. Due to (80), the orders of the stabilisers and hence the orbits will be divisors of $N$. In the non-anomalous case, the orbit dimensions can be any divisor of $N$, while in the maximally anomalous case all orbits are of length $N$. We will discuss this more in detail in section 3.

## 2.2 Background gauge fields and expectation values of topological operators

Recall that the partition function of the $\widetilde{G}$ gauge theory $\mathcal{T}$ on $\Sigma_g \times S^1$ corresponds to the insertion of the handle-gluing operator $\mathcal{H}$ in the $A$-model, so that:

$$Z_{\Sigma_g \times S^1}^{[\mathcal{T}]} = \langle 1 \rangle_{\Sigma_g} = \sum_{\hat{u} \in \mathcal{S}_{\mathrm{BE}}} \langle \hat{u} | \mathcal{H}^{g-1} | \hat{u} \rangle = \sum_{\hat{u} \in \mathcal{S}_{\mathrm{BE}}} \mathcal{H}(\hat{u})^{g-1}. \tag{86}$$

We are interested in computing the insertion of topological operators for $\Gamma^{(1)} \times \Gamma^{(0)}$ on $\Sigma_g$. This is equivalent to turning on background gauge fields for these discrete symmetries.

---

[6]That is, non-anomalous subgroups $\widetilde{\Gamma}^{(0)} \subset \Gamma^{(0)}$ with respect to $\Gamma^{(1)}$, *i.e.* such that the mixed 't Hooft anomaly for $\widetilde{\Gamma}^{(0)}$-$\Gamma^{(1)}$ vanishes. We could more generally consider non-anomalous subgroups $\widetilde{\Gamma}^{(0)} \subset \Gamma^{(p)}$, $p = 0, 1$ (with $\widetilde{\Gamma}^{(0)}$ and $\widetilde{\Gamma}^{(1)}$ not necessarily isomorphic), such that their mixed anomaly vanishes, by an obvious generalisation of the discussion above.

**Insertion of the 1-form symmetry operators** Let us first consider the insertion of the local operator $\Pi^\gamma$ on $\Sigma_g$. We simply have:

$$\langle \Pi^\gamma \rangle_{\Sigma_g} = \sum_{\hat{u} \in \mathcal{S}_{\mathrm{BE}}} \langle \hat{u} | \Pi^\gamma \mathcal{H}^{g-1} | \hat{u} \rangle = \sum_{\hat{u} \in \mathcal{S}_{\mathrm{BE}}} \Pi(\hat{u})^\gamma \mathcal{H}(\hat{u})^{g-1} \, . \tag{87}$$

Using the decomposition (55)-(56), we can write this as:

$$\langle \Pi^\gamma \rangle_{\Sigma_g} = \sum_{\chi \in \hat{\Gamma}^{(1)}} \chi(\gamma) \sum_{\hat{u} \in \mathcal{S}_{\mathrm{BE}}^\chi} \mathcal{H}(\hat{u})^{g-1} \, . \tag{88}$$

This insertion is equivalent to turning on a background gauge field $B_\gamma$,

$$Z_{\Sigma_g \times S^1}^{[\mathcal{T}]}(B_\gamma) = \langle \Pi^\gamma \rangle_{\Sigma_g} \, , \qquad B_\gamma \in H^2(\Sigma_g, \Gamma^{(1)}) \, , \tag{89}$$

with:

$$\int_{\Sigma_g} B_\gamma = \gamma \, . \tag{90}$$

For later purpose, it will be useful to define the discrete $\theta$-angle and its pairing to $B_\gamma$:

$$\vartheta(\gamma) \equiv e^{-i(\theta, B_\gamma)} \, , \qquad (\theta, B_\gamma) \equiv \theta \int_{\Sigma_g} B_\gamma \, , \qquad \vartheta \in \hat{\Gamma}^{(1)} \, . \tag{91}$$

Here, $\theta$ is the background gauge field for a $(-1)$-form symmetry, which will appear as a dual symmetry once we gauge $\Gamma^{(0)}$, and $(\theta, B)$ is the canonical pairing on $\Sigma_g$. For instance, for $\Gamma^{(1)} = \mathbb{Z}_N$, we have $\theta \in \frac{2\pi}{N} \mathbb{Z}_N$.

**Insertion of the 0-form symmetry operators** Next, we consider the insertion of 0-form symmetry operators along the Riemann surface $\Sigma_g$. These topological line operators can wrap any of the generators of $H_1(\Sigma_g, \mathbb{Z}) \cong \mathbb{Z}^{2g}$, which we denote by $\mathcal{C}^i$ ($i = 1, \cdots, 2g$). Thus, an insertion is specified by a $\Gamma^{(0)}$-valued 1-cycle $\boldsymbol{\gamma}$ corresponding to:

$$[\boldsymbol{\gamma}] = \sum_{i=1}^{2g} \gamma_i [\mathcal{C}_i] \in H_1(\Sigma_g, \Gamma^{(0)}) \, , \tag{92}$$

which is determined by the $2g$ group elements $\gamma_i \in \Gamma^{(0)}$. We will also use the notation $\boldsymbol{\gamma} = (\gamma_i) \in \Gamma^{2g}$, by a slight abuse of notation. We will then denote the topological operator as:

$$\mathcal{U}^{\boldsymbol{\gamma}} \equiv \prod_{i=1}^{2g} \mathcal{U}^{\gamma_i}(\mathcal{C}_i) \, . \tag{93}$$

This insertion is equivalent to turning on a background gauge field $C_{\boldsymbol{\gamma}}$ for $\Gamma^{(0)}$, with:

$$[C_{\boldsymbol{\gamma}}] \equiv \mathrm{PD}[\boldsymbol{\gamma}] \in H^1(\Sigma_g, \Gamma^{(0)}) \, . \tag{94}$$

Here $\mathrm{PD}[\boldsymbol{\gamma}]$ denotes the cohomology class Poincaré dual to (92). For future reference, let us also choose a symplectic basis of $a$- and $b$-cycles as shown in figure 3, with the standard intersection pairing:

$$[a_k], [b_l] \in H_1(\Sigma_g, \mathbb{Z}) \, , \qquad k, l = 1, \ldots, g \, , \qquad [a_k] \cdot [b_l] = -[b_l] \cdot [a_k] = \delta_{kl} \, , \tag{95}$$

and $[a_k] \cdot [a_l] = [b_k] \cdot [b_l] = 0$. In this basis, we have $\boldsymbol{\gamma} \equiv (\gamma_{a,k}, \gamma_{b,l})$.

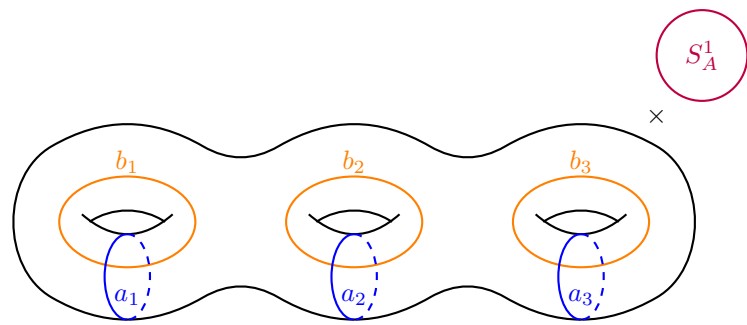

Figure 3: $a$- and $b$-cycles on the Riemann surface $\Sigma_g$, here depicted for genus $g = 3$.

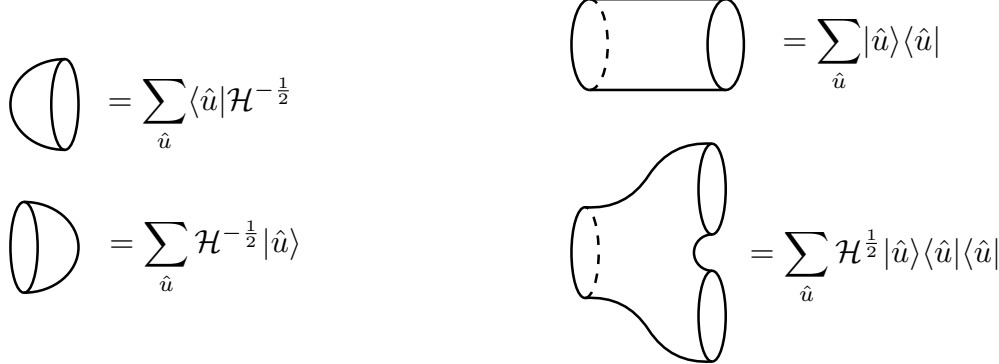

Figure 4: Operators in 2d TQFT corresponding to the cap, cylinder, and pair of pants. We can think as $\mathcal{H}^{\frac{1}{2}}$ as a formal square root of the handle-gluing operator $\mathcal{H}$, keeping in mind that we always obtain integer powers of $\mathcal{H}$ when computing observables on a closed $\Sigma$.

As reviewed in appendix A, the 3d $A$-model is a 2d TQFT. From this perspective, the twisted index (86) can be computed by basic surgery operations on the Riemann surface. The three basic ingredients are the cap, the cylinder and the pair of pants, to which the TQFT functor assigns states as summarised in figure 4. In the $\widetilde{G}$ theory with the $\Gamma^{(0)}$ symmetry, this TQFT prescription can be extended to compute the correlation functions of the topological line operators (93). The modified dictionary is shown in figure 5.

Let us first consider the case $g = 1$, the torus, with the notation $a_1 = \mathcal{C}$ and $b_1 = \widetilde{\mathcal{C}}$. Here $\mathcal{C}$ is considered as the spatial direction and $\widetilde{\mathcal{C}}$ is the Euclidean time direction. Setting $\boldsymbol{\gamma} = (\gamma_{a,1}, \gamma_{b,1}) = (\gamma, \delta)$, we have the general insertion:

$$Z^{[\mathcal{T}]}_{T^2 \times S^1}(C_{\boldsymbol{\gamma}}) = \left\langle \mathcal{U}^{\gamma}(\mathcal{C}) \, \mathcal{U}^{\delta}(\widetilde{\mathcal{C}}) \right\rangle_{T^2}, \tag{96}$$

which is of the form considered in figure 2a. We thus have the trace over the $\delta$-twisted sector:

$$Z^{[\mathcal{T}]}_{T^2 \times S^1}(C_{\boldsymbol{\gamma}}) = \sum_{\hat{u} \in \mathcal{S}^{(\delta)}_{\mathrm{BE}}} \langle \hat{u}; \delta | \mathcal{U}^{\gamma}(\mathcal{C}) | \hat{u}; \delta \rangle = \sum_{\hat{u} \in \mathcal{S}^{(\delta)}_{\mathrm{BE}}} \langle \hat{u}; \delta | \hat{u} + \gamma; \delta \rangle = \sum_{\hat{u} \in \mathcal{S}^{(\gamma,\delta)}_{\mathrm{BE}}} 1 = \left| \mathcal{S}^{(\gamma,\delta)}_{\mathrm{BE}} \right|, \tag{97}$$

with $\mathcal{S}^{(\delta)}_{\mathrm{BE}}$ defined as in (62), and we also defined $\mathcal{S}^{(\gamma,\delta)}_{\mathrm{BE}} \equiv \mathcal{S}^{(\gamma)}_{\mathrm{BE}} \cap \mathcal{S}^{(\delta)}_{\mathrm{BE}}$. This result is modular invariant, as expected for a 2d TQFT on $\Sigma_g$.

The generalisation to any $\Sigma_g$ with the operator (93) inserted can be derived using the 2d TQFT perspective sketched above [17, 19]. Following the decomposition of the Riemann

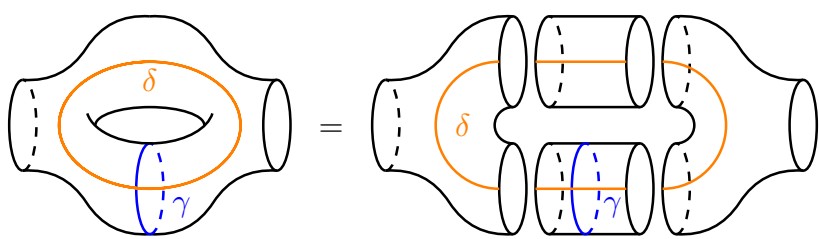

Figure 5: Operators in the TQFT corresponding to the cylinder and pair of pants with topological lines inserted. We could also consider the insertion of a line at the boundary of a cap, but this is topologically trivial; in this formalism, this follows from the fact that $\mathcal{H}(\hat{u} + \gamma) = \mathcal{H}(\hat{u})$, so that formally $\sum_{\hat{u}} \mathcal{H}^{-\frac{1}{2}} \mathcal{U}^{\gamma} |\hat{u}\rangle = \sum_{\hat{u}} \mathcal{H}^{-\frac{1}{2}} |\hat{u}\rangle$.

Figure 6: Decomposition of a handle-gluing operator with $\mathcal{U}^{\gamma}$ operator inserted.

surface $\Sigma_g$ with the insertions of $\Gamma^{(0)}$ symmetry operators as shown in figure 6, we find:

$$Z^{[\mathcal{T}]}_{\Sigma_g \times S^1}(C_{\gamma}) = \langle \mathcal{U}^{\gamma} \rangle_{\Sigma_g} = \sum_{\hat{u} \in \mathcal{S}^{(\gamma)}_{\mathrm{BE}}} \mathcal{H}(\hat{u})^{g-1}. \tag{98}$$

Here, for any set $(\gamma_i)$ of $n$ elements of $\Gamma^{(0)}$, we define the set of Bethe vacua $\mathcal{S}^{(\gamma_1, \cdots, \gamma_n)}_{\mathrm{BE}} \equiv \bigcap_{i=1}^{n} \mathcal{S}^{(\gamma_i)}_{\mathrm{BE}}$. Equivalently, let us denote by $\mathrm{H}^{(0)}_{\gamma} \subseteq \Gamma^{(0)}$ the smallest subgroup of $\Gamma^{(0)}$ that contains the $\gamma_i$'s – it is the smallest subgroup such that $[\gamma] \in H_1(\Sigma_g, \mathrm{H}_{\gamma})$. Then, we can define:

$$\mathcal{S}^{(\gamma)}_{\mathrm{BE}} \equiv \{\hat{u} \in \mathcal{S}_{\mathrm{BE}} \mid \hat{u} + \gamma^{(0)} \sim \hat{u} \,, \; \forall \gamma^{(0)} \in \mathrm{H}^{(0)}_{\gamma}\} \cong \mathcal{S}^{(\gamma_1, \cdots, \gamma_{2g})}_{\mathrm{BE}}. \tag{99}$$

Note that the expression (98) is invariant under large diffeomorphisms of $\Sigma_g$, which act as $Sp(2g, \mathbb{Z})$ transformations of $\gamma = (\gamma_{a,k}, \gamma_{b,l})$. Moreover, it is important to note that the correlator (98) only depends on the subgroup $\mathrm{H}^{(0)} \subseteq \Gamma^{(0)}$ and not on the specific insertion, namely:

$$\langle \mathcal{U}^{\gamma_1} \rangle_{\Sigma_g} = \langle \mathcal{U}^{\gamma_2} \rangle_{\Sigma_g} \,, \qquad \text{if} \quad \mathrm{H}^{(0)}_{\gamma_1} = \mathrm{H}^{(0)}_{\gamma_2}, \tag{100}$$

since $\mathcal{S}^{(\gamma_1)}_{\mathrm{BE}} = \mathcal{S}^{(\gamma_2)}_{\mathrm{BE}}$ in this case. In section 3, we will discuss these subgroups $\mathrm{H}^{(0)}_{\gamma}$ more explicitly in the case of $\Gamma^{(0)} \cong \mathbb{Z}_N$ a cyclic group.

**Mixed correlators** Finally, we may consider the simultaneous insertion of $\Gamma^{(1)}$ and $\Gamma^{(0)}$ background gauge fields, which gives us:

$$Z^{[\mathcal{T}]}_{\Sigma_g \times S^1}(B_{\gamma^{(1)}}, C_{\gamma^{(0)}}) = \left\langle \Pi^{\gamma^{(1)}} \mathcal{U}^{\gamma^{(0)}} \right\rangle_{\Sigma_g} = \sum_{\hat{u} \in \mathcal{S}^{(\gamma^{(0)})}_{\mathrm{BE}}} \Pi(\hat{u})^{\gamma^{(1)}} \mathcal{H}(\hat{u})^{g-1}. \tag{101}$$

Such mixed correlation functions necessarily vanish when the anomaly is non-trivial:

$$\left\langle \Pi^{\gamma^{(1)}} \mathcal{U}^{r^{(0)}} \right\rangle_{\Sigma_g} = 0, \qquad \text{if} \quad e^{2\pi i \mathcal{A}(\gamma^{(0)}, \gamma^{(1)})} \neq 1. \tag{102}$$

The expression (101) can also be written as:

$$\left\langle \Pi^{\gamma^{(1)}} \mathcal{U}^{r^{(0)}} \right\rangle_{\Sigma_g} = \sum_{\chi \in \hat{\Gamma}^{(1)}} \chi(\gamma^{(1)}) \sum_{\hat{u} \in \mathcal{S}_{\text{BE}}^{\chi} \cap \mathcal{S}_{\text{BE}}^{(\gamma^{(0)})}} \mathcal{H}(\hat{u})^{g-1}, \tag{103}$$

similarly to (88).

**3d modularity**   Correlation functions on $\Sigma_g \times S^1$ are not completely fixed by the above prescription, due to the ambiguity $\Pi^{\gamma} \to \chi_{\rho}(\gamma) \Pi^{\gamma}$ of the gauge flux operator from linear shifts of the twisted superpotential (53). A practical way to resolve this ambiguity is to demand modularity when inserting 3d lines on the three-torus. As before, for $g = 1$ we can choose $\mathcal{C}$ to be the spatial direction on $T^2$, and compare the correlator $\langle \chi_{\rho}(\gamma) \Pi^{\gamma} \rangle_{T^2}$ with $\langle \mathcal{U}^{\gamma}(\mathcal{C}) \rangle_{T^2}$. The ambiguity $\chi_{\rho}(\gamma)$ of the flux operator can be fixed by identifying those correlators (88) and (97),

$$\chi_{\rho}(\gamma) \sum_{\chi \in \hat{\Gamma}^{(1)}} \chi(\gamma) |\mathcal{S}_{\text{BE}}^{\chi}| = |\mathcal{S}_{\text{BE}}^{(\gamma)}|, \qquad \forall \gamma \in \Gamma. \tag{104}$$

Note that the RHS is a positive integer. If a solution $\chi_{\rho} \in \hat{\Gamma}^{(1)}$ exists, then it fixes a normalisation of the gauge flux operator for a given theory.

On physical grounds, we expect that a solution to (104) exist. Mathematically, the existence of a solution to (104) is not clear at all. As we show in section 3 for the case of the $SU(N)_K$ $\mathcal{N} = 2$ CS theory, the positive integers $|\mathcal{S}_{\text{BE}}^{(\gamma)}|$ depend intricately on the elements $\gamma$, and merely solving the equation for a generator $\gamma_0$ of $\Gamma$ – if one exists – may not be enough to determine $\chi_{\rho}$.[7] In section 3, we study in some detail the 3d modularity for the $\mathcal{N} = 2$ $SU(N)_K$ CS theory, and we find that a normalisation $\chi_{\rho}$ exists for all values of $N$ and $K$. We also encounter an interesting modular anomaly in this case; this will be discussed in section 3.2. This kind of 'modular anomaly' arises whenever the 3d topological lines for $\Gamma_{3d}^{(1)}$ have non-trivial braiding with the transparent line $\psi$ that couples the spin structure of $\mathcal{M}_3 = \Sigma_g \times S^1$ [21].

## 2.3   Gauging the 1-form symmetry $\Gamma^{(1)}$

Let us now gauge the $\Gamma^{(1)}$ symmetry – such a symmetry is always non-anomalous in 2d. This simply corresponds to summing over background gauge fields, as follows:

$$Z_{\Sigma_g \times S^1}^{[\mathcal{T}/\Gamma^{(1)}]}(\theta) = \frac{1}{|\Gamma|} \sum_{B \in H^2(\Sigma_g, \Gamma^{(1)})} e^{i(\theta, B)} Z_{\Sigma_g \times S^1}^{[\mathcal{T}]}(B). \tag{105}$$

Here, we weighted the contributions with the $\theta$-angle for the dual $(-1)$-form symmetry, as discussed in (91). This gives us:

$$Z_{\Sigma_g \times S^1}^{[\mathcal{T}/\Gamma^{(1)}]}(\theta) = \frac{1}{|\Gamma|} \sum_{\gamma \in \Gamma^{(1)}} \bar{\vartheta}(\gamma) \langle \Pi^{\gamma} \rangle_{\Sigma_g}, \tag{106}$$

---

[7]On the other hand, we may constrain the set of possible solutions somewhat, as follows. The *exponent* of the finite abelian group $\Gamma \cong \bigoplus_i \mathbb{Z}_{N_i}$, denoted by $\exp(\Gamma)$, is the least common multiple of the orders of all elements of the group. We then have $\exp(\Gamma)\gamma = 0$ for any $\gamma \in \Gamma$. Since $\chi_{\rho}$ is a group homomorphism, this forces $\chi_{\rho}(\gamma)^{\exp(\Gamma)} = 1$ and thus $\chi_{\rho}$ is necessarily an $\exp(\Gamma)$-th root of unity.

where $\bar{\vartheta}$ denotes the complex conjugate of $\vartheta$. Plugging in (88) and using the orthogonality of the characters,[8] we find that:

$$Z^{[\mathcal{T}/\Gamma^{(1)}]}_{\Sigma_g \times S^1}(\theta) = \sum_{\hat{u} \in \mathcal{S}^{\chi=\vartheta}_{\text{BE}}} \mathcal{H}(\hat{u})^{g-1} = \text{Tr}_{\mathcal{H}^{\vartheta}_{S^1}}(\mathcal{H}^{g-1}). \tag{107}$$

That is, gauging the one-form symmetry 'undoes decomposition' by projecting us onto a given $\chi = \vartheta$ sector [69]. The process is completely reversible, since gauging the $\Gamma^{(-1)}$ symmetry simply corresponds to summing over all possible $\theta$-angles:

$$Z^{[\mathcal{T}]}_{\Sigma_g \times S^1}(B) = \sum_{\theta} e^{-i(\theta, B)} Z^{[\mathcal{T}/\Gamma^{(1)}]}_{\Sigma_g \times S^1}(\theta). \tag{108}$$

It is worth commenting on the overall normalisation of the sum in (105) and in (108), respectively. Here we sum over all insertions with the overall normalisation $1/|\Gamma|$ when gauging $\Gamma^{(1)}$, and therefore we have a unit normalisation in (108) when gauging $\Gamma^{(-1)}$. This prescription is such that the result (107) for the gauged theory has a standard Hilbert space interpretation – it is simply given as a trace over the Hilbert space $\mathcal{H}^{\chi=\vartheta}_{S^1}$ defined in (55), with the local operator $\mathcal{H}$ being unaffected by the 1-form gauging.[9]

Note also that, in the gauged theory, the ambiguity (53) in defining the gauge flux operator $\Pi^{\gamma}$ is equivalent to a redefinition of angle $\theta$ according to $\theta \to \theta + 2\pi\rho$ (that is, $\bar{\vartheta}(\gamma) \to \chi_{\rho}(\gamma)\bar{\vartheta}(\gamma)$), which is simply a relabelling of the distinct universes in the decomposition (55).

## 2.4 Gauging the 0-form symmetry $\Gamma^{(0)}$

The discrete symmetry $\Gamma^{(0)}$ is non-anomalous in the 3d $A$-model for $\widetilde{G}$. Thus it can be gauged by summing over all $\Gamma^{(0)}$ gauge fields on $\Sigma_g$:

$$Z^{[\mathcal{T}/\Gamma^{(0)}]}_{\Sigma_g \times S^1}(C^D) = \frac{1}{|\Gamma|^{2g-1}} \sum_{C \in H^1(\Sigma_g, \Gamma^{(0)})} e^{2\pi i(C^D, C)} Z^{[\mathcal{T}]}_{\Sigma_g \times S^1}(C). \tag{109}$$

Note the normalisation constant in front, which differs from the 'symmetric' normalisation used in [19]. Here, $C^D$ is a background gauge field for the dual 0-form symmetry, $\Gamma^{(0)}_D \cong \Gamma$, which non-trivially permutes the twisted sectors. (This is sometimes known as the 'quantum symmetry.') We also defined the pairing:

$$(C^D, C) = \int_{\Sigma_g} C^D \cup C. \tag{110}$$

In the gauged theory, the topological line operator for $\gamma_D \in H_1(\Sigma_g, \Gamma^{(0)}_D)$ can be written as:

$$\mathcal{U}^{\gamma_D}_D(\mathcal{C}) = e^{i \int_{\mathcal{C}} C}, \qquad [C^D_{\gamma_D}] = \text{PD}[\gamma_D], \tag{111}$$

where $C$ is the dynamical gauge field for $\Gamma^{(0)}$. Note that we can view $\mathcal{U}^{\gamma_D}_D$ as a character:

$$\mathcal{U}^{\gamma_D}_D \in \text{Hom}(H_1(\Sigma_g, \Gamma^{(0)}), U(1)), \qquad \mathcal{U}^{\gamma_D}_D(\gamma) \equiv e^{2\pi i(C^D_{\gamma_D}, C_{\gamma})}. \tag{112}$$

---

[8] Namely, for any $\Gamma$ a discrete abelian group, and $\vartheta, \chi \in \hat{\Gamma}$, we have that $\sum_{\gamma \in \Gamma} \bar{\vartheta}(\gamma)\chi(\gamma) = |\Gamma|\delta_{\vartheta,\chi}$.

[9] Any other consistent normalisation, such as the one used in [19], would differ from ours by a factor of the form $\alpha^{g-1}$, corresponding to a rescaling of the handle-gluing operator (which is equivalent to adding a supersymmetry-preserving counterterm $\propto \int_{\Sigma} R$ – see e.g. [70]).

The partition function (109) can thus be written as:

$$Z^{[\mathcal{T}/\Gamma^{(0)}]}_{\Sigma_g \times S^1}(\gamma_D) = \frac{1}{|\Gamma|^{2g-1}} \sum_{[\gamma] \in H_1(\Sigma_g, \Gamma^{(0)})} \mathcal{U}^{\gamma_D}_D(\gamma) \, Z^{[\mathcal{T}]}_{\Sigma_g \times S^1}(\gamma). \tag{113}$$

Conversely, we have:

$$Z^{[\mathcal{T}]}_{\Sigma_g \times S^1}(\gamma) = \frac{1}{|\Gamma|} \sum_{[\gamma_D] \in H_1(\Sigma_g, \Gamma^{(0)}_D)} e^{2\pi i (C_\gamma, C^D_{\gamma_D})} \, Z^{[\mathcal{T}/\Gamma^{(0)}]}_{\Sigma_g \times S^1}(\gamma_D), \tag{114}$$

wherein gauging the dual 0-form symmetry gives us back the theory we started with. The partition function (113) can be written as a trace over the gauged Hilbert space [19]. The Hilbert space of the 2d theory $\mathcal{T}/\Gamma^{(0)}$ takes the form:

$$\mathcal{H}^{[\mathcal{T}/\Gamma^{(0)}]}_{S^1} \cong \mathrm{Span}_{\mathbb{C}} \left\{ |\hat\omega; s_{\hat\omega}\rangle \,\middle|\, \hat\omega \in \mathcal{S}_{\mathrm{BE}}/\Gamma^{(0)} \,, \, s_{\hat\omega} = 1, \ldots, |\mathrm{Stab}(\hat\omega)| \right\}, \tag{115}$$

where $\hat\omega \equiv \mathrm{Orb}(\hat u)$ denote the distinct $\Gamma^{(0)}$-orbits of Bethe vacua of the $\widetilde{G}$ gauge theory, and $s_{\hat\omega}$ indexes the twisted sectors. If we first consider $C_D = 0$, for definiteness:

$$Z^{[\mathcal{T}/\Gamma^{(0)}]}_{\Sigma_g \times S^1} \equiv \frac{1}{|\Gamma|^{2g-1}} \sum_{[\gamma] \in H_1(\Sigma_g, \Gamma^{(0)})} \langle \mathcal{U}^\gamma \rangle_{\Sigma_g}, \tag{116}$$

we find that:

$$Z^{[\mathcal{T}/\Gamma^{(0)}]}_{\Sigma_g \times S^1} = \sum_{\hat\omega \in \mathcal{S}_{\mathrm{BE}}/\Gamma^{(0)}} |\mathrm{Stab}(\hat\omega)| \left( \frac{|\mathrm{Stab}(\hat\omega)|^2}{|\Gamma|^2} \mathcal{H}(\hat\omega) \right)^{g-1}, \tag{117}$$

where we used the notation:

$$\mathcal{H}(\hat\omega) \equiv \mathcal{H}(\hat u), \qquad \forall \hat u \in \hat\omega, \tag{118}$$

which is well defined since $\mathcal{H}(\hat u + \gamma) = \mathcal{H}(\hat u)$, $\forall \gamma \in \Gamma^{(0)}$. Indeed, starting with the expression for the partition function of the gauged theory (116), we have:

$$\begin{aligned}
Z^{[\mathcal{T}/\Gamma^{(0)}]}_{\Sigma_g \times S^1} &= \frac{1}{|\Gamma|^{2g-1}} \sum_{[\gamma] \in H_1(\Sigma_g, \Gamma^{(0)})} \sum_{\hat u \in \mathcal{S}^{(\gamma)}_{\mathrm{BE}}} \mathcal{H}(\hat u)^{g-1} \\
&= \frac{1}{|\Gamma|^{2g-1}} \sum_{\substack{(\hat u, \gamma) \in \mathcal{S}_{\mathrm{BE}} \times \Gamma^{2g} \\ \hat u \in \mathcal{S}^{(\gamma)}_{\mathrm{BE}}}} \mathcal{H}(\hat u)^{g-1} \\
&= \frac{1}{|\Gamma|^{2g-1}} \sum_{\hat u \in \mathcal{S}_{\mathrm{BE}}} \sum_{\gamma \in \mathrm{Stab}(\hat u)^{2g}} \mathcal{H}(\hat u)^{g-1} \\
&= \frac{1}{|\Gamma|^{2g-1}} \sum_{\hat\omega \in \mathcal{S}_{\mathrm{BE}}/\Gamma^{(0)}} \mathcal{H}(\hat\omega)^{g-1} |\hat\omega| \, |\mathrm{Stab}(\hat\omega)|^{2g},
\end{aligned} \tag{119}$$

where in the third equality we exchanged the order of the two sums. Moreover, in the last step we used (118) and $|\mathrm{Stab}(\hat\omega)| \equiv |\mathrm{Stab}(\hat u)|, \forall \hat u \in \hat\omega$. Then, using the stabiliser-orbit theorem (63), the final expression above simplifies to (117).

Note that we chose the overall normalisation in (109) such that the expression (117) can be interpreted as a trace over the Hilbert space (115) of the $\Gamma^{(0)}$-gauged theory, including all twisted-sector states. Note also that, for $g = 0$, this implies that we have:

$$Z^{[\mathcal{T}/\Gamma^{(0)}]}_{S^2 \times S^1} = |\Gamma| Z^{[\mathcal{T}]}_{S^2 \times S^1}. \tag{120}$$

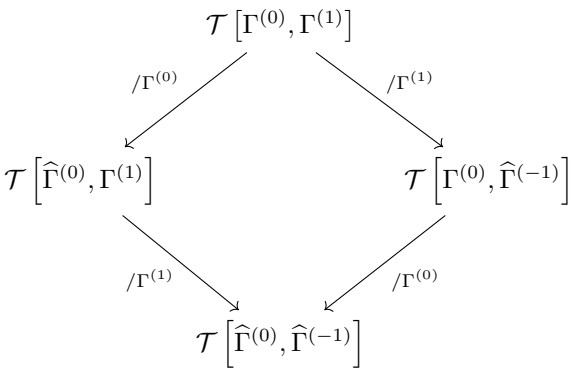

Figure 7: Gaugings of 0-form and 1-form symmetries of a 2d theory $\mathcal{T}$. In the absence of a 't Hooft anomaly, the diagram commutes. If $\Gamma^{(1)}_{3d}$ is anomalous, the bottom theory does not exist and the diagram truncates.

More generally, if we turn on a background gauge field for $\Gamma^{(0)}_D$ as in (113), a similar computation gives us:

$$Z^{[\mathcal{T}/\Gamma^{(0)}]}_{\Sigma_g \times S^1}(\gamma_D) = \sum_{\hat{\omega} \in \mathcal{S}_{\text{BE}}/\Gamma^{(0)}} |\text{Stab}(\hat{\omega})|^{-1} \left( \frac{\mathcal{H}(\hat{\omega})}{|\Gamma|^2} \right)^{g-1} \sum_{\gamma \in \text{Stab}(\hat{\omega})^{2g}} \mathcal{U}_D^{\gamma_D}(\gamma). \tag{121}$$

In the special case $\gamma_D = 0$ (so that $\mathcal{U}_D^{\gamma_D}(\gamma) = 1$), this reduces to (117).

## 2.5 Topologically twisted index for the gauge group $\widetilde{G}/\Gamma$

Finally, we can combine the above results to gauge the full $\Gamma^{(1)}_{3d}$ in the $A$-model description, assuming the symmetry is non-anomalous. We can then gauge the symmetries in whichever order, as indicated in figure 7. The general formula for the 3d theory with gauge group $G = \widetilde{G}/\Gamma$ on $\Sigma_g \times S^1$ is obtained by summing over all insertions of topological operators for $\Gamma^{(1)}_{3d}$:

$$Z^{\mathcal{T}/\Gamma^{(1)}_{3d}}(\theta, C^D) = \frac{1}{|\Gamma|^{2g}} \sum_{\delta \in \Gamma^{(1)}} \sum_{[\gamma] \in H_1(\Sigma_g, \Gamma^{(0)})} e^{i(\theta, B_\delta)} e^{2\pi i (C^D, C_\gamma)} \left\langle \Pi^\delta \mathcal{U}^\gamma \right\rangle_{\Sigma_g}. \tag{122}$$

Conceptually, it is simplest to consider gauging the symmetries one after the other. For instance, consider gauging $\Gamma^{(0)}$ first. The intermediate 2d theory on the left-hand corner of figure 7 still has a $\Gamma^{(1)}$ symmetry, and it therefore enjoys decomposition. The subsequent gauging of $\Gamma^{(1)}$ is then a projection on a particular universe. If we first gauge $\Gamma^{(1)}$ instead, we first project onto one universe and then consider the $\Gamma^{(0)}$ gauging inside that universe. Either way, we end up considering the Bethe vacua determined by the $\vartheta$-twisted Bethe equations (56), which we can suggestively rewrite as:

$$\mathcal{S}^\vartheta_{\text{BE}} \equiv \left\{ \hat{u} \in \mathfrak{t}/\Lambda^{\widetilde{G}}_{\text{mw}} \;\middle|\; \Pi^\gamma(\hat{u}) = \vartheta(\gamma), \forall \gamma \in \Lambda^{\widetilde{G}/\Gamma}_{\text{mw}} \quad \text{and} \quad w \cdot \hat{u} \neq \hat{u}, \forall w \in W_{\widetilde{G}} \right\}/W_{\widetilde{G}}. \tag{123}$$

Note that $\vartheta(\gamma) = 1$ for any $\gamma \in \Lambda^{\widetilde{G}}_{\text{mw}} \subset \Lambda^{\widetilde{G}/\Gamma}_{\text{mw}}$. We then find:

$$Z^{\mathcal{T}/\Gamma^{(1)}_{3d}}_{\Sigma_g \times S^1}(\theta, C^D_{\gamma_D}) = \sum_{\hat{\omega} \in \mathcal{S}^\vartheta_{\text{BE}}/\Gamma^{(0)}} |\text{Stab}(\hat{\omega})|^{-1} \left( \frac{\mathcal{H}(\hat{\omega})}{|\Gamma|^2} \right)^{g-1} \sum_{\gamma \in \text{Stab}(\hat{\omega})^{2g}} \mathcal{U}_D^{\gamma_D}(\gamma). \tag{124}$$

Finally, let us consider the special case $\theta = \gamma_D = 0$. The twisted index for the $G = \widetilde{G}/\Gamma$ gauge theory then takes the simple form:

$$Z^{\mathcal{T}/\Gamma_{3d}^{(1)}}_{\Sigma_g \times S^1} = \text{Tr}_{\mathcal{H}_{S^1}^{[\widetilde{G}/\Gamma]}} \left( \mathcal{H}_G^{g-1} \right), \tag{125}$$

where the trace is over the Hilbert space of the $A$-model for the $G = \widetilde{G}/\Gamma$ gauge theory:

$$\mathcal{H}_{S^1}^{[\widetilde{G}/\Gamma]} \cong \text{Span}_{\mathbb{C}} \left\{ |\hat{\omega}; s_{\hat{\omega}}\rangle \ \middle| \ \hat{\omega} \in \mathcal{S}_{\text{BE}}^{\vartheta=1}/\Gamma^{(0)}, \ s_{\hat{\omega}} = 1, \dots, |\text{Stab}(\hat{\omega})| \right\}. \tag{126}$$

Here, we have introduced the handle-gluing operator $\mathcal{H}_G$, which acts on the Bethe vacua of the $G$ gauge theory as:

$$\mathcal{H}_G |\hat{\omega}; s_{\hat{\omega}}\rangle = \frac{\mathcal{H}(\hat{\omega})}{|\hat{\omega}|^2} |\hat{\omega}; s_{\hat{\omega}}\rangle, \tag{127}$$

where $\mathcal{H} = \mathcal{H}_{\widetilde{G}}$ is the ordinary handle-gluing operator of the $\widetilde{G}$ gauge theory, and we used the notation (118).

This completes our general discussion of the topologically twisted index for 3d $\mathcal{N} = 2$ supersymmetric gauge theories with a general gauge group $G$ and a $U(1)_R$ symmetry. The same caveats that held for $\widetilde{G}$ apply here [8] – in particular, we implicitly assumed that the set of Bethe vacua is discrete. Another important caveat, which was not explicitly stated in previous literature, is that our approach in this section gives the correct result for the twisted indices if and only if the 3d topological lines that we insert to gauge $\Gamma_{3d}^{(1)}$ are all bosonic (that is, they have trivial braiding with the 'spin-structure' transparent line $\psi$). This ensures that, starting from a 3d $\widetilde{G}$ gauge theory where all the Bethe vacua are bosonic, the set of Bethe vacua in the $G = \widetilde{G}/\Gamma$ theory are also all bosonic. The treatment of fermionic Bethe vacua will be addressed in future work [21].

Finally, we note that it would be straightforward to generalise our discussion to 'non-Lagrangian' 3d $\mathcal{N} = 2$ field theories with one-form symmetries for which the $A$-model formalism is available. For the $T[\mathcal{M}_3]$ theories of the 3d/3d correspondence [71–73], the computation of $\Sigma = T^2$ was first discussed in [18].

## 3 The 3d $\mathcal{N} = 2$ $SU(N)_K$ Chern–Simons theory

In the rest of this paper, we mainly set $\widetilde{G} = SU(N)$. This is a simply-connected gauge group with centre $Z(SU(N)) = \mathbb{Z}_N$, and we can thus study all the possible quotients $SU(N)/\mathbb{Z}_r$, where $r$ is a divisor of $N$. This gives us a rich structure of gauge theories obtained by discrete gauging of subgroups, which depends crucially on the arithmetic properties of the integer $N$ – see *e.g.* [15, 16, 74] for closely related discussions.

In this section, we specifically study the $\mathcal{N} = 2$ supersymmetric Chern–Simons theory $SU(N)_K$ (with supersymmetric CS level $K \geq N$). Upon integrating out the gauginos, this is equivalent to the bosonic CS theory $SU(N)_k$ with $k = K - N \geq 0$. Since many explicit results are available for this 3d TQFT, this serves as a useful testing ground for the general formalism of the previous section. We also obtain some seemingly new results.

### 3.1 The $A$-model for the $SU(N)_K$ theory and its higher-form symmetries

The $SU(N)_K$ CS theory has a one-form symmetry $\Gamma_{3d}^{(1)} = \mathbb{Z}_N$, which descends to the higher-form symmetry $\mathbb{Z}_N^{(0)} \oplus \mathbb{Z}_N^{(1)}$ in the 2d description. In this subsection, we give an explicit parametrisation of the Bethe vacua for this theory and we study how the higher-form symmetries act on them.

**Effective twisted superpotential and dilaton**   Although $SU(N) = N$ has rank $N - 1$, it is convenient to use a slightly redundant description with $N$ variables $u_a$, $a = 1, \cdots, N$, together with the tracelessness condition:

$$\sum_{a=1}^{N} u_a = 0 \,. \tag{128}$$

The effective twisted superpotential is given by:

$$\mathcal{W}(u) = \frac{K}{2} \sum_{a=1}^{N} u_a^2 + u_0 \sum_{a=1}^{N} u_a \,, \tag{129}$$

where $u_0$ is a Lagrange multiplier for the constraint (128). The effective dilaton potential is given by:

$$e^{2\pi i \Omega(u)} = - \prod_{\substack{a,b=1 \\ a \neq b}}^{N} \left( 1 - \frac{x_a}{x_b} \right)^{-1} \,. \tag{130}$$

Here, for later convenience, we turned on an effective CS level $K_{RR} \in 2\mathbb{Z}+1$ for the R-symmetry $U(1)_R$. We also use the notation $x_a \equiv e^{2\pi i u_a}$ and $q \equiv e^{2\pi i u_0}$.

**Parametrising the Bethe vacua**   The Bethe equations that follow from the effective twisted superpotential (129) are:

$$\Pi_a(u) \equiv q x_a^K = 1 \,, \quad a = 1, \ldots, N \,, \qquad \Pi_0(u) \equiv \prod_{a=1}^{N} x_a = 1 \,. \tag{131}$$

By taking the product of the $N$ flux operators $\Pi_a$, we see that $q$ must be an $N$-th root of unity:

$$\hat{q} = e^{2\pi i \frac{\ell}{N}} \,, \qquad \ell \in \mathbb{Z}_N \,. \tag{132}$$

Substituting $q = \hat{q}$ into $\Pi_a$, one finds the Bethe vacuum solutions:

$$\hat{u}_a = \left( \frac{l_a}{K} - \frac{\ell}{KN} \right) \bmod 1 \,, \qquad a = 1, \ldots, N \,, \tag{133}$$

for $l_a \in \mathbb{Z}_K$, and with the tracelessness constraint:

$$\sum_{a=1}^{N} l_a - \ell \in K\mathbb{Z} \,. \tag{134}$$

We index the Bethe vacua by the $(N + 1)$-tuples $\underline{l}$ defined as:

$$\underline{l} = (l_1, \ldots, l_N; \ell) \,, \qquad \hat{u}_{\underline{l}} \equiv \hat{u}_a e^a \,. \tag{135}$$

The Weyl symmetry $W_{\widetilde{G}} = S_N$ permutes the $u_a$ variables, and we can therefore choose the ordered $N$-tuples $\{l_1, \ldots, l_N\} \subset \{0, 1 \ldots, K - 1\}$ by fixing a gauge. That is, we must have $l_a > l_b$ if $a > b$. Before imposing the condition (134), there are thus $N\binom{K}{N}$ possibilities for $\underline{l}$. Evaluating the trace $\sum_{a=1}^{N} l_a - \ell$ (mod $K$) partitions this set of tuples into $K$ sets of equal size. Selecting the traceless tuples thus gives us the total number of Bethe vacua, which reproduces the well-known Witten index [47, 75]:

$$\mathbf{I}_{\mathrm{W}}[SU(N)_K] \equiv Z_{T^3}[SU(N)_K] = \frac{N}{K} \binom{K}{N} = \binom{K - 1}{N - 1} \,. \tag{136}$$

In summary, the Bethe vacua $\hat{u}_{\underline{l}} \in \mathcal{S}_{\mathrm{BE}}$ are in one-to-one correspondence with the elements $\underline{l}$ in the indexing set:

$$\mathcal{J}_{N,K} \equiv \left\{ (l_1, \ldots, l_N; \ell) \in \mathbb{Z}_K^N \oplus \mathbb{Z}_N, \, \Big| \, 0 \le l_1 < \ldots < l_N \le K, \, \sum_{a=1}^N l_a - \ell \in K\mathbb{Z} \right\}, \qquad (137)$$

with $\hat{u}_{\underline{l}}$ given in (133). An equivalent determination of this set is obtained by first looking at the $\binom{K}{N}$ ordered sets $(l_1, \cdots, l_N) \in \mathbb{Z}_K^N$, defining $\ell \equiv \sum_{a=1}^N l_a \bmod K$, and retaining the resulting sets $\underline{l}$ if and only if $\ell < N$.[10]

**Handle-gluing operator and the twisted index**  The twisted index can be computed as in (86). Using the parametrisation (137) for the Bethe vacua, the handle-gluing operator evaluated on-shell gives us:

$$\mathcal{H}(\hat{u}_{\underline{l}}) = N \left( \frac{K}{2^N} \right)^{N-1} \prod_{1 \le a < b \le N} \sin^{-2} \left( \frac{\pi(l_a - l_b)}{K} \right), \qquad (138)$$

and the topologically twisted index of the $SU(N)_K$ theory is then given by:

$$Z_{\Sigma_g \times S^1}[SU(N)_K] = N^{g-1} \left( \frac{K}{2^N} \right)^{(g-1)(N-1)} \sum_{\underline{l} \in \mathcal{J}_{N,K}} \prod_{1 \le a < b \le N} \left( \sin \frac{\pi(l_a - l_b)}{K} \right)^{2-2g}, \qquad (139)$$

where the sum is over the elements of the set (137). This is the well-known Verlinde formula for the $SU(N)_k$ bosonic CS theory with $k = K - N$ [34, 37].

Let us consider a few explicit examples. For $K = N$ and $K = N + 1$, the Verlinde formula simplifies to:

$$Z_{\Sigma_g \times S^1}[SU(N)_N] = 1, \qquad Z_{\Sigma_g \times S^1}[SU(N)_{N+1}] = N^g, \qquad (140)$$

This is can be easily understood from the level/rank duality $SU(N)_K \leftrightarrow U(K-N)_{-K,-N}$ (see *e.g.* [76, 77]). With some more effort, a few more complicated examples can be evaluated explicitly as a function of $g$. For instance, we find:

$$Z_{\Sigma_g \times S^1}[SU(3)_6] = \frac{1}{12} \left( 3 \cdot 4^g + 8 \cdot 9^g + 36^g \right), \qquad (141)$$

$$Z_{\Sigma_g \times S^1}[SU(4)_6] = 6^{g-1}(4^g + 2) + 2^{3g-1}. \qquad (142)$$

Note that these are integers for any $g$, and that they give 1 at $g = 0$, as expected for any 3d TQFT. See [38, 78, 79] for comments on closed-form summations of trigonometric sums of these kinds.

### 3.1.1 Two-dimensional 1-form symmetry, gauging and decomposition

In the $A$-model description, we have a two-dimensional one-form symmetry whose topological operator is the flux operator that inserts some magnetic flux $\gamma$ along $\Sigma$, corresponding to a $PSU(N) \equiv SU(N)/\mathbb{Z}_N$ bundle that cannot be lifted to an $SU(N)$ bundle:

$$\gamma \in \Lambda_{\mathrm{mw}}^{PSU(N)} / \Lambda_{\mathrm{mw}}^{SU(N)} \cong \mathbb{Z}_N. \qquad (143)$$

---

[10]As an example, consider the $SU(3)_6$ $\mathcal{N} = 2$ theory, which has 10 vacua. Using the presentation (137), we can list them as:

$$(0,1,5;0), \quad (1,2,3;0), \quad (3,4,5;0), \quad (0,2,5;1), \quad (1,2,4;1),$$
$$(0,3,4;1), \quad (0,3,5;2), \quad (1,2,5;2), \quad (1,3,4;2), \quad (0,2,4;0).$$

In our parameterisation of $u \in \mathfrak{t} \subset \mathfrak{su}(N)$ with $u = u_a e^a$ as above, the generator $\gamma_0$ of $\mathbb{Z}_N$ in (143) can be chosen to be:[11]

$$\gamma_0 = \gamma_{0,a} e^a, \qquad \gamma_{0,a} = -\frac{1}{N} + \delta_{a,1}, \qquad a = 1, \ldots, N. \tag{144}$$

We then write the elements (143) as $\gamma = n\gamma_0$ for $n \in \mathbb{Z}_N$ an integer modulo $N$. The corresponding $\mathbb{Z}_N^{(1)}$ operators are $\Pi^\gamma \equiv \mathcal{U}^\gamma(S_A^1)$ with:

$$\Pi^{\gamma_0}(u) \equiv \prod_{a=1}^{N} \Pi_a^{\gamma_{0,a}}(u). \tag{145}$$

On-shell, this evaluates to:

$$\Pi^{\gamma_0}(u_{\underline{l}}) = (-1)^{N-1} \hat{q}^{-1} = (-1)^{N-1} e^{-\frac{2\pi i \ell}{N}}, \tag{146}$$

where we have fixed the overall phase by hand for future convenience, as we will discuss momentarily. Hence $\mathbb{Z}_N^{(1)}$ acts on the Bethe vacua as:

$$\Pi^{m\gamma_0}|\hat{u}_{\underline{l}}\rangle = (-1)^{m(N-1)} e^{-2\pi i \frac{m\ell}{N}} |\hat{u}_{\underline{l}}\rangle, \qquad m = 0, 1, \ldots, N-1, \tag{147}$$

where recall that $\ell = \sum_a l_a \bmod K$. We can then directly evaluate the expectation value of this flux operator on $\Sigma_g$:

$$\langle \Pi^{m\gamma_0} \rangle_{\Sigma_g} = N^{g-1} \left(\frac{K}{2^N}\right)^{(g-1)(N-1)} \sum_{\underline{l} \in \mathcal{J}_{N,K}} (-1)^{m(N-1)} e^{-2\pi i \frac{m\ell}{N}} \prod_{1 \le a < b \le N} \left(\sin \frac{\pi(l_a - l_b)}{K}\right)^{2-2g}. \tag{148}$$

For $g = 1$, we find the following explicit evaluation formula:

$$\langle \Pi^{m\gamma_0} \rangle_{T^2} = \sum_{\underline{l} \in \mathcal{J}_{N,K}} (-1)^{m(N-1)} e^{-2\pi i \frac{m\ell}{N}}$$

$$= \delta_{K \bmod d, 0} \binom{\frac{K}{d} - 1}{\frac{N}{d} - 1}, \qquad \text{with} \quad d \equiv \frac{N}{\gcd(m, N)}. \tag{149}$$

This is an 'experimental' result that follows from the physical expectation of modularity on $T^3$, as we will explain below.[12] In particular, (149) is always a non-negative integer.

**Gauging of $\mathbb{Z}_N^{(1)}$** To gauge $\mathbb{Z}_N^{(1)}$ in the $A$-model description, we simply sum over the insertions (148) for all possible values of $m$, as in (105):

$$Z_{\Sigma_g \times S^1} \left[SU(N)_K / \mathbb{Z}_N^{(1)}\right](\theta_s) = \frac{1}{N} \sum_{m=0}^{N-1} e^{i(\theta_s, m\gamma_0)} \langle \Pi^{m\gamma_0} \rangle_{\Sigma_g}. \tag{150}$$

The discrete $\theta$-angle for the dual $(-1)$-form symmetry takes values:

$$\theta_s \equiv 2\pi \frac{s}{N}, \qquad s = 0, \ldots, N-1, \tag{151}$$

---

[11]The shift by the integer $\delta_{a,1}$ is such that $\sum_{a=1}^{N} \gamma_{0,a} = 0$. Note also that $\mathfrak{m} \in \Lambda_{\text{mw}}^{SU(N)}$ is similarly defined as $\mathfrak{m} = \mathfrak{m}_a e^a$ with $\mathfrak{m}_a \in \mathbb{Z}$ and $\sum_{a=1}^{N} \mathfrak{m}_a = 0$.

[12]We checked this identity on a computer for a large number of values of $N$ and $K$. We leave finding a mathematical proof as a challenge for the interested reader.

so that $(\theta_s, m\gamma_0) = 2\pi \frac{sm}{N}$ in (150). Let us define the geometric series:

$$\Delta_\ell^N(s) \equiv \frac{1}{N}\sum_{m=0}^{N-1}(-1)^{m(N-1)}e^{2\pi i m\frac{s-\ell}{N}} = \begin{cases} \delta_{s-\ell+\frac{N}{2}\bmod N,0}, & \text{if } N \text{ is even,} \\ \delta_{s-\ell \bmod N,0}, & \text{if } N \text{ is odd.} \end{cases} \tag{152}$$

We then have:

$$Z_{\Sigma_g \times S^1}\left[SU(N)_K/\mathbb{Z}_N^{(1)}\right](\theta_s) = N^{g-1}\left(\frac{K}{2^N}\right)^{(g-1)(N-1)}$$
$$\times \sum_{\underline{l}\in\mathcal{J}_{N,K}}\Delta_\ell^N(s)\prod_{1\le a<b\le N}\left(\sin\frac{\pi(l_a - l_b)}{K}\right)^{2-2g}. \tag{153}$$

This result is in agreement with (107) and with the decomposition $\mathcal{S}_{\text{BE}} \cong \bigoplus_s \mathcal{S}_{\text{BE}}^{\vartheta_s}$ induced by the 1-form symmetry on the Bethe vacua of the $SU(N)_K$ theory, with:

$$\hat{u}_{\underline{l}} \in \mathcal{S}_{\text{BE}}^{\vartheta_s} \longleftrightarrow \underline{l} \in \mathcal{J}_{N,K}^s \equiv \left\{\underline{l}\in\mathcal{J}_{N,K}\,\middle|\,(-1)^{N-1}e^{2\pi i\frac{\ell}{N}} = e^{2\pi i\frac{s}{N}}\right\}, \tag{154}$$

and thus (153) is simply a sum over the Bethe vacua $\hat{u} \in \mathcal{S}_{\text{BE}}^{\vartheta_s}$ indexed by $\mathcal{J}_{N,K}^s$:

$$Z_{\Sigma_g \times S^1}\left[SU(N)_K/\mathbb{Z}_N^{(1)}\right](\theta_s) = \sum_{\hat{u}\in\mathcal{S}_{\text{BE}}^{\vartheta_s}}\mathcal{H}(\hat{u})^{g-1}. \tag{155}$$

We thus see that the 2d universes indexed by $\theta_s$ are equivalently indexed by $\ell$, as:

$$\ell = \begin{cases} s + \frac{N}{2}\bmod N, & \text{if } N \text{ is even,} \\ s, & \text{if } N \text{ is odd.} \end{cases} \tag{156}$$

For $N=2$ and $s=0$, (153) of course reproduces (15). In the special case $g=1$, we have:

$$Z_{T^3}\left[SU(N)_K/\mathbb{Z}_N^{(1)}\right](\theta_s) = \left|\mathcal{J}_{N,K}^s\right|. \tag{157}$$

Interestingly, 'experimentally' we find that, if $N$ is odd and prime, then $|\mathcal{J}_{N,K}^s|$ is given by:

$$\left|\mathcal{J}_{N,K}^s\right| = \frac{1}{N}\binom{K-1}{N-1} + \delta_{K\bmod N,0}\left(\delta_{s,0} - \frac{1}{N}\right), \tag{158}$$

which is always an integer. We will derive this below using 3d modularity. Of course, we see that $\sum_{s=0}^{N-1}|\mathcal{J}_{N,K}^s| = |\mathcal{J}_{N,K}|$.

**Gauging of a subgroup $\mathbb{Z}_r^{(1)}$**    For later purpose, we can also consider the gauging of a subgroup $\mathbb{Z}_r^{(1)} \subseteq \mathbb{Z}_N^{(1)}$, for $r$ any divisor of $N$ and $K$:

$$Z_{\Sigma_g \times S^1}\left[SU(N)_K/\mathbb{Z}_r^{(1)}\right](\theta_s^{(\mathbb{Z}_r)}) = \frac{1}{r}\sum_{m=0}^{r-1}e^{2\pi i\frac{ms}{r}}\left\langle\Pi^{m\frac{N}{r}\gamma_0}\right\rangle_{\Sigma_g}, \tag{159}$$

where $\theta_s^{(\mathbb{Z}_r)} = 2\pi\frac{s}{r}$ with $s\in\mathbb{Z}_r$. Note the normalisation of the sum in (159). The sum over $m$ generalises (152) to:

$$\Delta_\ell^{N,r}(s) \equiv \frac{1}{r}\sum_{m=0}^{r-1}(-1)^{m\frac{N}{r}(N-1)}e^{2\pi i m\frac{s-\ell}{s}}$$
$$= \begin{cases} \delta_{s-\ell-\frac{N(N-1)}{2}\bmod r,0}, & \text{if } N \text{ is even and } \frac{N}{r} \text{ is odd,} \\ \delta_{s-\ell\bmod r,0}, & \text{otherwise.} \end{cases} \tag{160}$$

Hence, the sum (159) over the $\mathbb{Z}_r^{(1)}$ topological operators projects us onto universes indexed by $\theta_s^{(\mathbb{Z}_r)}$, and (160) tells us that we can equivalently parameterise these universes by $\ell \bmod r$, with the exact relation being:

$$
s = \begin{cases} \ell + \frac{N(N-1)}{2} \bmod r\,, & \text{if } N \text{ is even and } \frac{N}{r} \text{ is odd,} \\ \ell \bmod r\,, & \text{otherwise.} \end{cases} \tag{161}
$$

We then have:

$$
Z_{\Sigma_g \times S^1}\big[SU(N)_K / \mathbb{Z}_r^{(1)}\big](\theta_s^{(\mathbb{Z}_r)}) = \sum_{\hat{u} \in \mathcal{S}_{\mathrm{BE}}^{\vartheta_s^{(\mathbb{Z}_r)}}} \mathcal{H}(\hat{u})^{g-1}\,, \tag{162}
$$

where we defined:

$$
\hat{u}_{\underline{l}} \in \mathcal{S}_{\mathrm{BE}}^{\vartheta_s^{(\mathbb{Z}_r)}} \quad \longleftrightarrow \quad \underline{l} \in \mathcal{J}_{N,K}^{s,r} \equiv \left\{ \underline{l} \in \mathcal{J}_{N,K} \,\Big|\, (-1)^{\frac{N}{r}(N-1)} e^{2\pi i \frac{\ell}{r}} = e^{2\pi i \frac{s}{r}} \right\}\,. \tag{163}
$$

This obviously generalises (155). Note that, for a divisor $r < N$, the 'universes' $\mathcal{S}_{\mathrm{BE}}^{\vartheta_s^{(\mathbb{Z}_r)}}$ permuted by the dual $(-1)$-form symmetry $\mathbb{Z}_r^{(-1)}$ are generally larger than the universes $\mathcal{S}_{\mathrm{BE}}^{\vartheta_s^{(\mathbb{Z}_N)}} = \mathcal{S}_{\mathrm{BE}}^{\vartheta_s}$ permuted by $\mathbb{Z}_N^{(-1)}$.

### 3.1.2 Two-dimensional 0-form symmetry and orbits of Bethe vacua

Let us now consider the action of the 0-form symmetry $\mathbb{Z}_N^{(0)}$ on the Bethe vacua. On the cylinder, the $\mathcal{U}^\gamma$ operator acts as:

$$
\mathcal{U}^\gamma |\hat{u}_{\underline{l}}\rangle = |\hat{u}_{\underline{l}} + \gamma\rangle\,. \tag{164}
$$

Given the choice of generator $\gamma_0$ in (144), we see that, for $n \in \mathbb{Z}_N$:

$$
\hat{u}_{\underline{l},a} \to \hat{u}_{\underline{l}',a} = (\hat{u}_{\underline{l}} + n\gamma_0)_a = \frac{1}{NK}(N\,l_a - \ell - nK) \bmod 1\,, \tag{165}
$$

where $\hat{u}_{\underline{l}} = (\hat{u}_{\underline{l},a})$ is defined up to permutations of the $u_a$'s. The corresponding action $\underline{l} \to \underline{l}'$ on the labels $\underline{l} = (l_a; \ell)$ reads:

$$
\ell \to \ell' = (\ell + nK) \bmod N\,, \qquad l_a \to l_a' = l_a - \frac{nK + \ell - \ell'}{N} \bmod K\,. \tag{166}
$$

Note that $\ell$ is preserved by the action of $\mathbb{Z}_N^{(0)}$ if and only if $K \in N\mathbb{Z}$. Since $\ell$ indexes the decomposition of the Hilbert space induced by $\mathbb{Z}_N^{(1)}$, as in (156), this means that the action of $\mathbb{Z}_N^{(0)}$ generally does not preserve decomposition.[13] This is a manifestation of the mixed 't Hooft anomaly between $\mathbb{Z}_N^{(1)}$ and $\mathbb{Z}_N^{(0)}$, as we will explain momentarily.

---

[13] As an aside, note that $\mathbb{Z}_N^{(0)}$ acts by permutation on the Bethe vacua, hence there exists a homomorphism:

$$
\varphi \,:\, \mathbb{Z}_N^{(0)} \longrightarrow S_{\binom{K-1}{N-1}} \,:\, \gamma \mapsto \varphi_\gamma\,.
$$

For the generator $\gamma_0$, the permutation $\varphi_{\gamma_0}$ is implicitly given by the action (165). It can sometimes be instructive to study the permutation $\varphi_{\gamma_0}$ in cycle notation. As an example, consider the permutation of the Bethe vacua for $N = 3$ and $K = 6$,

$$
SU(3)_6 : \quad \varphi_{\gamma_0} = (1\,2\,3)(4\,5\,6)(7\,8\,9)(10)\,,
$$

where the vacua are as in footnote 10. There are three cycles of length 3, while there is a unique fixed point in this case – this is the vacuum $\underline{l} = (0, 2, 4; 0)$.

Let $\mathcal{C}$ denote a basis element of $H_1(\Sigma_g, \mathbb{Z}) \cong \mathbb{Z}^{2g}$, and let us wrap a single topological line of charge $\gamma = n\gamma_0$ along $\mathcal{C}$. Its expectation value is given by (98), namely:

$$\langle \mathcal{U}^\gamma(\mathcal{C}) \rangle_{\Sigma_g} = \sum_{\hat{u} \in \mathcal{S}_{\mathrm{BE}}^{(\gamma)}} \mathcal{H}(\hat{u})^{g-1} \,, \tag{167}$$

where $\mathcal{S}_{\mathrm{BE}}^{(\gamma)}$ denotes the subset of the Bethe vacua that are fixed under the action of $\gamma \in \mathbb{Z}_N^{(0)}$. In particular, for $g = 1$ this counts the number of fixed points:

$$\langle \mathcal{U}^\gamma(\mathcal{C}) \rangle_{T^2} = \left| \mathcal{S}_{\mathrm{BE}}^{(\gamma)} \right| \,. \tag{168}$$

By modularity on $T^3$, we expect that:

$$\langle \Pi^\gamma \rangle_{T^2} = \langle \mathcal{U}^\gamma(\mathcal{C}) \rangle_{T^2} \,, \tag{169}$$

where $\langle \Pi^\gamma \rangle_{T^2}$ is given in (149). Indeed, this is simply the constraint (104) described in section 2.2.

We will now calculate (168) explicitly, for any $\gamma = n\gamma_0$ and any value of $N$ and $K$. As previously alluded to, the flavour of the discussion of fixed points and orbits depends crucially on the subgroup structure of the zero-form symmetry $\mathbb{Z}_N^{(0)}$. The simplest case is where $N$ is a prime number. Then, $\mathbb{Z}_N$ has the special property that every element except the trivial element generates the full group. As a consequence, $\mathbb{Z}_N$ has $N-1$ generators, and the result of the fixed point counting will be independent on the choice of non-zero element $\gamma \in \mathbb{Z}_N^{(0)}$. This is in strong contrast to the case where $N$ has divisors, in which case elements $\gamma \in \mathbb{Z}_N^{(0)}$ can lie in proper subgroups of $\mathbb{Z}_N$, which are labelled by the divisors of $N$.

**Fixed points for $N$ a prime integer**   Let us first consider the case where $N$ is a prime number. Since there are no nontrivial subgroups of $\mathbb{Z}_N$, there are only two different cases depending on the CS level $K$. If $K$ is a multiple of $N$, the theory is non-anomalous, while if $K$ is not a multiple of $N$ then the theory is maximally anomalous. In section 2.1.2, we described how the mixed anomaly constrains the orbit structure. In the maximally anomalous case, all orbits are of 'maximal' length $N$. As a consequence, there cannot be any fixed points. If $K$ is a multiple of $N$ on the other hand, the orbit lengths are all divisors of $N$ – in our case where $N$ is prime, these are just 1 and $N$ itself.

We can easily find all fixed points. Assume that the generator $\gamma_0$ leaves invariant a particular Bethe vacuum, $\hat{u} = \hat{u} + \gamma$. By Weyl transformations of the associated tuples $(l_1, \ldots, l_N; \ell)$, (165) implies that the values $(l_1, \ldots, l_N)$ can at most be permuted, while $\ell$ has to be invariant. From (166) it is clear that $\ell$ is preserved if and only if $K \in N\mathbb{Z}$. Thus, this is a necessary condition to have a fixed point. Moreover, the fixed points satisfy $(l_a) \sim (l_a + \kappa)$, with $\kappa \equiv \frac{K}{N} \in \mathbb{Z}$. The equivalence above is up to permutation. Any solution must consequently take the form:

$$(l_a; \ell) = \left( s, s+\kappa, s+2\kappa, \ldots, s+(N-1)\kappa; Ns + \kappa \frac{N(N-1)}{2} \bmod \kappa N \right), \tag{170}$$

for an integer $s$ such that $0 \le s < \kappa$, and with the constraint that $\ell < N$. For $N$ odd, we see that there is a unique solution with $s = 0$, since the constraint on $\ell$ reads $\ell = Ns < N$, and thus $\ell = 0$. In appendix D.1, we prove the existence and uniqueness of the fixed point under the generator $\gamma_0 \in \mathbb{Z}_N$ for all integers $N$, with $K$ a multiple of $N$.

Returning to the case $N \ge 3$ prime, every nontrivial element $\gamma \ne 0$ is a generator, and thus we have shown:

$$\langle \mathcal{U}^\gamma(\mathcal{C}) \rangle_{T^2} = \delta_{K \bmod N, 0} \,, \tag{171}$$

for all $\gamma \in \mathbb{Z}_N \setminus \{0\}$. For the identity, we trivially have $\mathcal{U}^0(\mathcal{C}) = \mathbb{1}$ and thus $\langle \mathcal{U}^0(\mathcal{C}) \rangle_{T^2} = \mathbf{I}_W[SU(N)_K]$.

In order to determine the expectation value $\langle \mathcal{U}^\gamma(\mathcal{C}) \rangle_{\Sigma_g}$ for arbitrary genus, we need to evaluate the handle-gluing operator (138) at the fixed point. This is done by combining (167) with (D.5), such that:[14]

$$\mathcal{H}(\hat{u}_{\text{fixed}}) = N \left( \frac{K}{2^N} \right)^{N-1} \prod_{1 \le a < b \le N} \sin^{-2}\left( \frac{\pi(a-b)}{N} \right) = \left( \frac{K}{N} \right)^{(N-1)}, \tag{172}$$

which is an integer by the assumption that $N|K$. Since $\hat{u}_{\text{fixed}}$ is fixed under the generator $\gamma_0$ of $\mathbb{Z}_N$, it is also fixed under the action of $\mathcal{U}^{n\gamma_0}$ for $n = 1, 2, \dots, N-1$. Therefore, we have

$$\langle \mathcal{U}^{n\gamma_0}(\mathcal{C}) \rangle_{\Sigma_g} = \mathcal{H}(\hat{u}_{\text{fixed}})^{g-1} = \left( \frac{K}{N} \right)^{(g-1)(N-1)}, \tag{173}$$

for all $n = 1, \dots, N-1$, while for $n = 0$ it is simply $\langle 1 \rangle_{\Sigma_g}$.

**Fixed points for general $N$**   For each divisor $d$ of $N$, we have a cyclic subgroup $\mathbb{Z}_d$ of $\mathbb{Z}_N$. If the $SU(N)_K$ theory is non-anomalous, we can discretely gauge the full centre $\mathbb{Z}_N$, or indeed any of its subgroups $\mathbb{Z}_d$. If the $SU(N)_K$ theory is anomalous, there can still be a non-anomalous subgroup $\mathbb{Z}_d$ that we can discretely gauge – we will spell it out more explicitly in section 3.1.3 below. Especially including background fields for the higher form symmetries, this picture has been shown in other contexts to lead to a rich web of theories related by gaugings of discrete symmetries [15, 80].

It is easy to see that the order of the group generated by $\gamma = n\gamma_0$ is the smallest integer $d$ such that $N|dn$, which is given by $d = \frac{N}{\gcd(n,N)}$. Therefore:

$$\langle n\gamma_0 \rangle \cong \mathbb{Z}_{d(n)}, \qquad d(n) \equiv \frac{N}{\gcd(n,N)}, \tag{174}$$

which gives an explicit relation between the integers $n$ and divisors $d$ of $N$. See figure 8 for an explicit example.

Before studying the orbits in general, let us first count the number of fixed points under the subgroup $\langle n\gamma_0 \rangle$ – that is, the orbits of length 1. Clearly, this number can only depend on the divisor $d(n)$ (174). For $N$ prime, the only divisors are $d = 1$ and $d = N$, which gives the previous result (171). When $N$ has a nontrivial divisor $d \notin \{1, N\}$, then there can be multiple fixed points under a $\mathbb{Z}_d$ subgroup, as long as the CS level $K$ is a multiple of $d$ as well. In appendix D.2, we prove that the number of fixed points under the action of a $\mathbb{Z}_d$ subgroup is given by the simple formula:[15]

$$\langle \mathcal{U}^{n\gamma_0}(\mathcal{C}) \rangle_{T^2} = \binom{\frac{K}{d(n)} - 1}{\frac{N}{d(n)} - 1} \delta_{K \bmod d(n), 0}. \tag{175}$$

---

[14]In the second equation, we use the identity

$$\prod_{1 \le a < b \le N} \sin^2\left( \frac{\pi(a-b)}{N} \right) = (2^{1-N} N)^N,$$

which in fact holds for any $N$. To prove this, note that:

$$\prod_{1 \le a < b \le N} \sin^2\left( \frac{\pi(a-b)}{N} \right) = \prod_{d=1}^{N-1} \sin\left( \frac{\pi d}{N} \right)^{2(N-d)}.$$

Using $\sin(\pi - \theta) = \sin(\theta)$, we can rearrange the factors, such that it gives $N$ products of $\prod_{d=1}^{N-1} \sin\left( \frac{\pi d}{N} \right) = 2^{1-N} N$, which is a standard identity.

[15]This number is simply the Witten index (136) of the $SU(\frac{N}{d})_{\frac{K}{d}}$ theory, but this appears to be a coincidence, *i.e.* there does not seem to be any deeper meaning to this fact.

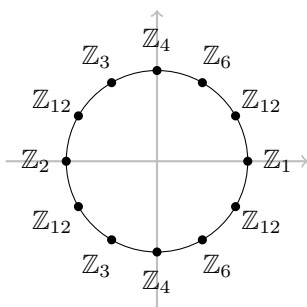

Figure 8: The full list of subgroups of the cyclic group $\mathbb{Z}_{12}$ is $\mathbb{Z}_1$, $\mathbb{Z}_2$, $\mathbb{Z}_3$, $\mathbb{Z}_4$, $\mathbb{Z}_6$ and $\mathbb{Z}_{12}$. They are generated by elements $n \in \mathbb{Z}_{12}$ as visualised on this clock – for instance, $n = 3$ generates $\mathbb{Z}_4 \subset \mathbb{Z}_{12}$.

Inserting the explicit $n$-dependence from (174), the number of fixed points (175) reads:

$$\langle \mathcal{U}^{n\gamma_0}(\mathcal{C}) \rangle_{T^2} = \binom{\frac{K}{N}\gcd(n,N)-1}{\gcd(n,N)-1} \delta_{K \bmod \frac{N}{\gcd(n,N)}, 0} \,. \tag{176}$$

Note that when $n = 1$, then $\frac{N}{\gcd(n,N)} = N$, such that the expression (176) is zero unless $N|K$, for which the binomial coefficient evaluates to 1, giving back (171) which holds for any value of $N$ if $\gamma = \gamma_0$. Moreover, if $N$ is prime and $n = 1, \ldots, N-1$ is arbitrary, then $\frac{N}{\gcd(n,N)} = N$, while $\gcd(n,N) = 1$ such that we arrive at (171) again, in agreement with the fact that $n\gamma_0$ is a generator of $\mathbb{Z}_N$. Finally, (176) is valid as well for $n = 0$, for which we get the trivial subgroup $\langle 0\gamma_0 \rangle \cong \mathbb{Z}_1$. All Bethe vacua are fixed under the trivial subgroup, and we get back the $SU(N)_K$ Witten index (136).

The result (175) now allows us to perform an exact test of the 3d modularity expectation (169). Indeed, the insertions of $\mathcal{U}^\gamma(\mathcal{C})$ (175) and $\Pi^\gamma$ (149) on $T^2$ agree precisely for all $\gamma$, as expected. This is due to the convention of the gauge flux operator (146), where the overall sign $(-1)^{N-1}$ has been inserted to eliminate any relative minus signs. Thus, assuming 3d modularity, we have now established (149) as well.

**Orbit structure** As outlined in section 2.2, the orbit structure under $\mathbb{Z}_N$ is constrained by the mixed 't Hooft anomaly $\mathcal{A}$. In particular, any anomaly-free subgroup $\mathbb{Z}_d$ admits an orbit of dimension $N/d$. However, this does not answer the questions of how many orbits exist or, more specifically, of how many orbits of dimension $N/d$ exist.

The first question can be answered using the Cauchy–Frobenius lemma.[16] It asserts that the total number of orbits, $|\mathcal{S}_{\mathrm{BE}}/\mathbb{Z}_N^{(0)}|$, is given by the sum over $\gamma \in \mathbb{Z}_N^{(0)}$ of the fixed points under the $\gamma$-action, divided by the order of the group.[17] Since the number of fixed points are simply given by the expectation values of the $\mathcal{U}^\gamma(\mathcal{C})$ operator on the two-torus, we have

$$\left| \mathcal{S}_{\mathrm{BE}}/\mathbb{Z}_N^{(0)} \right| = \frac{1}{N} \sum_{\gamma \in \mathbb{Z}_N^{(0)}} \langle \mathcal{U}^\gamma(\mathcal{C}) \rangle_{T^2} \,. \tag{178}$$

---

[16]This lemma is also known as the orbit-counting theorem, or the *lemma that is not Burnside's* [81].

[17]Recall the lemma: Let $G$ be a finite group that acts on a set $X$. The lemma asserts that $|X/G| = \frac{1}{|G|}\sum_{g \in G}|X^g|$, where $X^g$ are the $g$-fixed points in $X$. The proof goes as follows: write $\sum_{g \in G}|X^g| = \sum_{x \in X}|G_x|$, where $|G_x| \subset G$ is the stabiliser subgroup of $x \in X$. This identity simply re-expresses the sum over all $(g,x) \in G \times X$ with $g \cdot x = x$. Using the orbit-stabiliser theorem $|G_x| = |G|/|G \cdot x|$, we write this as a sum over orbits $G \cdot x$. Finally, every element $x$ in a given orbit has the same orbit length,

$$\sum_{g \in G}|X^g| = \sum_{x \in X}|G_x| = |G|\sum_{x \in X}\frac{1}{|G \cdot x|} = |G|\sum_{A \in X/G}\sum_{x \in A}\frac{1}{|A|} = |G||X/G| \,. \tag{177}$$

Sums of these kinds occur when gauging the 0-form symmetry – of course, on $T^2$ we essentially need two such insertions, which we will discuss in detail below. Amusingly, by 3d modularity, the sum (178) is also equal to the sum over $\langle \Pi^\gamma \rangle$, namely the gauging of $\mathbb{Z}_N^{(1)}$ given by (157) with $s = 0$, so that the total number of $\mathbb{Z}_N^{(0)}$ orbits is also the number $\left| \mathcal{S}_{\mathrm{BE}}^{\vartheta_{s=0}} \right|$ of ground states in the $\theta_s = 0$ universe of the decomposition imposed by $\mathbb{Z}_N^{(1)}$:

$$\left| \mathcal{S}_{\mathrm{BE}} / \mathbb{Z}_N^{(0)} \right| = \left| \mathcal{J}_{N,K}^{s=0} \right| . \tag{179}$$

Labelling the group elements by $\gamma = n\gamma_0$, the sum (178) can be written as a sum over $n$. For an arbitrary integer $N$, due to (175), some terms inside the sum can be identical for different values of $n$ – this occurs for any two integers $n, n'$ related by $d(n) = d(n')$. Indeed, the sum over the full $\mathbb{Z}_N$ group collapses to a sum over the divisors $d$ of $N$ and $K$ only. We are thus interested in the distribution of subgroups generated by all elements of $\mathbb{Z}_N$. The number of elements of $\mathbb{Z}_N$ generating a $\mathbb{Z}_d$ subgroup for given $d$ is given by Euler's totient function $\varphi(d)$. It counts the positive integers up to a given integer $d$ that are relatively prime to $d$.[18] Clearly, $\varphi(1) = 1$, and $\varphi(d) = d - 1$ if $d$ is prime. The identity:

$$\sum_{d \mid n} \varphi(d) = n , \tag{180}$$

due to Gauss, partitions any integer $n \in \mathbb{N}$ into the Euler totients of its positive divisors $d$. We list some useful properties of $\varphi$ in appendix E.2. This allows us to simplify the sum over $n$ as a sum over divisors $d$ of $N$ and $K$,

$$\sum_{n=0}^{N-1} \langle \mathcal{U}^{n\gamma_0}(\mathcal{C}) \rangle_{T^2} = \sum_{d \mid \gcd(N,K)} \varphi(d) \binom{\frac{K}{d} - 1}{\frac{N}{d} - 1} . \tag{181}$$

As a consistency check, let $N$ be prime and $N \mid K$. Then the only divisors are $d = 1$ and $d = N$, and using $\varphi(N) = N - 1$ we obtain $\langle 1 \rangle_{T^2} + N - 1$, which agrees with (171) since there are $N - 1$ non-identity elements in $\mathbb{Z}_N$. Assuming again the 3d modularity relation (169), (181) gives us furthermore the result (158) for the 1-form symmetry gauging in the case where $N$ is prime.

The expression (181) is always a multiple of $N$. This is not clear from the sum itself, and generally no summand inside the sum (181) is divisible by $N$. Rather, the divisibility is a consequence of the Cauchy–Frobenius lemma (178):

$$\left| \mathcal{S}_{\mathrm{BE}} / \mathbb{Z}_N^{(0)} \right| = \frac{1}{N} \sum_{d \mid \gcd(N,K)} \varphi(d) \binom{\frac{K}{d} - 1}{\frac{N}{d} - 1} . \tag{182}$$

Note that this formula holds for any value of $N$ and $K$. For instance, in the maximally anomalous case where $K$ and $N$ do not share any divisors other than $d = 1$, we demonstrated above that all orbits are of length $N$. This is consistent with (182), which implies that there are $|\mathcal{S}_{\mathrm{BE}}| / N$ orbits of equal length, which must therefore be of length $N$.[19]

---

[18]In other words, it is the number of integers $k$ in the range $1 \le k \le d$ for which the greatest common divisor $\gcd(d, k)$ is equal to 1.

[19]The integer-valued number-theoretic function (182) is well-known in the combinatorics literature: The right-hand-side of (182) is known as the number of cycles of Bulgarian solitaire [82, 83], and it is also the number of *necklaces* of type $K$, with $N$ 0's and $K - N$ 1's [84]. See also the OEIS sequence A047996.

The Cauchy–Frobenius formula does *not* tell us directly how many orbits of a given dimension exist, beyond the total number of orbits given by (182). Indeed, we can imagine partitioning the orbits $\{\hat{\omega} \in \mathcal{S}_{\mathrm{BE}}/\mathbb{Z}_N^{(0)}\}$ into all orbits of the same length, or equivalently, into all orbits with fixed stabiliser. Since the stabiliser orders – or equivalently the orbit lengths – are all divisors of $N$, this suggests that we can rewrite (182) into a sum over orbits of length $d$. We will explain below that this is not a simple rearrangement of the sum – that is, the numbers of orbits of fixed lengths cannot, in general, be expressed in terms of simple binomial factors.

**Insertion of zero-form symmetry operators** Let us now consider the insertion of arbitrary line operators for the zero-form symmetry, as set up in section 2.2. Consider first the case $g = 1$, which has a spatial direction $\mathcal{C}$ and a Euclidean time direction $\tilde{\mathcal{C}}$. According to (97), the general insertion of $\mathcal{U}^\gamma(\mathcal{C})\,\mathcal{U}^{\gamma'}(\tilde{\mathcal{C}})$ on $T^2$ counts the number of fixed points under both $\gamma$ and $\gamma'$. This amounts to solving a slightly modified counting problem: if some Bethe vacuum $\hat{u}$ is fixed under $\gamma \in \mathbb{Z}_N$, it is consequently also fixed under $n\gamma$, for any $n$. Thus if $n'$ is a multiple of $n$, then the fixed points under both $n\gamma_0$ and $n'\gamma_0$ are simply the ones fixed under $n\gamma_0$ (and vice versa). But even if neither $n$ or $n'$ is a multiple of the other, they can still be embedded into a larger subgroup of $\mathbb{Z}_N$.[20]

To make this precise, consider two group elements $n\gamma_0$ and $n'\gamma_0$, labelled by $n, n' = 1, \ldots, N-1$. Then $\langle n\gamma_0 \rangle$ and $\langle n'\gamma_0 \rangle$ generate subgroups of order $\frac{N}{\gcd(n,N)}$ and $\frac{N}{\gcd(n',N)}$. Both of these groups sit in the larger group $\langle \gcd(n,n')\gamma_0 \rangle$ inside $\mathbb{Z}_N$, in the sense that the latter contains both elements $n\gamma_0$ and $n'\gamma_0$. The subgroup $\langle \gcd(n,n')\gamma_0 \rangle = \mathrm{H}^{(0)}_{(n\gamma,n'\gamma)}$ is precisely the one described above equation (99). Therefore, the vacua that are fixed under both $n\gamma_0$ and $n'\gamma_0$ are exactly those fixed by $\gcd(n,n')\gamma_0$, hence we have:

$$\langle \mathcal{U}^{n\gamma_0}(\mathcal{C})\,\mathcal{U}^{n'\gamma_0}(\tilde{\mathcal{C}}) \rangle_{T^2} = \langle \mathcal{U}^{\gcd(n,n')\gamma_0}(\mathcal{C}) \rangle_{T^2}. \tag{183}$$

We can alternatively understand this identity from the geometry of the two-torus inside $T^3 = T^2 \times S^1$. We can associate to the operator $\mathcal{U}^{n\gamma_0}(\mathcal{C})\,\mathcal{U}^{n'\gamma_0}(\tilde{\mathcal{C}})$ the cycle $n\mathcal{C} + n'\tilde{\mathcal{C}}$ in the first homology $H_1(T^2, \mathbb{Z})$. The mapping class group of $T^2$ allows us to simplify this cycle to $\gcd(n,n')\mathcal{C}$, which is again associated to the operator $\mathcal{U}^{\gcd(n,n')\gamma_0}(\mathcal{C})$.

Using (176), we can make (183) completely explicit. Since the gcd is associative, it follows that:

$$\langle \mathcal{U}^{n\gamma_0}(\mathcal{C})\,\mathcal{U}^{n'\gamma_0}(\tilde{\mathcal{C}}) \rangle_{T^2} = \binom{\frac{K}{N}\gcd(n,n',N)-1}{\gcd(n,n',N)-1} \delta_{K \bmod \frac{N}{\gcd(n,n',N)},0}. \tag{184}$$

The insertion of an arbitrary topological line for $\mathbb{Z}_N^{(0)}$ on $\Sigma_g$ follows analogously. When inserting the generic topological operator $\mathcal{U}^{\boldsymbol{\gamma}}$ (93), we sum over the fixed points $\mathcal{S}_{\mathrm{BE}}^{(\boldsymbol{\gamma})}$, where $\boldsymbol{\gamma} = (\gamma_i) \in \mathbb{Z}_N^{2g}$. For $g > 1$, the insertion furthermore involves the evaluation of the handle-gluing operator $\mathcal{H}$ on the fixed points, as in (98). Unlike in the case where there is only one fixed point and we were able to evaluate (172), we do not have such a simple formula available in the case of multiple fixed points. Indeed, we do not expect such a simple formula for arbitrary $\boldsymbol{\gamma}$ – after all, even for $\boldsymbol{\gamma}$ the trivial element, $\langle \mathcal{U}^{\boldsymbol{\gamma}=0} \rangle_{\Sigma_g} = \langle 1 \rangle_{\Sigma_g}$ is given by the Verlinde formula (139), for which no 'simpler' algebraic expression is known.

---

[20]As an example, consider $N = 12$, where the subgroups are depicted in figure 8. Then $n = 4$ and $n' = 6$ generate a $\mathbb{Z}_3$ and $\mathbb{Z}_2$ subgroup of $\mathbb{Z}_{12}$, respectively. However, even though the orders of the subgroups are coprime, they both sit in a $\mathbb{Z}_{\mathrm{lcm}(2,3)} = \mathbb{Z}_6$ subgroup of $\mathbb{Z}_{12}$, generated of course by $2\gamma_0$. The fixed points under both $4\gamma_0$ and $6\gamma_0$ thus include the fixed points under $2\gamma_0$. But these are in fact all fixed points that are shared by $4\gamma_0$ and $6\gamma_0$. This is because the group generated by $4\gamma_0$ and $6\gamma_0$ necessarily contains $2\gamma_0$, and thus if $\hat{u}$ is fixed by both $4\gamma_0$ and $6\gamma_0$, then it is also fixed by $2\gamma_0$.

### 3.1.3 't Hooft anomaly and the allowed $(SU(N)/\mathbb{Z}_r)_K$ CS theories

The 't Hooft anomaly (76) can be computed from the twisted superpotential (129). It reads:

$$\mathcal{A}(\gamma_{(0)}, \gamma_{(1)}) = \gamma_{(1)} K \gamma_{(0)} \bmod 1 = \frac{(N-1)K}{N} nm \bmod 1, \tag{185}$$

hence the anomaly coefficient that enters in (83) is $\mathfrak{a} = -K \bmod N$. Here we parameterise $\mathbb{Z}_N^{(1)} \oplus \mathbb{Z}_N^{(0)}$ by the two integers $m, n \in \mathbb{Z}_N$, with $\gamma_{(1)} = m\gamma_0$ and $\gamma_{(0)} = n\gamma_0$. We thus have the anomalous commutator:

$$\Pi^{m\gamma_0} \mathcal{U}^{n\gamma_0} = e^{-2\pi i mn \frac{K}{N}} \mathcal{U}^{m\gamma_0} \Pi^{n\gamma_0}. \tag{186}$$

This directly implies that certain mixed correlation functions must vanish:

$$mn\frac{K}{N} \notin \mathbb{Z} \quad \Longrightarrow \quad \langle \Pi^{m\gamma_0} \mathcal{U}^{n\gamma_0}(\mathcal{C}) \rangle_{\Sigma_g} = 0. \tag{187}$$

Note that having an anomalous commutator is a sufficient condition for the correlator to vanish, but not a necessary one. In fact, 3d modularity implies that the mixed correlator in (187) vanishes if and only if $d = N / \gcd(m, n, N)$ is not a divisor of $K$, that is, if $\gcd(m, n, N)K/N$ is not an integer – this will be discussed in subsection 3.2 below.

It is interesting to look for subsets of topological operators that are mutually commuting, so that the corresponding symmetries can be gauged. These correspond to $n, m$ such that:

$$mn\frac{K}{N} \in \mathbb{Z} \quad \Longleftrightarrow \quad mn \in \frac{N}{\gcd(N, K)}\mathbb{Z}. \tag{188}$$

As discussed above (see (174) in particular), each element $n \in \mathbb{Z}_N$ generates a subgroup $\mathbb{Z}_{d(n,N)} \subseteq \mathbb{Z}_N$. Then, one finds that the possible non-anomalous subgroups $\mathbb{Z}_d^{(1)} \oplus \mathbb{Z}_{d'}^{(0)} \subseteq \mathbb{Z}_N^{(1)} \oplus \mathbb{Z}_N^{(0)}$ are:

$$\left\{ \mathbb{Z}_d^{(1)} \oplus \mathbb{Z}_{d'}^{(0)} \,\middle|\, d, d'|N \text{ and } dd' \in \gcd(N, K)N\mathbb{Z} \right\}. \tag{189}$$

We are particularly interested in the case $d = d' \equiv r$, so that $\mathbb{Z}_r$ can be interpreted as a one-form symmetry in 3d. These are the subgroups:

$$\left\{ (\mathbb{Z}_r^{(1)})_{3d} \subseteq (\mathbb{Z}_N^{(1)})_{3d} \,\middle|\, r|N \text{ and } r^2 \in \gcd(N, K)N\mathbb{Z} \right\}, \tag{190}$$

which simply correspond to a particular subset of divisors of $N$. Note that having a non-anomalous $(\mathbb{Z}_r^{(1)})_{3d}$ is equivalent to the existence of the corresponding Chern–Simons theory on spin three-manifolds:

$$\frac{KN}{r^2} \in \mathbb{Z} \quad \Longleftrightarrow \quad \text{3d } \mathcal{N} = 2 \text{ CS } (SU(N)/\mathbb{Z}_r)_K \text{ exists.} \tag{191}$$

See *e.g.* [31, 48] for recent discussions of this condition.

### 3.2 Mixed correlators and a modular anomaly on $T^3$

It is interesting to compute the action the $\mathbb{Z}_N^{(1)}$ operators on Bethe states fixed by (part of) $\mathbb{Z}_N^{(0)}$. In the simplest case, consider the case where $\hat{u}^{(\mathbb{Z}_N)}$ is fixed by the full $\mathbb{Z}_N^{(0)}$. Such vacua only occur in the non-anomalous case, $\kappa \equiv \frac{K}{N} \in \mathbb{Z}$, and the calculation (D.2) in Appendix D.1 shows that:

$$\Pi^{\gamma_0} |\hat{u}^{(\mathbb{Z}_N)}\rangle = (-1)^{(N-1)(\kappa-1)} |\hat{u}^{(\mathbb{Z}_N)}\rangle. \tag{192}$$

This directly implies that:

$$\langle \Pi^{\gamma_0} \mathcal{U}^{\gamma_0}(\mathcal{C}) \rangle_{T^2} = \sum_{\hat{u} \in \mathcal{S}_{BE}^{(\gamma_0)}} \langle \hat{u} | \Pi^{\gamma_0} | \hat{u} \rangle = \langle \hat{u}^{(\mathbb{Z}_N)} | \Pi^{\gamma_0} | \hat{u}^{(\mathbb{Z}_N)} \rangle = (-1)^{(N-1)(\kappa-1)}. \tag{193}$$

Hence we have:

$$\langle \Pi^{\gamma_0} \mathcal{U}^{\gamma_0}(\mathcal{C}) \rangle_{T^2} = (-1)^{(N-1)(\kappa-1)} \langle \mathcal{U}^{\gamma_0}(\mathcal{C}) \rangle_{T^2}. \tag{194}$$

On $T^3$, we should be able to map the 1-cycle $\mathcal{C} + [S_A^1]$ to the 1-cycle $\mathcal{C}$ at no cost by a 3d modular transformation, hence the non-trivial sign when $N$ and $\kappa$ are both even is interpreted as a non-trivial 3d modular anomaly. (Note that the simplest instance of this anomaly is for $N = 2$ and $K \in 4\mathbb{Z}$, as discussed already in (20) in section 1.1.)

More generally, we may consider the action of $\Pi^{m\gamma_0}$ on the Bethe states fixed by $\mathbb{Z}_d^{(0)} \subseteq \mathbb{Z}_N^{(0)}$:

$$\Pi^{m\gamma_0} | \hat{u}^{(\mathbb{Z}_d)} \rangle = (-1)^{m(N-1)} e^{-2\pi i m \frac{\ell}{N}} | \hat{u}^{(\mathbb{Z}_d)} \rangle. \tag{195}$$

Here, the action is generally by a $N$-th root of unity, and different states is the same orbit $\hat{\omega}$ with $\text{Stab}(\hat{\omega}) = \mathbb{Z}_d$ can have different values of $\ell$ (they differ by integer multiples of $K$ mod $N$, as implied by (166)). The explicit computation of mixed correlations functions on $T^3$ is therefore more involved. We find:

$$\langle \Pi^{m\gamma_0} \mathcal{U}^{n\gamma_0}(\mathcal{C}) \rangle_{T^2} = (-1)^{mn(\frac{K}{N}+1)(N-1)} \langle \mathcal{U}^{\gcd(m,n)\gamma_0}(\mathcal{C}) \rangle_{T^2}, \tag{196}$$

which naturally generalises (194). Here the correlators on both sides vanish whenever $mn\left(\frac{K}{N}+1\right) \notin \mathbb{Z}$, which can be shown to follow from (184). For the non-zero correlators, we find the non-trivial sign in (196) – this give us the general form of the 3d modular anomaly on $T^3$.

This anomaly should be understood as a 3d mixed anomaly between gravity (the 3d Lorentz group) and the one-form symmetry $(\mathbb{Z}_N^{(1)})_{3d}$, or more generally between gravity and a non-anomalous subgroup $(\mathbb{Z}_r^{(1)})_{3d}$ for $r|N$. The corresponding four-dimensional anomaly theory takes the form [42]:

$$S_{\text{grav anomaly}} = 2\pi h \int_{\mathfrak{M}_4} w_2(\mathfrak{M}_4) \cup B^{(r)}, \qquad h \equiv \frac{(K-N)N(r-1)}{2r^2} \text{ mod } 1, \tag{197}$$

where $w_2(\mathfrak{M}_4) \in H^2(\mathfrak{M}_4, \mathbb{Z}_2)$ is the second Stiefel–Whitney class of the 4-manifold and $B^{(r)}$ is the $(\mathbb{Z}_r^{(1)})_{3d}$ background gauge field. This is a $\mathbb{Z}_2$-valued anomaly: for $(\mathbb{Z}_r^{(1)})_{3d}$ a non-anomalous subgroup, the anomaly coefficient $h$ in (197) takes the values $h = 0$ or $\frac{1}{2}$, as one can readily check $e.g.$ using the property (191) that $\frac{KN}{r^2} \in \mathbb{Z}$. In fact, one can check that:

$$h = \begin{cases} \frac{1}{2}, & \text{if } N \text{ is even}, \frac{N}{r} \text{ is odd and } \frac{K}{r} \text{ is even}, \\ 0, & \text{otherwise}. \end{cases} \tag{198}$$

Indeed, we have $h = \frac{1}{2}$ if and only if the sign appearing in (196) is non-trivial.[21] Physically, this anomaly is best understood by considering the spin of the topological lines in the underlying bosonic 3d TQFT [42], which is the $\mathcal{N} = 0$ $SU(N)_k$ CS theory with $k = K - N$. As we will discuss further in section 4.1, the $\mathbb{Z}_N$ one-form symmetry is generated by a particular Wilson line in 3d, the abelian anyon denoted by $a \cong \mathcal{U}^{\gamma_0}$ that satisfies $a^N = 1$ under fusion. Given the $(\mathbb{Z}_N^{(1)})_{3d}$ 't Hooft anomaly $\mathfrak{a} = -K \text{ mod } N$, one can show that the 3d spin of the invertible symmetry line $a^n \cong \mathcal{U}^{n\gamma_0}$ is given by:

$$h[a^n] = \frac{(K-N)n(N-n)}{2N} \text{ mod } 1. \tag{199}$$

---

[21]This is explained in detail in appendix D.3.

The line $a^n$ generates the subgroup $(\mathbb{Z}_r^{(1)})_{3d}$ with $r = N/n$, assuming $\gcd(N, n) = n$ without loss of generality. Hence we have:

$$h[a^n] = \frac{(K-N)N(r-1)}{2r^2} \bmod 1. \tag{200}$$

This satisfies $h[a^n] \in \frac{1}{2}\mathbb{Z}$ if and only $(\mathbb{Z}_r^{(1)})_{3d}$ is non-anomalous. Furthermore, we directly notice that $h[a^n]$ exactly reproduces the gravity-$(\mathbb{Z}_r^{(1)})_{3d}$ anomaly coefficient $h$ in (197), not coincidentally, hence the generating line $a^n$ has half-integer spin depending on the properties of $N$, $\frac{K}{r}$ and $\frac{N}{r}$ exactly as in (198).

The generating line $a^n$ being non-anyonic (that is, having $h \in \frac{1}{2}\mathbb{Z}$) is necessary for having a consistent gauging of the 3d TQFT. While the underlying $\mathcal{N} = 0$ CS theory $SU(N)_{K-N}$ is a bosonic TQFT, the CS theory $(SU(N)/\mathbb{Z}_r)_{K-N}$ that results from the $\mathbb{Z}_r$ one-form gauging is actually a spin-TQFT if and only if $h = \frac{1}{2}$, as explained in [42]. The fact that gauging a one-form symmetry in a bosonic TQFT results in a spin-TQFT is the most physical way to understand the existence of the mixed anomaly (197) – see $e.g.$ [42,85]. Thus, we also find the condition for the bosonic CS theory after gauging to be fermionic – that is, a spin-TQFT:

$$\mathcal{N} = 0 \ (SU(N)/\mathbb{Z}_r)_{K-N} \text{ is a spin-TQFT} \iff N \text{ even}, \frac{N}{r} \text{ odd and } \frac{K}{r} \text{ even.} \tag{201}$$

Correspondingly, the 2d WZW$[G_{k=K-N}]$ models for $G = SU(N)/\mathbb{Z}_r$ are fermionic CFTs in those cases. For $r = N$, this is the well-known statement that the $\mathcal{N} = 0 \ PSU(N)_{k=K-N}$ CS theory is a spin-TQFT if only if $N$ is even and $\frac{k}{N}$ is odd [40]. The result (201) is also in agreement with the literature on 2d (bosonic) WZW$[G]$ for $G$ non-simply-connected [40, 41, 86–88]. When $h[a] = \frac{1}{2}$, we should actually replace $a^n$ by $\widetilde{a}^n = a^n \psi$, where $\psi$ is the transparent line that couples to the spin structure of the 3-manifold. Then $h[\widetilde{a}^n] = 0$ and we can proceed with anyon condensation as in the bosonic case [42]. In this sense, we can always trivialise the $\mathbb{Z}_2$ anomaly (197) because the $\psi$ line always exists in our supersymmetric field theories. Nonetheless, the interpretation in terms of Bethe vacua is non-trivial, as we start with Bethe vacua that are all bosonic and build Bethe states in the $G = \widetilde{G}/\Gamma$ theory that can be fermionic, thus affecting the supersymmetric index counting, sometimes in intricate ways [21]. In this paper, we have only counted the Bethe states irrespective of their eigenvalue under $(-1)^F$, which does not give us an index in general. When this 'modular anomaly' vanishes, however, all Bethe states are bosonic and thus the naive sum over the Bethe states of the $G$ theory does give us the correct twisted index.

## 3.3 The $PSU(N)_K$ twisted index for $N$ prime

In this subsection, we study the case of $N$ an odd prime number.[22] For $N$ prime, $\mathbb{Z}_N$ has no non-trivial subgroup and every non-zero element $n \in \mathbb{Z}_N$ is a generator. Then, as explained above, there is a single fixed point if $\mathfrak{a} \equiv -K \bmod N = 0$ for every $\gamma = n\gamma_0$, and no fixed point if $\mathfrak{a} \neq 0$. Indeed, this follows from our general discussion of the $\Gamma^{(0)}$ orbits of Bethe vacua in section 2.1.2. For $N$ prime, we either have no 't Hooft anomaly ($\mathfrak{a} = 0$) or the maximal anomaly ($\mathfrak{a} \neq 0$); in the former case, we have a unique orbit of dimension 1 and all the other orbits are of dimension $N$, while in the latter case all orbits must be of dimension $N$. This directly implies that the Witten index of the $\mathbb{Z}_N^{(0)}$-gauged $A$-model is given by:

$$Z_{T^3}\left[SU(N)_K/\mathbb{Z}_N^{(0)}\right] = \frac{1}{N}\binom{K-1}{N-1} + \delta_{\mathfrak{a},0}\left(-\frac{1}{N} + N\right), \tag{202}$$

---

[22]The case of an even prime number, $N = 2$, was discussed in the introduction. It involves additional subtleties common to all even $N$, which we will address in the next subsections.

where we are just counting the number of ground states as in (115), including the twisted sectors. Indeed, for $\mathfrak{a} = 0$ we have $(\binom{K-1}{N-1} - 1)/N$ maximal orbits plus $N$ twisted sectors from the length-1 orbit, while for $\mathfrak{a} \neq 0$ we only have $\binom{K-1}{N-1}/N$ maximal orbits.

Let us now gauge the $\mathbb{Z}_N^{(1)}$ and $\mathbb{Z}_N^{(0)}$ separately, before combining the two if $\mathfrak{a} = 0$. The gauging of $\mathbb{Z}_N^{(1)}$ was already discussed in subsection 3.1.1. In particular, the Witten index of the $\mathbb{Z}_N^{(1)}$-gauged $A$-model for $N \geq 3$ prime is given in (158). If we only gauge $\mathbb{Z}_N^{(0)}$ instead, we have:

$$Z_{\Sigma_g \times S^1}\left[ SU(N)_K / \mathbb{Z}_N^{(0)} \right] = \frac{1}{N^{2g-1}}\left( \langle 1 \rangle_{\Sigma_g} + (N^{2g} - 1)\langle \mathcal{U}^{\gamma_0}(\mathcal{C}) \rangle_{\Sigma_g} \right), \tag{203}$$

where $\mathcal{C}$ is any basis element of $H_1(\Sigma_g, \mathbb{Z})$. Here, we used the fact that, for $N$ prime, any insertion of the form (93) is equivalent to inserting $\mathcal{U}^{\gamma_0}(\mathcal{C})$ because $\mathbb{Z}_N$ has no non-trivial subgroups. Note that we will not keep track of the dual 1-form symmetry in what follows – that is, we choose $C^D = 0$ in the notation of section 2.4. For $g = 1$, we directly see that (203) reproduces (202).

**The Witten index of the $PSU(N)_K$ theory ($N$ prime)** For $K \in N\mathbb{Z}$, the 3d 1-form symmetry is non-anomalous and we can gauge it fully. We can thus compute the full partition function on $\Sigma_g \times S^1$ of the resulting $PSU(N)_K$ theory, including the dependence on $\theta_s$, as:

$$Z_{\Sigma_g \times S^1}\left[ PSU(N)_K^{\theta_s} \right] = \frac{1}{N^{2g}} \sum_{m=0}^{N-1} e^{2\pi i \frac{sm}{N}} \left( \langle \Pi^{m\gamma_0} \rangle_{\Sigma_g} + (N^{2g} - 1)\langle \Pi^{m\gamma_0} \mathcal{U}^{\gamma_0}(\mathcal{C}) \rangle_{\Sigma_g} \right). \tag{204}$$

In particular, setting $g = 1$, we find the Witten index of the $PSU(N)_K$ $\mathcal{N} = 2$ CS theory with $\theta_s$ turned on:

$$\mathbf{I}_W\left[ PSU(N)_K^{\theta_s} \right] = \frac{1}{N^2}\left( \binom{K-1}{N-1} - 1 \right) + \delta_{0,s} N. \tag{205}$$

It is a non-trivial mathematical fact that (205) defines an integer for any prime number $N$ and $K \in N\mathbb{Z}$. Let us define the rational number $M_{N,K}$ through the decomposition:

$$\binom{K-1}{N-1} = 1 + M_{N,K} N^2. \tag{206}$$

For $N \geq 3$ prime, the numbers $M_{N,2N}$ are integers called Babbage quotients [89], while $\frac{1}{N} M_{N,2N}$ for $N \geq 5$ prime are integers known as Wolstenholme quotients [90]. Crucially, if $N \geq 3$ is prime and $N|K$, then $M_{N,K}$ is an integer. This is a consequence of Glaisher's theorem [91], which we discuss in some detail in appendix E.1 (see in particular Proposition 1). From (206), we therefore have that:

$$\mathbf{I}_W\left[ PSU(N)_K^{\theta_s} \right] = M_{N,K} + N\delta_{s,0}, \tag{207}$$

is the Witten index of the pure $PSU(N)_K$ CS theory with theta angle $\theta_s$. This gives the seemingly arbitrarily defined integer $M_{N,K}$ a physical meaning as the Witten index of the $PSU(N)_K^{\theta_s}$ theory with $\theta_s \neq 0$.

**The higher-genus twisted index of the $PSU(N)_K$ theory ($N$ prime)** For any $g$, the twisted index (204) can be written as:

$$Z_{\Sigma_g \times S^1}\left[ PSU(N)_K^{\theta_s} \right] = \frac{1}{N^{2g-1}}\left( \sum_{\hat{u} \in \mathcal{S}_{BE}^{\theta_s}} \mathcal{H}(\hat{u})^{g-1} + (N^{2g} - 1) \sum_{\hat{u} \in \mathcal{S}_{BE}^{\theta_s} \cap \mathcal{S}_{BE}^{(\gamma_0)}} \mathcal{H}(\hat{u})^{g-1} \right)$$

$$= \frac{1}{N^{2g-1}}\left( \sum_{\underline{l} \in \mathcal{J}_{N,K}^s} \mathcal{H}(\hat{u}_{\underline{l}})^{g-1} + (N^{2g} - 1)\delta_{s,0}\left( \frac{K}{N} \right)^{(g-1)(N-1)} \right), \tag{208}$$

where the set of Bethe vacua in the universe set by $\theta_s$ was defined in (154), and in the second line we used (172) and the fact that the unique $\mathbb{Z}_N^{(0)}$ fixed point lives in the $\theta_s = 0$ sector. One easily checks that setting $g = 1$ in (208), together with (158), reproduces (205).

**Explicit examples** For $N$ prime and any $g$, we find:

$$Z_{\Sigma_g \times S^1}\big[PSU(N)_N^0\big] = N. \tag{209}$$

With some effort, we can work out some closed-form formulas as a function of $g$ in some special instances. Two such examples are:

$$Z_{\Sigma_g \times S^1}\big[PSU(3)_6^0\big] = 4^g, \tag{210}$$

$$\begin{aligned}
Z_{\Sigma_g \times S^1}\big[PSU(5)_{10}^0\big] = \frac{1}{400}\Big(&\big(9 + 4\sqrt{5}\big)4^{-6g}5^{-5g}\big(\sqrt{5} - 1\big)^{4g}\big(3\ 20^{5g}\big(\sqrt{5} - 5\big)^{2g} \\
&+ 2\big(5 - \sqrt{5}\big)^{7g}\big(5 + \sqrt{5}\big)^{5g}\big) + 5\big(9 - 4\sqrt{5}\big)\big(1 + \sqrt{5}\big)^{4g} \\
&\times \Big(\frac{1}{2}\big(5 + \sqrt{5}\big)\Big)^{2g} + 32\ 5^{g+1} + 75\ 2^{4g+1}\Big).
\end{aligned} \tag{211}$$

Amazingly, the latter expression is an integer. For $g = 0,\ 1,\ 2,\ 3,\ 4,\ 5, \ldots$, we have:

$$Z_{\Sigma_g \times S^1}\big[PSU(5)_{10}^0\big] = 1,\ 10,\ 1546,\ 2062386,\ 2958360826,\ 4246815114466,\ \ldots \tag{212}$$

More generally, we checked numerically, in many cases, that the formula (208) always returns an integer, as expected physically.

## 3.4 Gauging the 3d one-form symmetry on $T^3$

Let us now compute the Witten index – that is, the $T^3$ partition function – for the 3d $(SU(N)/\mathbb{Z}_r)_K$ $\mathcal{N} = 2$ Chern–Simons theories, for any $N$ and $r$. We do this by gauging a non-anomalous $\mathbb{Z}_r^{(1)} \oplus \mathbb{Z}_r^{(0)}$ symmetry in the $A$-model on $T^2$.

**Gauging $\mathbb{Z}_N^{(0)}$ on $T^2$** We first consider gauging only $\mathbb{Z}_N^{(0)}$, which corresponds to summing over all insertions on $T^2$. Using (184), we have:

$$\begin{aligned}
Z_{T^3}\big[SU(N)_K/\mathbb{Z}_N^{(0)}\big] &= \frac{1}{N} \sum_{n,n' \in \mathbb{Z}_N} \langle \mathcal{U}^{n\gamma_0}(\mathcal{C})\,\mathcal{U}^{n'\gamma_0}(\widetilde{\mathcal{C}})\rangle_{T^2} \\
&= \frac{1}{N} \sum_{n,n' \in \mathbb{Z}_N} \binom{\frac{K}{N}\gcd(n,n',N) - 1}{\gcd(n,n',N) - 1} \delta_{K \bmod \frac{N}{\gcd(n,n',N)},0},
\end{aligned} \tag{213}$$

This can be written more elegantly. Similarly to the single sum (181), the double sum in (213) can be reorganised as a sum over the divisors $d$ of $N$. Thus, for each divisor $d$, we need to count the pairs $(n, n') \in \mathbb{Z}_N^2$ such that $\gcd(n, n')\gamma_0$ generates $\mathbb{Z}_d \subseteq \mathbb{Z}_N^{(0)}$. This number is determined by Jordan's totient function $J_k(d)$ (which we review below) for $k = 2$, so that:

$$Z_{T^3}\big[SU(N)_K/\mathbb{Z}_N^{(0)}\big] = \frac{1}{N} \sum_{d \mid \gcd(N,K)} J_2(d) \binom{\frac{K}{d} - 1}{\frac{N}{d} - 1}. \tag{214}$$

For $N$ prime, in particular, we have $J_2(N) = N^2 - 1$ and $J_2(1) = 1$, and then one can easily check that (214) reproduces (202). We also checked 'experimentally' that (214) returns an integer for any $N$.[23] This provides an important consistency check of our overall normalisation, consistent with the discussion of section 2.4. Indeed, (214) should be integer because it is the 2d Witten index of the orbifolded $A$-model.

---

[23]We checked this statement numerically for $N, K \leq 4000$. A mathematical proof would be nice to have.

**Basics of Jordan's totient function**    Let us briefly review Jordan's totient function $J_k(d)$ for general $k$, as it will be useful below. Given two positive integers $k$ and $d$, $J_k(d)$ equals the number of $k$-tuples of positive integers $(n_i)_{i=1}^k$ that are less or equal to $d$ and such that the $k+1$ integers $(n_1, \cdots, n_k, d)$ are coprime. It can be explicitly computed as:

$$J_k(d) = d^k \prod_{p|d} \left(1 - \frac{1}{p^k}\right), \tag{215}$$

where $p$ ranges through all prime divisors of $d$. Note that this obviously generalises Euler's totient function $\varphi(d) = J_1(d)$. The Jordan's totient function allows us to decompose the $k$-power of any integer $N$ into the divisors of $N$:

$$\sum_{d|N} J_k(d) = N^k. \tag{216}$$

Note also that $J_k(1) = 1$ and that, for any prime number $p$, we have $J_k(p) = p^k - 1$. We list further relevant properties of $J_k(d)$ in appendix E.2.

Jordan's totient function $J_2(d)$ has appeared sporadically in the physics literature, for instance in the context of cyclic group orbifolds in monstrous moonshine [92, 93], and in the enumeration of the allowed electric-magnetic charge lattices in 4d $\mathcal{N} = 4$ SYM with gauge algebra $\mathfrak{su}(N)$ [94]. More relevantly to the present context, this function has appeared in the mathematical literature on dimensions of Verlinde bundles over curves [36, 50–53].

**The (naive) Witten index for $PSU(N)_K$ for any $N$ and $\theta_s = 0$**    Assuming for now that $\mathfrak{a} = -K \bmod N = 0$, we can gauge the full $\mathbb{Z}_N^{(1)} \oplus \mathbb{Z}_N^{(0)}$ to obtain the 3d Witten index of the $PSU(N)_K$ $\mathcal{N} = 2$ CS theory. This amounts to summing the topological lines over all three generators of $H_1(T^3, \mathbb{Z}) \cong \mathbb{Z}^3$:

$$\mathbf{I}_W[PSU(N)_K^0] = Z_{T^3}\left[PSU(N)_K^0\right] = \frac{1}{N^2} \sum_{m,n,n' \in \mathbb{Z}_N} \langle \Pi^{m\gamma_0} \mathcal{U}^{n\gamma_0}(\mathcal{C}) \, \mathcal{U}^{n'\gamma_0}(\widetilde{\mathcal{C}}) \rangle_{T^2}. \tag{217}$$

Whenever we have the full 3d modularity, with the trivial sign in (196), it is clear from the above discussions that we can massage the sum (217) into a sum similar to (214), with $J_2$ replaced with $J_3$ – see the definition of $J_k$ above (215). The modular anomaly (196) spoils this naive expectation, however. In fact, one can easily check that this naive expectation would lead to non-integer results for the index in examples where the $T^3$ anomaly is non-trivial – thus, the subtle sign in (196) is absolutely crucial to obtain sensible physical results.

Given (196) together with the $T^2$ modularity (183), we have:

$$\langle \Pi^{m\gamma_0} \mathcal{U}^{n\gamma_0}(\mathcal{C}) \mathcal{U}^{n'\gamma_0}(\widetilde{\mathcal{C}}) \rangle_{T^2} = (-1)^{m \gcd(n,n')\left(\frac{K}{N}+1\right)(N-1)} \langle \mathcal{U}^{\gcd(m,n,n')\gamma_0}(\mathcal{C}) \rangle_{T^2}. \tag{218}$$

Now we need to count the triples $(m, n, n')$ so that $\gcd(m, n, n')\gamma_0$ generates $\mathbb{Z}_d$ with some specific signs. For each divisor $d|N$, we can partition Jordan's totient function as $J_3(d) = n_+ + n_-$, where the contributions $n_\pm$ are those that come with the $\pm$ signs. Due to the specific partially symmetric form of the sign in (218), the number of positive and negative terms are related as $3n_+ = 4n_-$ for any divisor $d$ giving rise to a minus sign. We are then interested in the number $n_+ - n_- = \frac{1}{7}J_3(d)$ after the cancellation. This suggests to use the following refinement of Jordan's totient $J_3(d)$, which depends on specific arithmetic properties of $K$ and $N$:

$$\mathscr{J}_3^{N,K}(d) \equiv \begin{cases} \frac{1}{7}J_3(d), & \text{for } N \text{ even}, \frac{N}{d} \text{ odd}, \frac{K}{d} \text{ even}, \\ J_3(d), & \text{otherwise}. \end{cases} \tag{219}$$

Then we have that, for all $N$ and all $K$:

$$\sum_{m,n,n'\in\mathbb{Z}_N}\langle\Pi^{m\gamma_0}\mathcal{U}^{n\gamma_0}(\mathcal{C})\,\mathcal{U}^{n'\gamma_0}(\widetilde{\mathcal{C}})\rangle_{T^2}=\sum_{d\,|\,\gcd(N,K)}\mathscr{I}_3^{N,K}(d)\binom{\frac{K}{d}-1}{\frac{N}{d}-1}. \tag{220}$$

Let us comment briefly on the non-trivial denominator in the definition (219). First of all, since $\mathscr{I}_3^{N,K}(d)=\frac{1}{7}J_3(d)$ only if $N$ is even and $\frac{N}{d}$ is odd, this value only occurs if $d$ is even. In that case, one can show that $7|J_3(d)$ for all even integers $d$.[24] This number has a rather natural interpretation as $J_3(2)=7$.[25] Thus we have that $\mathscr{I}_3^{N,K}:\mathbb{N}\to\mathbb{N}$ is a well-defined integral map for all values of $N$ and all $K$.

Coming back to the case $K|N$, we have thus written the $PSU(N)_K\ T^2$ index as:

$$\mathbf{I}_{\mathrm{W}}\big[PSU(N)_K^0\big]=\frac{1}{N^2}\sum_{d|N}\mathscr{I}_3^{N,K}(d)\binom{\frac{K}{d}-1}{\frac{N}{d}-1}. \tag{221}$$

We list this index for small allowed values of $N$ and $K$ in table 1. Since the full symmetry $(\mathbb{Z}_N^{(1)})_{3\mathrm{d}}$ is non-anomalous, the resulting $PSU(N)_K$ theory exists and the index should be a non-negative integer.[26] We conjecture this is the case, namely that:

$$\mathbf{I}_{\mathrm{W}}\big[PSU(N)_K^0\big]\in\mathbb{N}, \tag{222}$$

for any $N|K$.[27] As a small consistency check for the formula (221), consider the case where $N\geq 3$ is prime and $N|K$. Then there are only two terms in the sum, $d=1$ and $d=N$, and using $J_3(p)=p^3-1$ for $p$ prime (see appendix E.2) we precisely reproduce (205) for $s=0$. Another small check of (221) is that, for $N=2$, it precisely reproduces (22).[28]

As already mentioned, the naive index (222) is actually a supersymmetric index only when the 'modular anomaly' vanishes and the corresponding $\mathcal{N}=0$ CS theory $PSU(N)_{k=K-N}$ is bosonic (that is, if $(N-1)k/N$ is even).

**The (naive) Witten index for $(SU(N)/\mathbb{Z}_r)_K$** Given the above discussion, it is now straightforward to consider gauging a non-anomalous subgroup $\mathbb{Z}_r\subset(\mathbb{Z}_N^{(1)})_{3\mathrm{d}}$ on $T^3$. As described in subsection 3.1.3, we can find a non-anomalous subgroup as long as $N$ and $K$ have a common divisor. These common divisors are precisely labelled as the summation index in (220). This suggests that the 'maximal' non-anomalous subgroup we can gauge is $\mathbb{Z}_{\gcd(N,K)}$.

Summing over the topological lines on $T^3$ for the maximal non-anomalous subgroup $\mathbb{Z}_{\gcd(N,K)}\subseteq\mathbb{Z}_N$ also involves a sum over all non-anomalous subgroups contained in $\mathbb{Z}_N$. These are precisely the groups $\mathbb{Z}_d$ with $d|\gcd(N,K)$. The result for the torus partition function of the $(SU(N)/\mathbb{Z}_{\gcd(N,K)})_K$ theory then takes a very similar form as before, with normalisation factor $|\Gamma|^2$ adjusted to the cardinality of the group we sum over:

$$\mathbf{I}_{\mathrm{W}}\big[(SU(N)/\mathbb{Z}_{\gcd(N,K)})_K^0\big]=\frac{1}{\gcd(N,K)^2}\sum_{d\,|\,\gcd(N,K)}\mathscr{I}_3^{N,K}(d)\binom{\frac{K}{d}-1}{\frac{N}{d}-1}. \tag{223}$$

---

[24]We leave this as an exercise to the enthusiastic reader. See also appendix E.2 for more divisibility properties.

[25]Recall that $J_k(p)=p^k-1$ for $p$ prime and $k\in\mathbb{N}$. Jordan's totient is a multiplicative function, see (E.11). Thus in particular if $d$ is even and $\frac{d}{2}$ is odd, then $\frac{1}{7}J_3(d)=J_3(\frac{d}{2})$. Clearly this is not the case for all values $d$ of which $\mathscr{I}_3^{N,K}(d)$ is evaluated in (220): For instance, $d=4$ is an even divisor of $N=12$ such that $\frac{N}{d}$ is odd, but $\frac{d}{2}$ is even as well. Therefore, $\mathscr{I}_3^{N,K}(d)$ does not have a simple interpretation as being either $J_3(d)$ or $J_3(\frac{d}{2})$.

[26]This Witten index cannot be negative because it is also counting the lines in a 3d TQFT. See the discussion in section 4.1.

[27]We have checked this numerically for $N\leq K\leq 100000$.

[28]The function (221) seems to be relatively unexplored in the literature. For $K=3N$, $\mathbf{I}_{\mathrm{W}}[PSU(N)_{3N}^0]$ is the integer sequence A244036 and it appears in the study of chiral algebras [95, Section 3.1.2].

Table 1: The (naive) Witten index $\mathbf{I}_W\big[PSU(N)^0_{\kappa N}\big]$, evaluated from (221), for small values of $N$ and $\kappa \equiv \frac{K}{N} \in \mathbb{Z}$. The cases when the naive index is not a Witten index (when both $N$ and $\kappa$ are even) are shown in red.

| $\kappa\backslash N$ | 2 | 3 | 4 | 5 | 6 | 7 | 8 | 9 | 10 |
|---|---|---|---|---|---|---|---|---|---|
| 1 | 2 | 3 | 4 | 5 | 6 | 7 | 8 | 9 | 10 |
| 2 | 1 | 4 | 4 | 10 | 16 | 42 | 108 | 312 | 930 |
| 3 | 3 | 6 | 16 | 45 | 186 | 798 | 3860 | 19305 | 100235 |
| 4 | 2 | 9 | 32 | 160 | 942 | 6048 | 41144 | 290592 | 2119200 |
| 5 | 4 | 13 | 68 | 430 | 3328 | 27454 | 240448 | 2188095 | 20545320 |
| 6 | 3 | 18 | 116 | 955 | 9030 | 91770 | 982884 | 10942308 | 125656965 |
| 7 | 5 | 24 | 192 | 1860 | 20868 | 250446 | 3171084 | 41742027 | 566724020 |
| 8 | 4 | 31 | 288 | 3295 | 42628 | 591633 | 8645360 | 131347320 | 2058115980 |
| 9 | 6 | 39 | 420 | 5435 | 79794 | 1254589 | 20780280 | 357870942 | 6356282290 |
| 10 | 5 | 48 | 580 | 8480 | 139092 | 2446486 | 45294044 | 871916841 | 17310311600 |

More generally, let us consider the discrete gauging of any non-anomalous subgroup $\mathbb{Z}_r$, where $r$ is a divisor of $\gcd(N,K)$. Then the sum over the $\mathbb{Z}_r$ topological lines involves a sum over all divisors $d$ of $r$. Accordingly, the Witten index reads:

$$\mathbf{I}_W\big[(SU(N)/\mathbb{Z}_r)^0_K\big] = \frac{1}{r^2}\sum_{d|r}\mathscr{I}_3^{N,K}(d)\binom{\frac{K}{d}-1}{\frac{N}{d}-1}. \tag{224}$$

We again conjecture that this is an integer, as we also checked numerically for a large set of values of $N$, $K$ and $r$. Let us note that the normalisation $1/r^2$ in (224) is the 'maximal' allowed, *i.e.* 2 is the largest exponent of the order of the gauged subgroup that leads to an integral index. We emphasise that this is a rather strong consistency check – in general, none of the summands in (224) are integers, and only the whole sum has the right divisibility property. Note also that the sum (224) automatically 'skips' the would-be contribution from anomalous subgroups of $\mathbb{Z}_N$. Finally, we extend the conjecture (222) to the general case, as:

$$r\,|\,\gcd(N,K) \quad \Rightarrow \quad \mathbf{I}_W\big[(SU(N)/\mathbb{Z}_r)^0_K\big] \in \mathbb{N}. \tag{225}$$

A mathematical proof of this 'physical fact' would be extremely interesting.[29]

In summary, the expression (224) gives the (naive) Witten index for all possible 3d $\mathcal{N}=2$ Chern–Simons theories with gauge algebra $\mathfrak{su}(N)$, which are indexed by $N$, $K$ and $r$ such that $NK/r^2 \in \mathbb{Z}$ as discussed above (191). At fixed $N$ and $K$, the allowed integers $r$ run over the divisors of $\gcd(N,K)$.

**The (naive) Witten index with a non-trivial $\theta$-angle** The $(SU(N)/\mathbb{Z}_r)_K$ theory has a non-trivial $(\mathbb{Z}_r^{(0)})_{3d}$ 0-form symmetry, whose quantum numbers are the $\mathbb{Z}_r$-valued 'Stiefel–Whitney' classes that indicate whether a $SU(N)/\mathbb{Z}_r$ principal bundle can be lifted to an $SU(N)$ bundle – see *e.g.* [14, 15]. We can keep track of this symmetry in the Witten index by introducing a discrete fugacity for it. In the $A$-model formalism, this is precisely the $\theta_s$ index introduced

---

[29]A curiosity for $N$ prime concerns the fact that $\mathbf{I}_W[PSU(N)^0_K]$ for $N|K$ is divisible again by $N$, but only for $N \geq 5$. This is a consequence of Proposition 2 in Appendix E.1, which refines Proposition 1 by excluding the case $N = 3$. For this case, the integers $M_{N,K}$ defined in (206) are divisible again by $N$.

above, which at the same time serves as a background gauge field for the dual $(-1)$-form symmetry.

Let us first consider the $PSU(N)_K$ theory with $N|K$. Then, the sum over topological lines in (217) generalises to:

$$\mathbf{I}_W\left[PSU(N)_K^{\theta_s}\right] = Z_{T^3}\left[PSU(N)_K^{\theta_s}\right] = \frac{1}{N^2}\sum_{m,n,n'\in\mathbb{Z}_N} e^{2\pi i\frac{ms}{N}}\langle\Pi^{m\gamma_0}\mathcal{U}^{n\gamma_0}(\mathcal{C})\,\mathcal{U}^{n'\gamma_0}(\widetilde{\mathcal{C}})\rangle_{T^2}\,,\quad(226)$$

with $s\in\mathbb{Z}_N$. The oscillating phase makes the counting problem more involved. It turns out to probe additional arithmetic features of the theory such as the divisibility of $N$ by square numbers: if $N$ is *square-free*, all theories are in the same gauge 'orbit'.[30] If $N$ is not square-free, then $\mathbb{Z}_N$ is not a product of cyclic groups and there can be disjoint gauge orbits. Furthermore, these cases can feature mixed anomalies between gauged and residual symmetries [15, 16].

The sum (226) is most strategically analysed in three steps: $N$ square-free, $N$ arbitrary but with $K$ such that the gravitational anomaly (197) vanishes for all $r|N$, and finally the generic case with both $N$ and $K$ arbitrary. The even simpler case where $N$ is prime is discussed in the previous section, where the theta angle $\theta_s$ enters the result (205) depending only on whether $s$ is divisible by $N$ or not. In general, the nontrivial phases $e^{2\pi i\frac{ms}{N}}$ affect crucially the counting of signs that were important in the definition of the symbol $\mathscr{J}_3$ (219). Consequently, for generic values of $N$, we are looking for a generalisation of the symbol $\mathscr{J}_3$ depending on the value $s$ interacting with the mixed anomaly signs (218) in the geometric sums. It is rather elaborate to work out the explicit $s$-dependence of (226), and we discuss these subtleties for general $N$ and $K$ in appendix D.4. At the moment, we are only able to obtain a general result for the second step, that is, for any $N$ such that the gravitational anomaly (197) vanishes for *all* divisors $r$ of $N$. A practical way to guarantee this is to choose $K$ to be a square-free integer: if for some divisor $r$, $N$ is even, $\frac{N}{r}$ is odd and $\frac{K}{r}$ is even, then $r$ is automatically even and thus $K$ is divisible by $4 = 2^2$ and hence not square-free. This condition is not an 'if and only if' statement – for instance, $SU(4)_{12}$ has subgroups $\mathbb{Z}_r$ with $r = 1, 2, 4$, but all $(SU(4)/\mathbb{Z}_r)_{12-4}$ CS theories are bosonic (see (201)), while $K = 12$ is not square-free.

In order to state the result here, let us define for any integer $d$ the *radical* $\text{rad}(d)$ as the product of the distinct prime numbers dividing $d$.[31] Then we refine Jordan's totient $J_k(d)$ as [50]:[32]

$$J_k(d,s) \equiv d^k\delta_{s\bmod\frac{d}{\text{rad}(d)},0}\prod_{p|d}\left(\delta_{s\bmod p^{e_d(p)},0} - \frac{1}{p^k}\right),\qquad(227)$$

where, for any prime divisor $p$ of $d$, $e_d(p)$ is the maximal exponent of which $p$ appears in the prime factor decomposition of $d$.[33] By setting $s = 0$, we obtain back the ordinary Jordan totient (215). The number-theoretic interpretation is that $J_k(d,s)$ is the contribution of the partition of $N^k$ times the Kronecker delta $\delta_{s\bmod N,0}$ into a sum over divisors $d$ of any integer $N$:

$$\sum_{d|N}J_k(d,s) = N^k\delta_{s\bmod N,0}\,,\qquad(228)$$

generalising (216). It occurs due to the convoluted geometric series appearing in the sum over $m$ in (226). This function is suitable to calculate (226) for any value of $N$, as long as the gravitational anomaly (197) vanishes for all $r|N$ – in particular, for $K$ a square-free integer. In this case, we find:

$$\mathbf{I}_W\left[PSU(N)_K^{\theta_s}\right] = \frac{1}{N^2}\sum_{d|N}J_3(d,s)\binom{\frac{K}{d}-1}{\frac{N}{d}-1}\,.\qquad(229)$$

---

[30]Square-free means that no prime factor appears more than once in the prime factor decomposition.

[31]The radical is also known as largest square-free factor or square-free kernel.

[32]This refined Jordan's totient was defined in [50], where the notations are related as $J_{2g}(d,s) = \left\{\frac{s}{d}\right\}_g$.

[33]That is, we can write $d = \prod_{p|d}p^{e_d(p)}$.

Let us provide some evidence to this result. When $N$ is prime and $K$ divisible by $N$, then the sum collapses to the divisors $d = 1$ and $d = N$. We have $J_3(N, s) = N^3 \delta_{s \bmod N, 0} - 1$, while $J_3(1, s) = 1$. Combining both contributions exactly reproduces the previous result (205). Another check is the case $N = 2$ where we require that $K \in 2\mathbb{Z}$ is square-free and a multiple of 2, that is, $K \in 2 + 4\mathbb{Z}$. In that case, we calculate $\mathbf{I}_W[SO(3)_K^{\theta_s}] = \left(K - 2 + 8\,\delta_{s \bmod 2, 0}\right)/4$ from (229), which matches precisely with the $N = 2$ calculation (22).

This formula (229) does not hold for generic values of $K$. As we discuss briefly in appendix D.4, when $K$ is not square-free for instance, we obtain alternating geometric series rather than geometric series, which shifts $s$ inside the $\delta$-symbols (227) – a similar modification is necessary when summing over the gauge flux operators, see (152). We expect that, in the case of arbitrary $N$ and $K$, there exists a modification of (227) taking care of these special cases, which we will leave as a problem for future work.

The discrete gauging of a proper non-anomalous subgroup proceeds as explained above in the case $\theta_s = 0$. If the $SU(N)_K$ theory has a non-anomalous subgroup $\mathbb{Z}_r$, then we truncate the sum (229) at $d = r$, that is, we only sum over divisors of $r$. At the same time, the correct normalisation is $r^{-2}$ rather than $N^{-2}$, corresponding to the order of the $\mathbb{Z}_r$ group we discretely gauge, and we have:

$$\mathbf{I}_W\left[(SU(N)/\mathbb{Z}_r)_K^{\theta_s}\right] = \frac{1}{r^2} \sum_{d|r} J_3(d, s) \binom{\frac{K}{d} - 1}{\frac{N}{d} - 1}. \tag{230}$$

in the cases without modular anomaly. This is a proper Witten index in those cases.

## 3.5 Gauging the 3d one-form symmetry on $\Sigma_g \times S^1$

As a final application of our detailed study of the 3d $A$-model for the $SU(N)_K$ theory and of its higher-form symmetries, let us consider the gauging of the 3d one-form symmetry for the theory on $\Sigma_g \times S^1$. This gives us the topologically twisted index on $\Sigma_g$ for any allowed $(SU(N)/\mathbb{Z}_r)_K$ $\mathcal{N} = 2$ Chern–Simons theory. The twisted index is obtained by basic surgery operations on the Riemann surface, which for $g > 1$ includes the insertion of the handle-gluing operator $\mathcal{H}$. In contrast to the case where $N$ is prime discussed above, in general there are several fixed points, which makes the evaluation more intricate.

**Gauging the 2d 0-form symmetry**    Let us first consider the gauging of the non-anomalous 0-form symmetry $\mathbb{Z}_N^{(0)}$ for arbitrary genus $g$, following the discussion in section 2.4. Recall that we label the $2g$ cycles on $\Sigma_g$ by $\mathcal{C}_i$ and associate to them the topological operators $\mathcal{U}^{\gamma_i}(\mathcal{C}_i)$, whose product (93) we denote by $\mathcal{U}^{\gamma}$. Let us further label the $\mathbb{Z}_N$ elements by $\gamma_i = n_i \gamma_0$, and collect $\boldsymbol{n} = (n_1, \ldots, n_{2g})$. As described in (98), the insertion of $\mathcal{U}^{\gamma}$ on the surface $\Sigma_g$ amounts to summing over the Bethe vacua fixed by all elements $\gamma$ simultaneously, which using our analysis for $SU(N)_K$ can be written as:

$$\langle \mathcal{U}^{\gamma} \rangle_{\Sigma_g} = \sum_{\hat{u} \in \mathcal{S}_{\text{BE}}^{(\gcd(\boldsymbol{n})\gamma_0)}} \mathcal{H}(\hat{u})^{g-1}. \tag{231}$$

Then we gauge the discrete 0-form symmetry $\mathbb{Z}_N$ by summing over all inserted lines:

$$Z_{\Sigma_g \times S^1}\left[SU(N)_K/\mathbb{Z}_N^{(0)}\right] = \frac{1}{N^{2g-1}} \sum_{\boldsymbol{n} \in \mathbb{Z}_N^{2g}} \sum_{\hat{u} \in \mathcal{S}_{\text{BE}}^{(\gcd(\boldsymbol{n})\gamma_0)}} \mathcal{H}(\hat{u})^{g-1}, \tag{232}$$

with the normalisation (116) as before. Of course, this double sum can be drastically simplified by realising that the second sum depends only on the value of $\gcd(\boldsymbol{n})$. Similarly to

the genus 1 analysis above, we can enumerate the $N^{2g}$ numbers $\boldsymbol{n}$ with fixed gcd by the sum $N^{2g} = \sum_{d|N} J_{2g}(d)$ (see (216)). Then the sum over the $2g$ cycles collapses to a sum over divisors $d$ of $N$. Note that $\gcd(\boldsymbol{n})\gamma_0$ generates the same subgroup as $\gcd(\boldsymbol{n}, N)\gamma_0$, which due to (174) is $\mathbb{Z}_d$ with $d = N/\gcd(\boldsymbol{n}, N)$. Thus, for each divisor $d$, the set of fixed vacua is simply $\mathcal{S}_{\mathrm{BE}}^{(\frac{N}{d}\gamma_0)}$, which one may also denote by $\mathcal{S}_{\mathrm{BE}}^{\mathbb{Z}_d}$. We therefore obtain:

$$Z_{\Sigma_g \times S^1}\left[SU(N)_K/\mathbb{Z}_N^{(0)}\right] = \frac{1}{N^{2g-1}} \sum_{d|N} J_{2g}(d) \sum_{\hat{u} \in \mathcal{S}_{\mathrm{BE}}^{\mathbb{Z}_d}} \mathcal{H}(\hat{u})^{g-1}. \tag{233}$$

We checked for small values of $g$, $N$ and $K$ that (233) leads to integer partition functions, with precisely this normalisation.[34]

**Gauging the 2d 1-form symmetry**   The gauging of the 2d 1-form symmetry $\mathbb{Z}_N^{(1)}$ has been discussed in the general case in 3.1.1. While for $g = 1$ the insertion of a single line in $T^3$ enjoys full 3d modularity and we were able to relate the 1-form gauging to an enumeration of fixed points, for higher genus this is not possible. Instead, it is still the case that gauging $\mathbb{Z}_N^{(1)}$ with a fixed $\theta_s$ projects us onto one of the disjoint universes of decomposition, giving us (155).

**Gauging the full 3d 1-form symmetry**   The full 3d 1-form symmetry can be gauged by summing over all insertions $\Pi^\delta \mathcal{U}^\gamma$ on $\Sigma_g \times S^1$, as described in section 2.5. This amounts to combining the expressions (233) with (155) in a suitable way: Since the insertion of the 0-form operators $\mathcal{U}^\gamma$ project onto the fixed points, we may simply evaluate the sum over the gauge flux operators $\Pi^\delta$ on those fixed points. This immediately gives the $\Sigma_g$ twisted index:

$$Z_{\Sigma_g \times S^1}\left[PSU(N)_K^{\theta_s}\right] = \frac{1}{N^{2g-1}} \sum_{d|N} J_{2g}(d) \sum_{\hat{u} \in \mathcal{S}_{\mathrm{BE}}^{\vartheta_s, \mathbb{Z}_d}} \mathcal{H}(\hat{u})^{g-1}, \tag{234}$$

where the second sum is over the Bethe vacua in the universe set by $\theta_s$ which are invariant under the action of $\mathbb{Z}_d$:

$$\mathcal{S}_{\mathrm{BE}}^{\vartheta_s, \mathbb{Z}_d} \equiv \mathcal{S}_{\mathrm{BE}}^{\vartheta_s} \cap \mathcal{S}_{\mathrm{BE}}^{\mathbb{Z}_d}. \tag{235}$$

By direct computation, we checked that this formula returns an integer for small values of the parameters. The expression (234) generalises known expressions in the mathematical literature, where Verlinde dimensions for $PSU(N)_K$ with $N$ odd were already written in terms of the genus-$g$ generalisation of Jordan's totient [50]. In the mathematical setup, Jordan's totient $J_{2g}(d)$ has a geometric meaning as the number of elements of order $d$ in the Jacobian variety $J(\Sigma_g)$ associated with the Riemann surface [50].[35] Finally, the result (234) appears to agree with some explicit results of [19] as expected, as well as with the mathematical framework for non-simply connected groups [36, 54]. To the best of our knowledge, the result (234) has not been obtained before in the mathematical literature for $N$ and $K/N$ both even, essentially due to the difficulties related to the gravity-$(\mathbb{Z}_N^{(1)})_{3d}$ anomaly discussed in section 3.2. It would be interesting to give a firm mathematical footing to the formula (234) in this case.

It is rather difficult to obtain more explicit (*i.e.* more efficiently computable) results for genus-$g$ twisted indices for all values of $g$. One simple result is:

$$Z_{\Sigma_g \times S^1}\left[PSU(N)_N^0\right] = N, \tag{236}$$

---

[34]One may check as well that this the 'maximal' normalisation, meaning that if we divide by $N$ another time (*i.e.* $N^{2g}$ in the denominator) we do note get integers, in general.

[35]This Jacobian is the variety that carries the Verlinde bundles.

generalising (209) to non-prime $N$. One can further check that, for $g = 1$ and $s = 0$, the result (234) is compatible with (221). This identification expresses $\mathscr{J}_3(d)$ as a combination of $N$, $J_2(d)$ and the Kronecker-delta $\Delta^N_{\ell(\hat{u})}$ over the fixed Bethe vacua, as:

$$NJ_2(d)\left|\mathcal{S}^{\vartheta_{s=0},\mathbb{Z}_d}_{\mathrm{BE}}\right| = \mathscr{J}_3(d)\left|\mathcal{S}^{\mathbb{Z}_d}_{\mathrm{BE}}\right|. \tag{237}$$

This relation characterises the distribution of values $\ell(\hat{u})$ among the vacua $\hat{u}$ fixed under a $\mathbb{Z}_d$ subgroup. We checked it explicitly in examples for small values of $N$ and $K$, but we leave a proof for future work. Another check is the comparison to $N \geq 3$ prime and $N|K$. In that case, the only divisors are $d = 1$ and $d = N$. Using $J_{2g}(N) = N^{2g} - 1$ and (172), a simple calculation derives the previous result (208) from (234).

**The $\Sigma_g$ twisted index for $(SU(N)/\mathbb{Z}_r)_K$**   As anticipated from the presentation (234) of the $PSU(N)_K$ twisted index, it is straightforward to consider gauging only a subgroup $\mathbb{Z}_r$ rather than the full $\mathbb{Z}_N$, and we can also gauge any non-anomalous $(\mathbb{Z}^{(1)}_r)_{3d}$ even if $(\mathbb{Z}^{(1)}_N)_{3d}$ has a 't Hooft anomaly. The twisted index of the $(SU(N)/\mathbb{Z}_r)_K$ theory is simply obtained by summing only over the topological lines operators for the $\mathbb{Z}_r$ subgroup. Following the logic from the previous subsection, we simply restrict the divisor sum to divisors of $r$ only:

$$Z_{\Sigma_g \times S^1}\left[(SU(N)/\mathbb{Z}_r)^{\theta_s}_K\right] = \frac{1}{r^{2g-1}}\sum_{d|r}J_{2g}(d)\sum_{\hat{u}\in\mathcal{S}^{\mathbb{Z}_d}_{\mathrm{BE}}}\Delta^{N,r}_\ell(s)\mathcal{H}(\hat{u})^{g-1}$$

$$= \frac{1}{r^{2g-1}}\sum_{d|r}J_{2g}(d)\sum_{\hat{u}\in\mathcal{S}^{\vartheta^{(\mathbb{Z}_r)}_s,\mathbb{Z}_d}_{\mathrm{BE}}}\mathcal{H}(\hat{u})^{g-1}, \tag{238}$$

where $\Delta^{N,r}_\ell(s)$ as defined in (160), and $\mathcal{S}^{\vartheta^{(\mathbb{Z}_r)}_s,\mathbb{Z}_d}_{\mathrm{BE}} \equiv \mathcal{S}^{\mathbb{Z}_d}_{\mathrm{BE}} \cap \mathcal{S}^{\vartheta^{(\mathbb{Z}_r)}_s}_{\mathrm{BE}}$ similarly to (235). This final result includes as special cases all previous results – the $PSU(N)_K$ Witten index (221) and more generally the $(SU(N)/\mathbb{Z}_r)_K$ Witten index (224), as well as the $\Sigma_g$ twisted index (208) for $PSU(N)_K$ with $N$ prime.

# 4   Further aspects of the 3d $\mathcal{N} = 2$ $SU(N)_K$ CS theory

In this section, we study some further aspects of the $\mathcal{N} = 2$ supersymmetric $SU(N)_K$ Chern–Simons theory, which provides us with interesting consistency checks of our results above. In section 4.1, we study the gauging of the 3d 1-form symmetry using the fusion rules of the associated 3d TQFT, and find a precise (and non-trivial) agreement with our $A$-model result for the Witten index. In section 4.2, we briefly comment on the level/rank duality for the $\mathcal{N} = 2$ $SU(N)_K$ Chern–Simons theory, and on how the higher-form symmetries should match across the duality.

## 4.1   Gauging the 1-form symmetry in the 3d TQFT

Extending our discussion of the $SU(2)$ example from section 1.1 in the introduction, we can check some of our results for the $SU(N)_K$ theory using the purely 3d description of the underlying 3d TQFT. Indeed, we can consider gauging the 1-form symmetry $(\mathbb{Z}^{(1)}_N)_{3d}$ by the condensation of the abelian anyons for $\mathbb{Z}_N$. Here we follow closely the approach of [42].

Let us first review the fusion rules of the Wilson lines in the $SU(N)_K$ CS theory, which are the same as the fusion rules of chiral primaries in the corresponding 2d WZW model – see *e.g.* [41, 96–102]. They are best expressed in terms of Young diagrams. First, recall that the

$SU(N)_K$ $\mathcal{N} = 2$ supersymmetric CS theory has a spectrum of Wilson lines $W_{\lambda}$ indexed by the Young tableaux:

$$\lambda = [\lambda_1, \ldots, \lambda_{N-1}], \qquad \lambda_i \leq K - N, \ \forall i. \tag{239}$$

The generator of the 3d $\mathbb{Z}_N$ one-form symmetry is simply:

$$a = [K - N, 0, \ldots, 0], \tag{240}$$

where we used the obvious notation $W_{\lambda} = \lambda$. Linking any line with $a$, we pick up a phase:

$$W_{\lambda} \to e^{\frac{2\pi i n(\lambda)}{N}} W_{\lambda}, \qquad n(\lambda) \equiv \sum_{i=1}^{N-1} \lambda_i \bmod N, \tag{241}$$

where $n(\lambda)$ is the $N$-ality of the representation (the number of boxes in the Young tableau mod $N$). The fusion of any Wilson line with the abelian anyon $a$ is easily found to be:

$$a \times [\lambda_1, \lambda_2, \ldots, \lambda_{N-1}] = [K - N - \lambda_{N-1}, \lambda_1 - \lambda_{N-1}, \lambda_2 - \lambda_{N-1}, \ldots, \lambda_{N-2} - \lambda_{N-1}]. \tag{242}$$

It corresponds to adding the row $a$ on top of the Young tableau $\lambda$, and subsequently removing $\lambda_{N-1}$ columns on the left. As an example, consider the fusion of a Wilson line in the $SU(5)_{10}$ theory:

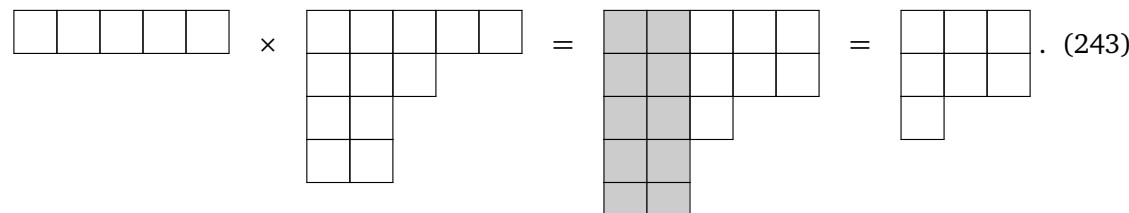

$$\tag{243}$$

Given these rules, we can carry out the three-step gauging procedure [42] at the level of the 3d line spectrum. Before studying the case where $N$ is prime (and then for $N$ arbitrary), let us demonstrate this procedure in a simple example.

**Example: $SU(3)_6$** The $SU(3)_6$ theory has 10 lines:

$$\lambda = [0, 0], [1, 0], [2, 0], [3, 0], [1, 1], [2, 1], [2, 2], [3, 1], [3, 2], [3, 3]. \tag{244}$$

Out of these, we have the $\mathbb{Z}_3$ lines:

$$[0, 0] = \mathbf{1}, \qquad [3, 0] = a, \qquad [3, 3] = a^2. \tag{245}$$

Keeping only the $\mathbb{Z}_3$-neutral lines (under linking), we are left with the four lines:

$$\mathbf{1}, \qquad a, \qquad W_{\text{adj}} \equiv [2, 1], \qquad a^2. \tag{246}$$

Next, we should identify lines related by the fusion (242). Note that $aW_{\text{adj}} = W_{\text{adj}}$, so it is left invariant under fusion as well. We then discard the non-trivial $\mathbb{Z}_3$ lines, but we need to keep three copies of $W_{\text{adj}}$. We then find the $PSU(3)_6$ lines:

$$\mathbf{1}, \qquad W_{\text{adj},1}, \qquad W_{\text{adj},2}, \qquad W_{\text{adj},3}. \tag{247}$$

This is a total of four lines, which of course agrees with our $A$-model computation (see *e.g.* table 1).

**$N$ prime**   Consider now the case where $N$ is an arbitrary prime number, and $K$ a multiple of $N$. Out of the $\binom{K-1}{N-1}$ Young tableaux $\boldsymbol{\lambda}$, as the first step we keep only the $\mathbb{Z}_N$-neutral lines, that is, the ones with $n(\boldsymbol{\lambda}) \equiv 0 \mod N$. In order to enumerate them, we use again the decomposition (206) of the $SU(N)_K$ index into the sum of 1 and $M_{N,K}N^2$. In the 3d prescription, the 1 has an important meaning: It is the unique neutral Wilson line which is left invariant under fusion with the abelian anyon. We can find a general form for this invariant Wilson line:

$$\boldsymbol{\lambda}_{\text{inv}} = \left(\tfrac{K}{N} - 1\right)[N-1, N-2, \ldots, 2, 1], \tag{248}$$

where the prefactor multiplies all number of boxes. It is clear that the fusion (242) with $a$ (240) leaves it invariant: The Young tableau $a$ has $\frac{K}{N} - 1$ boxes more than the first row of $\boldsymbol{\lambda}_{\text{inv}}$, whose last row has $\frac{K}{N} - 1$ boxes as well. Removing $\frac{K}{N} - 1$ columns on the left results in the same Young diagram $\boldsymbol{\lambda}_{\text{inv}}$.

The $N$-ality of (248) is $n(\boldsymbol{\lambda}_{\text{inv}}) = \left(\frac{K}{N} - 1\right)\frac{N-1}{2}N$, which is divisible by $N$ since $N \geq 3$ is prime and thus odd. Using (Captain) hook's formula, we find that it has dimension:

$$\dim \boldsymbol{\lambda}_{\text{inv}} = \left(\frac{K}{N}\right)^{K-N}, \tag{249}$$

and that it is self-dual, and thus a real representation of $SU(N)$ (see *e.g.* [103]).[36] For instance, the invariant Wilson line for $SU(5)_{15}$ is:

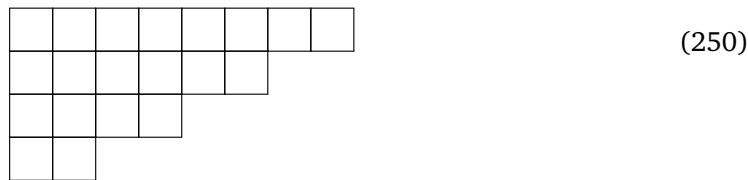

$$\tag{250}$$

This allows us to correctly enumerate the remaining lines in each step of the 3d gauging procedure [42]. In step 1, we discard the lines that are not neutral under $\mathbb{Z}_N$. The line $\boldsymbol{\lambda}_{\text{inv}}$ is neutral, while from the other $M_{N,K}N^2$ lines only every $N$-th line is neutral. This gives us:

$$\text{Step 1:} \quad 1 + M_{N,K}N^2 \longrightarrow 1 + M_{N,K}N, \tag{251}$$

for the number of invariant lines. In step 2, we identify all lines $W$ and $aW$ obtained by fusing with $a$. While $W_{\boldsymbol{\lambda}_{\text{inv}}}$ is invariant, this partitions the remaining $M_{N,K}N$ lines into $M_{N,K}$ orbits under the $a$-fusion of orbit length $N$ each. It is clear that this has to be the case, since the fusion with the abelian anyon constitutes a group action of $\mathbb{Z}_N$ on the collection of Wilson lines, and, since $\mathbb{Z}_N$ has no subgroups, there cannot be shorter orbits. This leaves us with:

$$\text{Step 2:} \quad 1 + M_{N,K}N \longrightarrow 1 + M_{N,K}, \tag{252}$$

for the number of lines. Finally, at step 3, for the fixed point $W_{\boldsymbol{\lambda}_{\text{inv}}}$ under the fusion we create $N$ copies. We thus obtain:

$$\text{Step 3:} \quad 1 + M_{N,K} \longrightarrow N + M_{N,K}, \tag{253}$$

for the number of Wilson lines, precisely agreeing with the earlier result (207) for the $PSU(N)_K$ Witten index with vanishing theta angle.

---

[36]This representation is the adjoint only for $N = 3$ at level $K = 6$.

**General $N$**   When $N$ is not prime, the only modification to the gauging procedure is the third step. Note that when $N$ is prime, then $a^n$ is a generator of the 3d 1-form symmetry for any positive integer $n$ with $n \pmod N \neq 0$. If $N$ has a divisor $d$, then $a^n$ can generate a proper $\mathbb{Z}_d$ subgroup of $\mathbb{Z}_N$ for specific values of $n$. Consequently, if $n$ is the smallest divisor of $N$ such that a line $W$ is invariant under the fusion with $a^n$, then we generate $N/n$ copies of $W$ in the third step [42].

The gauging procedure then proceeds almost as before. We start with the spectrum $\{W_\lambda\}$ of Wilson lines (239). Discarding the lines that are not invariant under the $\mathbb{Z}_N$ 1-form symmetry leaves us with a smaller set $\{W_\lambda\}^{\mathbb{Z}_N} = \{W_\lambda : n(\lambda) \equiv 0 \mod N\}$. The fusion with $a$ furnishes a partition into orbits:

$$\{W_\lambda\}^{\mathbb{Z}_N} = \bigcup_{\lambda'} \mathrm{Orb}(W_{\lambda'}). \tag{254}$$

Since $a$ constitutes a $\mathbb{Z}_N$ group action, the lengths $|\mathrm{Orb}(W_{\lambda'})|$ of the orbits are divisors of $N$. Any line in an orbit $\mathrm{Orb}(W_{\lambda'})$ is then invariant under the fusion with $a^m$, where $m = |\mathrm{Orb}(W_{\lambda'})|$. Consequently, for each orbit $\mathrm{Orb}(W_{\lambda'})$ we keep $N/|\mathrm{Orb}(W_{\lambda'})|$ copies. This gives us the $PSU(N)_K$ spectrum of lines:

$$\{W_\lambda\}/\mathbb{Z}_N^{(1)} = \bigcup_{\lambda'} \bigcup_{j=1}^{N/|\mathrm{Orb}(W_{\lambda'})|} W_{\lambda',j}, \tag{255}$$

from which we easily calculate

$$\mathbf{I}_W[PSU(N)_K] = \sum_{\lambda'} \frac{N}{|\mathrm{Orb}(W_{\lambda'})|}. \tag{256}$$

This matches with the case $N$ prime, where there is one orbit of length 1 and $M_{N,K}$ orbits of length $N$, giving back (253). For given arbitrary values of $N$ and $K$, it is straightforward to implement the fusion (242) on a computer, and we find that (256) agrees precisely with the general-$N$ result (221) for all $N \leq K \leq 19$. This provides us with another consistency check of our 3d $A$-model calculations. Since (256) does not explicitly take into account the $T^3$ modular anomaly discussed in section 3.2, the above 3d calculation provides us with some independent evidence for the correctness of the analysis that lead to the general formula (221). As mentioned in the previous section, the quantity (256) is only a 'naive' Witten index, but it gives the correct Witten index whenever the $PSU(N)_{k=K-N}$ pure CS theory is bosonic. When it is a spin-TQFT instead, the naive index still correctly counts the lines (in some sector to be specified), but it is not a supersymmetric index [21].

## 4.2   Comments on level/rank duality

It is also instructive to consider level/rank duality for our 3d $\mathcal{N} = 2$ supersymmetric CS theory [76, 77, 104–106]:

$$SU(N)_K \qquad \longleftrightarrow \qquad U(K-N)_{-K,-N}. \tag{257}$$

The underlying 3d field theory has a $\mathbb{Z}_N^{(1)}$ 1-form symmetry, but it is realised differently on either side – for instance, the fundamental Wilson line of $SU(N)$ maps to a Wilson line in the determinant representation of $U(K-N)$. These lines generate the 3d 1-form symmetry in each description. In the following, we make some preliminary comments on the duality (257) from the perspective of their respective 3d $A$-models. We comment on the matching of Bethe vacua between the two sides, but unfortunately we did not find an explicit duality map between the two descriptions; this is left as a challenge for future work.

**Bethe vacua**   Let us look at the vacua of the $U(K-N)_{-K,-N}$ theory, assuming $K > N > 0$. The Bethe equations read:

$$q(-x_a)^{-K} \det x = 1\,, \qquad a = 1,\ldots,K-N\,, \tag{258}$$

and here $q = e^{2\pi i \tau}$ is the complexified FI term for the $U(K-N)$ gauge group. The level/rank duality (257) is a little bit subtle (in particular due to $U(K-N)_{-K,-N}$ being spin if $N$ is odd [76]), and here we should really set $q = 1$. So, for simplicity, let us just assume the Bethe equations are:[37]

$$x_a^K = \det x\,, \qquad \forall a\,. \tag{259}$$

We can parameterise the solutions as:

$$\hat{u}_a = \frac{j_a}{KN}\,, \qquad (j_a) \subset \{0,1,\ldots,KN-1\}\,, \tag{260}$$

where $(j_a)$ is an ordered subset which satisfies the conditions (259), namely:

$$\sum_{b=1}^{K-N} j_b \equiv K j_a \mod KN\,, \tag{261}$$

for all $a = 1,\ldots,K-N$. As an example, consider $K = 4$ and $N = 2$. Then the $U(2)_{-4,-2}$ theory has 3 vacua with:

$$(j_a) = \{(2,6),\,(1,3),\,(5,7)\}\,. \tag{262}$$

Of course, this matches the expected number of vacua of $SU(2)_4$. It turns out that the first vacuum is the one fixed under the action of $\mathbb{Z}_2^{(0)}$, as we will discuss below.

**Witten index**   Let us count the Bethe vacua, *i.e.* solutions to (261), more systematically. Given any tuple $(j_1,\ldots,j_{K-N})$, denote their sum by $\sigma = \sum_{a=1}^{K-N} j_a$. Then from (261) we have that $\sigma$ is divisible by $K$. That is, we can write $\sigma = kK$ with some integer $k$. From (261), we find that $k \equiv j_a \mod N$ for each $a = 1,\ldots,K-N$. This in particular implies that the values of $l_a$ differ by multiples of $N$. Therefore, for any given $K-N$-tuple $(j_a)$, we can find a unique number $\tilde{k} = 0,\ldots,N-1$ such that $j_a \equiv \tilde{k} \mod N$ for all $a$, where $\tilde{k}$ is simply $k$ reduced modulo $N$.

Given any solution $(j_a)$ with a particular value of $\tilde{k}$, we can always construct a new solution $(j'_a)$ whose value is $\tilde{k} + 1$. This implies in particular that, for each value of $\tilde{k}$, there are the same number of solutions to (261). Since there are $N$ possible values of $\tilde{k}$, let us therefore study the case where $\tilde{k} = 0$.

For $\tilde{k} = 0$, we are then looking at tuples $(j_a)$ such that $j_a \equiv 0 \mod N$. Since the domain for the solutions is $j_a = 0,\ldots,KN-1$, this restricts the solution space to the values $j_a = 0,N,2N,\ldots,(K-1)N$. These are $K$ numbers, and we are selecting $K-N$ numbers from those $K$ numbers, for which there are $\binom{K}{K-N}$ possibilities. These candidate solutions are solutions of (261) if $\sigma \equiv 0 \mod KN$. Since each $j_a$ is divisible by $N$, it follows that only every $K$-th tuple satisfies this condition. Hence, the number solutions to the Bethe equations with $\tilde{k} = 0$ is $\frac{1}{K}\binom{K}{K-N}$. Since we have $N$ set of solutions of equal size for every $\tilde{k}$, as explained above, the total value for the Witten index is:

$$\mathbf{I}_W\big[U(K-N)_{-K,-N}\big] = \frac{N}{K}\binom{K}{K-N} = \binom{K-1}{N-1}\,. \tag{263}$$

As anticipated, this matches precisely with the Witten index (136) of the $SU(N)_K$ theory.

---

[37]That is, we assume $K$ even for simplicity. We can of course keep track of the signs for $K$ odd, but for our purposes here this will be immaterial.

This result suggests ideas to find an explicit isomorphism between the Bethe equations of the $SU(N)_K$ theory and that of the $U(K-N)_{-K,-N}$ theory. From (263) we see that the level/rank duality can be naturally formulated as complements of a set [7–9, 107]: while in the $SU(N)_K$ theory we consider selecting $N$ elements from a set with $K$ elements, in the $U(K-N)$ theory it is natural to select rather $K-N$ elements. These subsets are precisely complements of each other. It is still non-trivial to find the set $\{j_a\}$ labelling a vacuum of the $U(K-N)_{-K,-N}$ theory that corresponds to a specific $\underline{l} = (l_a; \ell)$ in the $SU(N)_K$ theory. We leave this important question as a challenge for future work.

**0-form and 1-form symmetry**    The action of $\mathbb{Z}_N^{(0)}$ on the Bethe vacua simply corresponds to $\hat{x}_a \mapsto e^{\frac{2\pi i}{N}} \hat{x}_a$, or $\hat{u}_a \to \hat{u}_a + \frac{1}{N}$ for all $a$. In other words, this acts as

$$\hat{u} \mapsto \hat{u} + \gamma : \qquad j_a \mapsto j_a + K \bmod KN \,, \tag{264}$$

up to a permutation of the $j_a$'s, which is the Weyl symmetry $S_{K-N}$. By the level-rank duality (257), this 0-form action is supposed to be reflected in the $SU(N)_K$ theory with the same $N$ and $K$. Indeed, from (133) and (166), this $\mathbb{Z}_N^{(0)}$ symmetry acts on the $SU(N)$ on-shell variables $\hat{x}_a$ as $\hat{x}_a \mapsto e^{\frac{2\pi i}{N}} \hat{x}_a$ as well.

The generator of the 1-form symmetry $\mathbb{Z}_N^{(1)}$ in the $A$-model for the $U(K-N)_{-K,-N}$ theory is the flux operator:

$$\Pi^{\gamma_0} = (\det x)^{-1} \,. \tag{265}$$

Comparing with the action of the 1-form symmetry operator (146) in the $SU(N)_K$ theory, we find the relation:

$$\frac{1}{K} \sum_b j_b = \ell + K \quad \bmod N \,. \tag{266}$$

Here, $\ell$ is defined for each $SU(N)_K$ Bethe vacuum as in (134), and it is essentially $\sum_a l_a$. From (261), it is clear that $\sum_b j_b$ is divisible by $K$, and hence the left-hand-side is indeed an integer. In some cases, (266) gives a unique relation between the $SU(N)_K$ vacua and the $U(K-N)_{-K,-N}$ vacua.

**IR duality**    Any infrared IR duality between two 3d $\mathcal{N} = 2$ theories $\mathcal{T}$ and $\mathcal{T}'$ induces an isomorphism between the ground-state Hilbert spaces:

$$\mathscr{D} : \mathcal{H}_{S^1}[\mathcal{T}] \longrightarrow \mathcal{H}_{S^1}[\mathcal{T}'], \tag{267}$$

which is a unitary transformation that commutes with all symmetry actions, including higher-form symmetries:

$$\mathscr{D}^\dagger \mathcal{U}_\mathcal{T} \mathscr{D} = \mathcal{U}_\mathcal{T} \,. \tag{268}$$

Here, $\mathcal{U}_\mathcal{T}$ represents the symmetry operators $\mathcal{U}^\gamma$ and $\Pi^\gamma$ in the $A$-model for the theory $\mathcal{T}$, for some symmetry group $\Gamma$. Then, a particular consequence of the duality is that all correlation functions of the symmetry operators necessarily agree, and therefore the allowed gauging operations for discrete symmetries will always give rise to dual theories $\mathcal{T}/\Gamma$ and $\mathcal{T}'/\Gamma$. For the level/rank duality at hand, we hope to construct the explicit isomorphism (267) in later work.

# 5    Including matter: $U(1)$ and $SU(N)$ theories

So far, we have been discussing higher-form symmetries for pure $(S)U(N)$ gauge theories. In this section, we couple the 3d vector multiplets to matter in chiral multiplets. These theories retain a one-form symmetry $\Gamma$ if the matter fields preserve some subgroup of the centre

symmetry of the pure gauge theory:

$$\Gamma \subseteq Z(\widetilde{G}). \tag{269}$$

We will first consider the very simple case of an abelian gauge theory with matter fields of non-minimal electric charge. Next, for definiteness, we will consider the $SU(N)_k$ Chern–Simons-matter theory with $n_{\text{adj}}$ chiral multiplets in the adjoint representation. We will only provide preliminary comments. More explicit computations of twisted indices for these theories could be performed *e.g.* using the computational algebraic geometric methods discussed in [108]. This is left as another challenge for future work.

## 5.1 $U(1)_k$ with matter

Consider a $U(1)_k$ vector multiplet coupled to chiral multiplets $\Phi_i$ with electric charge $Q_i \in \mathbb{Z}$. The UV effective CS level $k$ is related to the bare CS level $K \in \mathbb{Z}$ as:

$$K = k + \frac{1}{2} \sum_i Q_i^2. \tag{270}$$

Let us also denote by $Q_i = (Q_{i_+}, Q_{i_-})$ the positive and negative charges, respectively. The effective twisted superpotential reads:

$$\mathcal{W}(u, v) = \tau u + \frac{K}{2}(u^2 + u) + \frac{1}{(2\pi i)^2} \sum_i \text{Li}_2\big(e^{2\pi i Q_i u} y_i\big), \tag{271}$$

where $\tau$ is the complexified FI parameter and $y_i$ are flavour fugacities. We thus have the Bethe equation:

$$\Pi(u, v) \equiv q(-x)^K \prod_i \left(\frac{1}{1 - x^{Q_i} y_i}\right)^{Q_i} = 1, \tag{272}$$

with $q = e^{2\pi i \tau}$. The Witten index of this theory is simply [109]:

$$\mathbf{I}_W = |\mathcal{S}_{\text{BE}}| = \max\left(|K| + \sum_{i_{-\epsilon}} Q_{i_{-\epsilon}}^2, \sum_{i_\epsilon} Q_{i_\epsilon}^2\right), \qquad \epsilon \equiv \text{sign}(K). \tag{273}$$

This theory has a one-form symmetry:

$$\Gamma_{3d}^{(1)} \cong \mathbb{Z}_M, \qquad M \equiv \gcd(K, Q_i). \tag{274}$$

This act on the $u$ variable as:

$$u \mapsto u + \gamma_0, \qquad \gamma_0 \equiv \frac{1}{M}. \tag{275}$$

Indeed, the gauge flux operator $\Pi$ is invariant under such a shift, and it admits a $M$-th root:

$$\Pi^{\gamma_0} \equiv \Pi^{\frac{1}{M}}. \tag{276}$$

It is also easy to check that the one-form symmetry has a 't Hooft anomaly:

$$\mathcal{A}\big(\gamma_{(0)}, \gamma_{(1)}\big) = nm \frac{K}{M^2}, \qquad \textit{i.e.} \quad \mathfrak{a} = \frac{K}{M} \bmod M, \tag{277}$$

where we have $\gamma_{(1)} = m\gamma_0$ and $\gamma_{(0)} = n\gamma_0$.

Whatever the anomaly, the orbit structure of the Bethe vacua under $\mathbb{Z}_M^{(0)}$ is trivial, as all orbits are of dimension $M$. Indeed, we can solve the Bethe equation (272) by writing $x = e^{\frac{2\pi i \ell}{M}} z^{\frac{1}{M}}$ and solve for $z$. Then, for each solution $z = \hat{z}$, we will have $M$ Bethe roots:

$$\hat{x}_\ell(\hat{z}) = e^{\frac{2\pi i \ell}{M}} \hat{z}, \qquad \ell \in \mathbb{Z}_M, \tag{278}$$

which are permuted by $\mathbb{Z}_M^{(0)}$:

$$\mathcal{U}^{n\gamma_0} \; : \; |\hat{u}_\ell + n\gamma_0\rangle \mapsto |\hat{u}_{\ell+n}\rangle\,. \tag{279}$$

Moreover, all insertions of topological lines are trivial:

$$\langle \mathcal{U}^{n\gamma_0}(\mathcal{C})\rangle_{\Sigma_g} = \delta_{n,0}\langle 1\rangle_{\Sigma_g}\,, \qquad \langle \Pi^{m\gamma_0}\rangle_{\Sigma_g} = \delta_{m,0}\langle 1\rangle_{\Sigma_g}\,, \tag{280}$$

as one can readily check. If $K = 0 \bmod M$ the 't Hooft anomaly vanishes, and we can then gauge the full $(\mathbb{Z}_M^{(1)})_{3d}$, yet due to (280) this has only the trivial effect of dividing the twisted index by $M^{2g}$. The interpretation is simply that, in the absence of the anomaly, we can rescale the vector multiplet by $1/M$, which leads to a $U(1)$ theory with minimal charges $Q_i/M$.

**The pure $U(1)_K$ theory**   In the special case of the pure $U(1)\,\mathcal{N}=2$ CS theory without matter, we have $M = K$. Thus we have the Bethe equation and Bethe roots:

$$\Pi(u) = q(-x)^K\,, \qquad \hat{u}_\ell = \frac{\ell - \tau}{K} + \frac{1}{2} \bmod 1\,, \quad \ell \in \mathbb{Z}_K\,, \tag{281}$$

where the $K$ vacua are permuted by $\mathbb{Z}_K^{(0)}$. The handle-gluing operator is $\mathcal{H} = K$, and hence $Z_{\Sigma_g \times S^1} = K^g$ for this theory. The anomaly is $\mathfrak{a} = 1 \bmod K$ and it is maximal, in the sense of section 2.1.2. Nonetheless, we can still gauge a subgroup $\mathbb{Z}_r \subset \mathbb{Z}_K$ in 3d if $r^2|K$. In such a case, since all non-trivial insertions of symmetry operators vanish, the $\mathbb{Z}_r^{(1)} \oplus \mathbb{Z}_r^{(0)}$ gauging trivially gives:

$$Z_{\Sigma_g \times S^1}\left[U(1)_K/(\mathbb{Z}_M^{(1)})_{3d}\right] = \frac{1}{r^2} Z_{\Sigma_g \times S^1}[U(1)_K] = \left(\frac{K}{r^2}\right)^g\,. \tag{282}$$

This is interpreted as a rescaling of the vector multiplet by $1/r$, which gives us a well-defined $U(1)_{\frac{K}{r^2}}$ CS theory precisely if $r^2|K$.

## 5.2   $SU(N)_k$ with adjoint matter

Let us discuss some general aspects of the $SU(N)_k\,\mathcal{N}=2$ CS theory coupled with $n_{\text{adj}}$ chiral multiplets in the adjoint representation of the gauge group. For $n_{\text{adj}} = 1$, this theory allows one to compute the so-called equivariant Verlinde formula [56] for $\widetilde{G} = SU(N)$, which was extended to $G = \widetilde{G}/\Gamma$ in [19].

Here, the integer $k$ is the UV effective CS level. In our conventions (see *e.g.* [7]), it is related to the bare CS level $K$ that appear in the twisted superpotential as:

$$K = k + n_{\text{adj}}N\,. \tag{283}$$

The effective twisted superpotential of this theory is given by:

$$\mathcal{W}(u,v) = \frac{K}{2}\sum_{a=1}^{N} u_a^2 + u_0 \sum_{a=1}^{N} u_a + \frac{1}{(2\pi i)^2}\sum_{i=1}^{n_{\text{adj}}}\sum_{\substack{a,b=1\\a\neq b}}^{N} \text{Li}_2\left(e^{2\pi i(u_a-u_b)}y_i\right)\,, \tag{284}$$

where the first two contributions are the same as in (129), including the Lagrange multiplier $u_0$ that imposes tracelessness, while the dilogarithms are the contributions from the adjoint chiral multiplets, with $v_i$ their twisted masses and $y_i \equiv e^{2\pi i v_i}$. Then, the Bethe equations read:

$$\Pi_a(u,v) \equiv q x_a^k \prod_{i=1}^{n_{\text{adj}}} \prod_{\substack{b=1\\b\neq a}}^{N} \frac{x_a - y_i x_b}{x_b - y_i x_a} = 1\,, \qquad a = 1,\dots,N\,,$$

$$\Pi_0(u,v) \equiv \prod_{a=1}^{N} x_a = 1\,, \tag{285}$$

where we defined $q = e^{2\pi i u_0}$ exactly as in section 3.1.

**Eigenvalues of the *A*-operators**   Since the adjoint matter multiplets are not charged under the centre $\mathbb{Z}_N \subset SU(N)$, we retain the full 3d 1-form symmetry $(\mathbb{Z}_N^{(1)})_{3d}$. Using the explicit form of the effective twisted superpotential provided above, we can explicitly compute the eigenvalues of the $\Pi^\gamma$ operator (48) (which is diagonalised by the Bethe vacua):

$$\Pi^{\gamma_0}(\hat{u}) = \hat{x}_1^k \prod_{i=1}^{n_{\mathrm{adj}}} \prod_{b=2}^{N} \frac{\hat{x}_1 - y_i \hat{x}_b}{\hat{x}_b - y_i \hat{x}_1} = \hat{q}^{-1}, \tag{286}$$

where $\gamma_0$ is the generator of $\mathbb{Z}_N^{(1)}$ defined in (144) and $\hat{q}$ denotes the on-shell value of $q$, as we used the equations (285). Note that, in general, one might want to add an overall phase to (286), as discussed in the previous sections.

**The anomaly factor**   The fact that the adjoint matter is not charged under the $(\mathbb{Z}_N^{(1)})_{3d}$ symmetry directly implies that:

$$\mathcal{W}_{\mathrm{adj}}(u + \gamma_0) = \mathcal{W}_{\mathrm{adj}}(u), \tag{287}$$

as one can readily check. The same argument would apply to any gauge theory with matter preserving some 1-form symmetry. Hence, we see that the 't Hooft anomaly factor (76) only receives contributions only from the bare CS terms. We thus find that the 't Hooft anomaly is:

$$\mathfrak{a} = -K \bmod N = -k \bmod N, \tag{288}$$

where we use the relation (283). Hence the general constraints on orbit structures that follow from the presence of a 't Hooft anomaly are exactly the same as for the pure $\mathcal{N} = 2$ CS theory.

In principle, it is now straightforward to apply the formalism of section 2 to this theory. In practice, explicit computations remain challenging. We hope to come back to this problem in future works.

# 6   Conclusion and outlook

Three-dimensional $\mathcal{N} = 2$ supersymmetric gauge theories with a $U(1)_R$ R-symmetry allow for the computation of many exact observables, but previous methods on half-BPS three-manifolds $\mathcal{M}_3$ with non-trivial first homology were restricted to gauge groups $\tilde{G}$ whose fundamental group $\pi_1(G)$ is a free abelian group. In this work, we lifted this restriction in the case of $\mathcal{M}_3 = \Sigma_g \times S^1$, wherein the supersymmetric partition computes the topologically twisted index [10–13].

We developed a systematic formalism to compute the twisted index $Z_{\Sigma_g \times S^1}$ by gauging a one-form symmetry $(\Gamma^{(1)})_{3d} \cong \Gamma$ of the 3d gauge theory. We performed this gauging in the 3d *A*-model formalism on $\Sigma_g$, hereby gauging the two distinct discrete higher-form symmetries $\Gamma^{(1)}$ and $\Gamma^{(0)}$ in the 2d TQFT description:

$$Z_{\Sigma_g \times S^1}\left[\mathcal{T}/(\Gamma^{(1)})_{3d}\right] = \frac{1}{|\Gamma|^{2g}} \sum_{B \in H^2(\Sigma_g, \Gamma)} \sum_{C \in H^1(\Sigma_g, \Gamma)} Z_{\Sigma_g \times S^1}[\mathcal{T}](B, C). \tag{289}$$

This gauging was be done very explicitly, building directly on earlier insights [17–19]. In particular, we refined the results of [19] and connected them to the 2d Hilbert-space approach in the *A*-model. As an intermediate step, we carefully studied the correlation functions of topological point and line operators that implement the symmetries in the *A*-model:

$$Z_{\Sigma_g \times S^1}[\mathcal{T}](B, C) \cong \langle \Pi^\gamma(\mathrm{pt}) \mathcal{U}^\delta(\mathcal{C}) \rangle_{\Sigma_g}, \qquad \gamma \in \Gamma^{(1)}, \quad \delta \in \Gamma^{(1)}. \tag{290}$$

We also studied the 't Hooft anomalies that can affect these symmetries, and showed how they usefully constrain the structure of the 2d ground states (also known as Bethe vacua).

The core of this paper was the study of the $SU(N)_K$ $\mathcal{N} = 2$ Chern–Simons theory with a $\mathbb{Z}_N$ symmetry, for any $N$, and of all the $(SU(N)/\mathbb{Z}_r)_K$ $\mathcal{N} = 2$ CS theories that one can obtain by discrete gauging. Our analysis leads to very explicit formulas for the twisted indices, and especially for the Witten index (at $g = 1$), which we can write down in terms of simple number-theoretic functions for any $N$, $r$ and $K$. While our results match with many previous results in special cases (see *e.g.* [36, 44, 50, 54] in the mathematical literature), our most general formulas appear to be new results. In particular, we noticed and exploited a subtle gravitational-$(\Gamma^{(1)})_{3d}$ mixed anomaly of the $SU(N)_K$ theory for $N$ even, whose treatment is crucial in order to obtain the correct results for any $r|N$. As already emphasised in the introduction, this mixed anomaly is already present for $SU(2)_K$ with $K/2$ even, and it is always related to the corresponding $(SU(N)/\mathbb{Z}_r)_{k=K-N}$ $\mathcal{N} = 0$ CS theory being a spin-TQFT instead of a bosonic 3d TQFT. Unfortunately, the presence of the gravitational mixed anomaly actually means that our naive index computation in this case does not capture the full story. This will be explained and expanded on in future work [20, 21].

**Outlook**   Many different research directions could be followed to extend the results of this paper. Let us briefly enumerate the most salient ones:

- Using the formalism of this paper, one can compute twisted indices for any $\mathcal{N} = 2$ gauge theory with a real compact gauge group $G$, modulo the caveats already mentioned.[38] To do this as explicitly as possible, one should further develop powerful computational algebraic methods, as was recently done for unitary gauge groups [108] – in that language, $\Gamma^{(0)}$ and $\Gamma^{(1)}$ correspond to a non-trivial action and to a grading, respectively, on the Bethe ideal defined by the Bethe equations. It is then conceivable that a more algebraic approach to the computation of the correlators (290) is possible.

- One low-hanging fruit might be the explicit computation of Witten indices for general gauge groups with general $(\Gamma^{(1)})_{3d}$ a finitely-generated abelian group, possibly with matter, given the result for $G = \widetilde{G}$. For non-cyclic groups $\Gamma^{(1)}_{3d}$, we expect an analogous result as for the $\widetilde{G} = SU(N)$ case of this paper, with the enumerating functions being appropriate totient functions for finite abelian groups (see *e.g.* [110]).

- While we focussed on one-form symmetries of 3d $\mathcal{N} = 2$ theories, such field theories can admit even more interesting generalised symmetries, including non-invertible symmetries. This will be particularly important to explore when studying $SO(N)$ gauge theories, where we would need to study the gauging of both 1-form and 0-form symmetries in 3d, corresponding to rich combinations of 1-, 0- and $(-1)$-form symmetries in the $A$-model. See *e.g.* [34, 48, 80, 105, 111–116] for relevant studies.

- The approach of this paper can be extended to compute more general partition functions for $G = \widetilde{G}/\Gamma$, by inserting Seifert-fibering operators on $\Sigma_g$ to obtain $Z_{\mathcal{M}_3}$ for any Seifert 3-manifold $\mathcal{M}_3$ [7, 9, 117] – this will be discussed in [20].

- The $A$-model approach is also applicable to 4d $\mathcal{N} = 1$ supersymmetric field theories compactified on $T^2$ [10, 107, 118, 119], and it will be very interesting to study higher-form symmetries in that context. We expect 't Hooft anomalies and their consequences to be particularly intricate in this case.

---

[38]That is, with the usual assumptions for the 3d $A$-model to exist, including the existence of an $R$-symmetry, and also assuming the Bethe states of the $G$ theory are all bosonic.

Other interesting questions include the study of higher-form symmetries for the $T[M_3]$ theory of the 3d/3d correspondence [18, 71–73], and the application of the 2d perspective to other 3d TQFTs (whether or not the UV completion is supersymmetric), for instance the theories of [28, 120, 121] and of [122]. Finally, our explicit expressions for the $(SU(N)/\mathbb{Z}_r)_K$ indices seem amenable to large-$N$ studies, which could be very interesting to explore, for instance, in connection to supersymmetric black holes in holography [123].

# Acknowledgments

We are grateful to Riccardo Argurio, Lakshya Bhardwaj, Lea Bottini, Andrea Ferrari, Sergei Gukov, Dragos Oprea, Du Pei, Sakura Schafer-Nameki, and Shu-Heng Shao for correspondence and discussions, and we are particularly grateful to Brian Willett for allowing us to use his unpublished results in this work. CC also acknowledges many influential discussions with Heeyeon Kim and with Brian Willett on the subject of the 3d $A$-model over many years.

**Funding information** CC is a Royal Society University Research Fellow. EF is supported by the EPSRC grant "Local Mirror Symmetry and Five-dimensional Field Theory". The work of OK is supported by the School of Mathematics at the University of Birmingham. We thank the Galileo Galilei Institute for Theoretical Physics for the hospitality and the INFN for partial support during the completion of this work.

# A  3d $A$-model for $\widetilde{G}$: A lightning review

In this appendix, we review some aspects of the 3d $A$-model for 3d $\mathcal{N} = 2$ Chern–Simons-matter gauge theory $\mathcal{T}_{\widetilde{G}}$ with a gauge group $\widetilde{G}$, a product of simply connected compact Lie groups and of unitary gauge groups. We refer to [7, 8] for further background and explanations, as well as to [108] for a recent review in the case of unitary gauge groups.

## A.1  The Coulomb branch parameters

The main player in this discussion is the 3d classical Coulomb branch parameter, which we denote by $u$. To define this variable, we put our 3d theory on $\mathbb{R}^2 \times S^1_\beta$. Effectively, this gives us a 2d $\mathcal{N} = (2, 2)$ theory with an infinite number of massive Kaluza-Klein (KK) modes that carry momentum along the compactified dimension $S^1_\beta$ with radius $\beta$. In the 2d $\mathcal{N} = (2, 2)$ language, the vector multiplet is repackaged into a twisted chiral multiplet whose lowest component is a complex scalar $u$. This dimensionless scalar is defined by combining the real scalar $\sigma$ with the 3d gauge field along the $S^1$-direction, $A_3$:

$$u = \beta(i\sigma + A_3) \in \mathfrak{t}/\Lambda^{\widetilde{G}}_{\mathrm{mw}}. \tag{A.1}$$

Here, $\mathfrak{t}$ is the Cartan subalgebra of $\tilde{G}$, and $\Lambda^{\widetilde{G}}_{\mathrm{mw}}$ is the magnetic weight lattice. We refer to appendix B for a brief review of the relevant electric and magnetic weight lattices for general gauge groups. Choosing $\{e^a\}^{\mathrm{rk}\,\widetilde{G}}_{a=1}$ to be an integral basis for the magnetic weight lattice $\Lambda^{\widetilde{G}}_{\mathrm{mw}}$, we can expand $u$ as follows:

$$u = u_a e^a, \tag{A.2}$$

where the sum over the index $a = 1, \cdots, \mathrm{rank}(\widetilde{G})$ is assumed. This basis is integral in the sense that we have:

$$\rho(u) = \rho^a u_a, \qquad \rho^a \equiv \rho(e^a) \in \mathbb{Z}, \tag{A.3}$$

for any electric weight $\rho \in \Lambda_{\mathrm{w}}^{\widetilde{G}}$. Under large gauge transformations, the Coulomb branch parameters $u_a$ transform as:

$$u_a \sim u_a + n_a, \qquad n_a \in \mathbb{Z}. \tag{A.4}$$

Therefore, it is sometimes useful to introduce the single-valued parameters:

$$x_a \equiv e^{2\pi i u_a}, \qquad a = 1, \ldots, \mathrm{rk}\, \widetilde{G}. \tag{A.5}$$

One can play a similar game for any flavour symmetry group $G_{\mathrm{F}}$ that might be present in the theory. In this case, we consider a 3d $\mathcal{N} = 2$ vector multiplet with real scalar $m^{\mathrm{F}}$ and gauge field $A^{\mathrm{F}}$, and we define the 2d twisted masses:

$$\nu \equiv \beta(i m_{\mathrm{F}} + A_3^{\mathrm{F}}) \in \mathfrak{t}^{\mathrm{F}}/\Lambda_{\mathrm{mw}}^{G_{\mathrm{F}}}. \tag{A.6}$$

As in the case of the gauge group $\widetilde{G}$, we can pick an integral basis $\{e_{\mathrm{F}}^\alpha\}_{\alpha=1}^{\mathrm{rk}\, G_{\mathrm{F}}}$ for the flavour magnetic weight lattice $\Lambda_{\mathrm{mw}}^{G_{\mathrm{F}}}$, so that:

$$\nu = \nu_\alpha e_{\mathrm{F}}^\alpha, \tag{A.7}$$

where $\alpha = 1, \ldots, \mathrm{rank}(G_{\mathrm{F}})$ runs over a maximal torus of the flavour group.

## A.2 The effective twisted superpotential and the effective dilaton

The 2d $\mathcal{N} = (2,2)$ low energy effective description on $\Sigma$ is controlled by the so-called effective twisted superpotential $\mathcal{W}(u, \nu)$ and by the effective dilaton $\Omega(u, \nu)$, which one obtains upon integrating out the massive charged chiral multiplets on $\Sigma \times S^1$ [10].

**Effective twisted superpotential**   The twisted superpotential receives contributions from the CS action and from the 3d $\mathcal{N} = 2$ chiral multiplets. It has the following general form:

$$\mathcal{W}(u, \nu) = \mathcal{W}_{\mathrm{matter}}(u, \nu) + \mathcal{W}_{\mathrm{CS}}(u, \nu). \tag{A.8}$$

The matter contribution reads:

$$\mathcal{W}_{\mathrm{matter}}(u, \nu) \equiv \frac{1}{(2\pi i)^2} \sum_{(\rho, \rho^{\mathrm{F}}) \in \mathfrak{R} \times \mathfrak{R}^{\mathrm{F}}} \mathrm{Li}_2\left(e^{2\pi i(\rho(u) + \rho_{\mathrm{F}}(\nu))}\right), \tag{A.9}$$

where $\mathfrak{R} \times \mathfrak{R}^{\mathrm{F}}$ is the gauge and flavour representation of the 3d $\mathcal{N} = 2$ chiral multiplets $\Phi$ under $\widetilde{G} \times G_{\mathrm{F}}$, and $(\rho, \rho^{\mathrm{F}}) \in \Lambda_{\mathrm{mw}}^{\widetilde{G}} \times \Lambda_{\mathrm{mw}}^{G_{\mathrm{F}}}$ are the corresponding weights. The CS contributions are schematically given by:

$$\mathcal{W}_{\mathrm{CS}}(u) = \frac{1}{2} \sum_{a,b} K_{ab}(u_a u_b + \delta_{ab} u_a), \tag{A.10}$$

where $K_{ab}$ denote the effective UV CS levels associated with the gauge group $\widetilde{G}$ in the so-called $U(1)_{-\frac{1}{2}}$ quantization [7].[39] The expression (A.10) is the contribution from the gauge CS terms, but the flavour CS levels [124] contribute similarly.

---

[39]For a recent review, see also section 2.2 of [108].

**Effective dilaton** This effective dilaton is a holomorphic function that couples $u$ to the curvature of $\Sigma$ [10, 70]. For our gauge theory, it reads:

$$\Omega(u, v) = \Omega_{\mathrm{CS}}(u, v) + \Omega_{\mathrm{matter}}(u, v) + \Omega_{\mathrm{W\text{-}boson}}(u). \tag{A.11}$$

The CS contribution involves (mixed) CS levels for the $U(1)_R$ symmetry [8]:

$$\Omega_{\mathrm{CS}}(u, v) = K_{RG} \sum_{a=1}^{\mathrm{rk}(\widetilde{G})} u_a + \sum_{\alpha=1}^{\mathrm{rk}(G_F)} K_{R\alpha} v_\alpha + \frac{1}{2} K_{RR}. \tag{A.12}$$

The matter contribution reads:

$$\Omega_{\mathrm{matter}}(u, v) = -\frac{1}{2\pi i} \sum_{(\rho, \rho^F) \in \mathfrak{R} \times \mathfrak{R}^F} (r_{\rho^F} - 1) \log\left(1 - e^{2\pi i(\rho(u) + \rho^F(v))}\right), \tag{A.13}$$

where $r_{\rho^F}$ denote the R-charges of the chiral multiplets. Finally, the W-bosons contribute to the effective dilaton potential as:

$$\Omega_{\mathrm{W\text{-}bosons}}(u) = -\frac{1}{2\pi i} \sum_{\alpha \in \Delta} \log\left(1 - e^{2\pi i \alpha(u)}\right), \tag{A.14}$$

where the sum is over the roots of the gauge group $\widetilde{G}$.

## A.3 The 3d topologically twisted index

The topologically twisted index for a 3d $\mathcal{N} = 2$ gauge theory with gauge group $\widetilde{G}$ can be computed as a trace over certain operators in the 3d $A$-model [10, 13]. All these operators $\mathcal{O}$ are given 'off-shell' as function $\mathcal{O}(u)$ of the gauge and flavour parameters $u$ and $v$. They diagonalise the Bethe vacua, and their 'on-shell' values at solutions of the Bethe equations are denoted by $\mathcal{O}(\hat{u})$:

$$\mathcal{O}|u\rangle = \mathcal{O}(\hat{u})|u\rangle. \tag{A.15}$$

The Bethe equations themselves are written in terms of the gauge flux operators:

$$\Pi_a(u, v) = \exp\left(2\pi i \frac{\partial \mathcal{W}(u, v)}{\partial u_a}\right), \qquad a = 1, \ldots, \mathrm{rk}\,\widetilde{G}. \tag{A.16}$$

By definition, the gauge flux operators are trivial on-shell, $\Pi_a(\hat{u}) = 1$. The set of Bethe vacua is defined as in (46), namely:

$$\mathcal{S}_{\mathrm{BE}} \equiv \left\{\hat{u} \in \mathfrak{t}/\Lambda_{\mathrm{mw}}^{\widetilde{G}} : \Pi_a(\hat{u}, v) = 1, \forall a \quad \text{and} \quad w \cdot \hat{u} \neq \hat{u}, \forall w \in W_{\widetilde{G}}\right\}/W_{\widetilde{G}}. \tag{A.17}$$

Here we need to exclude putative solutions that have a non-trivial stabiliser for the action of the Weyl group $W_{\widetilde{G}}$, and we then identify all solutions related by the Weyl symmetry. Given a non-trivial flavour symmetry group $G_F$,[40] we similarly define the flavour flux operator:

$$\Pi_{F, \alpha}(u, v) = \exp\left(2\pi i \frac{\partial \mathcal{W}(u, v)}{\partial v_\alpha}\right), \qquad \alpha = 1, \ldots, \mathrm{rk}\,G_F. \tag{A.18}$$

Finally, the most important operator is the handle-gluing operator $\mathcal{H}_{\widetilde{G}}(u, v)$ which is given:

$$\mathcal{H}_{\widetilde{G}}(u, v) = \exp(2\pi i \Omega(u, v)) \det_{1 \leq a, b \leq \mathrm{rk}\,\widetilde{G}}\left(\frac{\partial^2 \mathcal{W}(u, v)}{\partial u_a \partial u_b}\right), \tag{A.19}$$

We simply denote it by $\mathcal{H}$ whenever it is clear that we are talking about the $\widetilde{G}$ theory. The insertion of this operator on $\Sigma$ has the effect of adding a handle, thus increasing the genus of the Riemann surface [10].

---

[40]Or rather a flavour symmetry algebra; we may assume that the fundamental group of $G_F$ is a free abelian group for our purposes here.

**The 3d flavoured twisted index from the *A*-model**   The 3d flavoured twisted index is the Witten index defined as a trace over the Hilbert space of the theory compactified on $\Sigma_g$ with a topological *A*-twist and with fugacities $y_\alpha \equiv e^{2\pi i \nu_\alpha}$ and background magnetic fluxes $\mathfrak{m}_F$ for the flavour symmetry:

$$
Z_{\Sigma_g \times S^1} = \mathrm{Tr}_{\mathcal{H}_{\Sigma_g; \mathfrak{m}_F}} \left( (-1)^F \prod_{\alpha=1}^{\mathrm{rk}\, G_F} y_\alpha^{Q_\alpha^F} \right). \tag{A.20}
$$

It can be computed in the *A*-model formalism as a trace over $\mathcal{H}_{S^1}$, the ground-state Hilbert space spanned by the Bethe vacua:

$$
Z_{\Sigma_g \times S^1} = \mathrm{Tr}_{\mathcal{H}_{S^1}} \left( \mathcal{H}^{g-1} \Pi_F^{\mathfrak{m}_F} \right). \tag{A.21}
$$

For the $\widetilde{G}$ gauge theory, this 2d TQFT formula takes the explicit form [13]:

$$
Z_{\Sigma_g \times S^1}[\widetilde{G}](\nu)_{\mathfrak{m}_F} = \sum_{\hat{u} \in \mathcal{S}_{\mathrm{BE}}} \mathcal{H}_{\widetilde{G}}^{g-1}(\hat{u}, \nu) \Pi(\hat{u}, \nu)^{\mathfrak{m}} \Pi_F(\hat{u}, \nu)^{\mathfrak{m}_F}, \tag{A.22}
$$

where we evaluate (A.18) and (A.19) at the Bethe vacua. Note that we use the shorthand notation $\Pi^{\mathfrak{m}_F} \equiv \prod_\alpha \Pi_\alpha^{\mathfrak{m}_{F,\alpha}}$.

The formula (A.21) holds for any 3d *A*-model, and in particular for any 3d $\mathcal{N}=2$ gauge theory with any choice of (compact, real) gauge group $G$. In the main text, we effectively compute $\mathcal{H}_G$ for $G = \widetilde{G}/\Gamma$ if $\Gamma$ is non-anomalous. We can further develop the 2d TQFT perspective by assigning Hilbert-space operations to basic two-dimensional cobordisms, as discussed in figure 4 in the main text.

# B   Lattices associated with Lie groups

In this appendix, we review some basic facts about Lie groups. In particular, starting from any (simple) Lie group $G$, one can build six generally distinct lattices which are important in the representation theory of $G$. See [14, 125, 126] for a more comprehensive treatment.

First, for any lattice $L \subseteq \mathbb{R}^n$, the dual lattice $L^*$ is defined as:

$$
L^* = \{ f \in (\mathrm{span}(L))^* \,|\, f(x) \in \mathbb{Z}, \, \forall x \in L \}. \tag{B.1}
$$

It holds that $(L^*)^* = L$. Without too much loss of generally, let us assume that the Lie algebra $\mathfrak{g} = \mathrm{Lie}(G)$ of $G$ is simple and compact.[41] Then, the complexification $\mathfrak{g}_\mathbb{C}$ of $\mathfrak{g}$ admits a decomposition:

$$
\mathfrak{g}_\mathbb{C} = \mathfrak{t}_\mathbb{C} \oplus \bigoplus_{\alpha \in \Phi} V_\alpha, \tag{B.2}
$$

where $\mathfrak{t}$ is the *Cartan subalgebra, i.e.* the maximal abelian subalgebra of $\mathfrak{g}$.[42] The set $\Phi \subseteq \mathfrak{t}^*$ are the set of roots, and $V_\alpha \subseteq \mathfrak{g}_\mathbb{C}$ are the root spaces. The Cartan subalgebra $\mathfrak{t}_\mathbb{C} \ni H$ acts on the root spaces $V_\alpha \ni X_\alpha$ as:

$$
[H, X_\alpha] = \alpha(H) X_\alpha. \tag{B.3}
$$

We can always "diagonalise" the $V_\alpha$. For $E_\alpha \in V_\alpha$, there is an $H_\alpha \in \mathfrak{t}$ for each $\alpha \in \Phi$ such that:

$$
[E_\alpha, E_{-\alpha}] = H_\alpha, \qquad [H_\alpha, E_{\pm\alpha}] = \pm 2 E_{\pm\alpha}. \tag{B.4}
$$

---

[41]Once we complexify the Lie algebra, it does not make sense to talk about whether the Lie algebra is compact or not. We can still use the fact that every complex semi-simple Lie algebra always possesses a real form that is compact.

[42]It is customary to be a bit careless about whether by $\mathfrak{g}$ one actually means $\mathfrak{g}_\mathbb{C}$.

The $E_\alpha$ are called *root vectors* and the $H_\alpha$ are the *coroot vectors*. They have a natural pairing: for any $\alpha, \beta \in \Phi$, $\alpha(H_\beta) \in \mathbb{Z}$. The coroots span the Cartan subalgebra $\mathfrak{t}_\mathbb{C}$ over $\mathbb{C}$.

The roots span a lattice called the *root lattice* $\Lambda_r \subseteq \mathfrak{t}^*$ of $\mathfrak{g}$, while the coroots span the coroot lattice $\Lambda_{cr} \subseteq \mathfrak{t}$. The root system typically has a large isometry group, which is called the Weyl group $W$ of $\mathfrak{g}$. The *weight lattice* $\Lambda_w = \Lambda_{cr}^*$ is defined as the dual of the coroot lattice. It contains the root lattice $\Lambda_r$ as a sublattice. The quotient

$$\Lambda_w / \Lambda_r \cong Z(\widetilde{G}), \tag{B.5}$$

is isomorphic (as an abelian group) to the centre of $\widetilde{G}$, which is the unique simply connected compact Lie group with $\mathrm{Lie}(G) = \mathrm{Lie}(\widetilde{G}) = \mathfrak{g}$. Furthermore, the *magnetic weight lattice of* $\mathfrak{g}$, $\Lambda_{mw} = \Lambda_r^*$, is defined as the dual of the root lattice. It contains the coroot lattice as sublattice, and their quotient is again isomorphic to $Z(\widetilde{G})$.

The four lattices $\Lambda_r$, $\Lambda_w$, $\Lambda_{mw}$ and $\Lambda_{cr}$ depend only on the Lie algebra $\mathfrak{g}$ and not on the global form of the Lie group $G$. In order to construct representations for $G$ rather than for $\mathfrak{g}$, one must study the exponential map $\exp : \mathfrak{g} \to G$, $H \mapsto e^{2\pi i H}$. A fundamental theorem states that, if $G$ is connected and compact, the exponential map is surjective, *i.e.* $\exp(\mathfrak{g}) = G$. By restricting the exponential map to the Cartan subalgebra, we can construct two new lattices which depend on the global form of $G$. Firstly, we have the lattice:

$$\Lambda_{mw}^G \equiv \ker \exp|_\mathfrak{t} = \{ H \in \mathfrak{t} \,|\, e^{2\pi i H} = \mathbb{1} \}, \tag{B.6}$$

which we call *the magnetic weight lattice of $G$*. This is the lattice of GNO-quantised magnetic fluxes for $G$ [127]. The dual lattice $\Lambda_w^G \equiv (\Lambda_{mw}^G)^*$ is *the weight lattice of the group $G$*. This is a sublattice of $\Lambda_w$ which always contains $\Lambda_r$. In summary, we have:

$$
\begin{array}{ccccccc}
\mathfrak{t}^* : & \Lambda_r & \overset{Z(G)}{\subseteq} & \Lambda_w^G & \overset{\pi_1(G)}{\subseteq} & \Lambda_w \\[2em]
& \updownarrow{\scriptstyle *} & & \updownarrow{\scriptstyle *} & & \updownarrow{\scriptstyle *} \\[2em]
\mathfrak{t} : & \Lambda_{mw} & \overset{Z(G)}{\supseteq} & \Lambda_{mw}^G & \overset{\pi_1(G)}{\supseteq} & \Lambda_{cr}
\end{array}
\tag{B.7}
$$

The groups $Z(G)$ and $\pi_1(G)$ above the inclusions denote the groups that the respective quotients are isomorphic to, *i.e.* $B \overset{\mathcal{G}}{\supseteq} A$ means that $B/A \cong \mathcal{G}$.

For the purpose of gauging higher-form symmetries, it is useful to further refine these sequences of sublattices. Let $\Gamma \subseteq Z(G)$ be a subgroup of the centre of $G$. Then there exists a magnetic weight lattice $\Lambda_{mw}^{G/\Gamma}$ such that:

$$\Gamma \cong \Lambda_{mw}^{G/\Gamma} / \Lambda_{mw}^G. \tag{B.8}$$

Its dual lattice $\Lambda_w^{G/\Gamma}$ in $\mathfrak{t}^*$ is a sublattice of $\Lambda_w^G$, with the quotient again being isomorphic to $\Gamma$. The other quotient is isomorphic to $Z(G)/\Gamma$, which is well-defined since $Z(G)$ is abelian and thus $\Gamma \triangleleft Z(G)$ is normal in $Z(G)$. We illustrate this in the following diagram:

$$
\begin{array}{ccccccc}
\mathfrak{t}^* : & \Lambda_r & \overset{Z(G)/\Gamma}{\subseteq} & \Lambda_w^{G/\Gamma} & \overset{\Gamma}{\subseteq} & \Lambda_w^G \\[2em]
& \updownarrow{\scriptstyle *} & & \updownarrow{\scriptstyle *} & & \updownarrow{\scriptstyle *} \\[2em]
\mathfrak{t} : & \Lambda_{mw} & \overset{Z(G)/\Gamma}{\supseteq} & \Lambda_{mw}^{G/\Gamma} & \overset{\Gamma}{\supseteq} & \Lambda_{mw}^G
\end{array}
\tag{B.9}
$$

Obviously, as suggested by the notation, $\Lambda_w^{G/\Gamma}$ is the weight lattice for the group $G/\Gamma$.

# C  Mixed 't Hooft anomaly from dimensional reduction

In this appendix, we further discuss the 4d anomaly theory (67) associated with the 3d 1-form symmetry $\Gamma^{(1)}_{3d} \cong \Gamma$. We reduce the theory along $S^1$ and show how the 't Hooft anomaly reduces to a mixed 't Hooft anomaly between the 2d 0-form and 1-form symmetries $\Gamma^{(0)}$ and $\Gamma^{(1)}$. We refer to [114, 128] for recent physics discussions and to [129, 130] for the original mathematical background.

Associated with $\Gamma$, there exists a unique abelian group $\hat{\mathcal{A}}(\Gamma)$, such that for any abelian group $\Gamma'$ there exists a quadratic function $\gamma : \Gamma \to \hat{\mathcal{A}}(\Gamma)$ that satisfies $q = \tilde{q} \circ \gamma$ for any quadratic function $q : \Gamma \to \Gamma'$ and some $\tilde{q} \in \text{Hom}(\hat{\mathcal{A}}(\Gamma), \Gamma')$ uniquely determined by $q$.[43] Let us choose an explicit set of generators of $\Gamma$ such that $\Gamma \cong \bigoplus_i \mathbb{Z}_{N_i}$. Then, the associated abelian group is given by:

$$\hat{\mathcal{A}}(\Gamma) = \bigoplus_i \hat{\mathcal{A}}(\mathbb{Z}_{N_i}) \oplus \bigoplus_{i<j} \mathbb{Z}_{N_i} \oplus \mathbb{Z}_{N_j}, \tag{C.1}$$

with:

$$\hat{\mathcal{A}}(\mathbb{Z}_{N_i}) = \begin{cases} \mathbb{Z}_{N_i}, & \text{if } N_i \in 2\mathbb{Z}+1, \\ \mathbb{Z}_{2N_i}, & \text{if } N_i \in 2\mathbb{Z}. \end{cases} \tag{C.2}$$

The map $\Gamma \to \hat{\mathcal{A}}(\Gamma)$ is also known as the universal quadratic functor.

**Pontryagin square and anomaly theory**  The 4d anomaly theory (67) is defined in terms of the Pontryagin square:

$$\mathcal{P} : H^2(\mathfrak{M}_4, \Gamma) \to H^4(\mathfrak{M}_4, \hat{\mathcal{A}}(\Gamma)). \tag{C.3}$$

The 4d topological action is essentially a multiple of $\mathcal{P}(B_{4d})$, with $B_{4d} \in H^2(\mathfrak{M}_4, \Gamma)$ being a background gauge field for the 3d 1-form symmetry $\Gamma$ extended into the 4d bulk. The explicit form of the Pontryagin square depends on $\Gamma$. For example, for $\Gamma = \mathbb{Z}_N$ and assuming that $H_1(\mathfrak{M}_4, \mathbb{Z})$ is torsion-free, as will be the case for us:

$$\mathcal{P}(B_{4d}) = \begin{cases} B_{4d} \cup B_{4d}, & N \in 2\mathbb{Z}+1, \\ \widetilde{B}_{4d} \cup \widetilde{B}_{4d} \mod 2N, & N \in 2\mathbb{Z}, \end{cases} \tag{C.4}$$

where $\widetilde{B}_{4d} \in H^2(\mathfrak{M}_4, \mathbb{Z})$ is an integral uplift of $B_{4d}$. In the more general case $\Gamma = \bigoplus_i \mathbb{Z}_{N_i}$, we decompose the gauge field as $B_{4d} = \sum_i B_i$ with $B_i \in H^2(\mathfrak{M}_4, \mathbb{Z}_{N_i})$. Then, the Pontryagin square can be expanded accordingly:

$$\mathcal{P}(B_{4d}) = \sum_i \mathcal{P}(B_i) + 2 \sum_{i<j} B_i \cup B_j. \tag{C.5}$$

The four-dimensional anomaly action for $\Gamma^{(1)}_{3d}$ is given by the natural generalisation of $\mathcal{P}(B_{4d})$ given in (68).

**Continuum formulation of the 4d anomaly theory and circle reduction**  Let us consider the continuum form of the anomaly theory (68). It reads [42, 68]:

$$S^{\text{anom}}_{4d} = \int_{\mathfrak{M}_4} \sum_{i,j} \frac{\mathfrak{a}_{ij} N_i N_j}{4\pi \gcd(N_i, N_j)} B_i \wedge B_j + \sum_i \frac{N_i}{2\pi} B_i \wedge dA_i, \tag{C.6}$$

---

[43] A map $q : \Gamma \to \Gamma'$ is said to be a quadratic function iff $q(\gamma) = q(-\gamma)$ and $\langle \gamma, \widetilde{\gamma} \rangle_q \equiv q(\gamma + \widetilde{\gamma}) - q(\gamma) - q(\widetilde{\gamma})$ is bilinear.

where $B_i \in H^2(\mathfrak{M}_4, U(1)^{(1)})$ and $A_i \in H^1(\mathfrak{M}_4, U(1)^{(0)})$. The integers $\mathfrak{a}_{ij}$ satisfy the periodicity conditions $\mathfrak{a}_{ij} \sim \mathfrak{a}_{ij} + \gcd(N_i, N_j)$ on spin four-manifolds.[44] The topological action (C.6) enjoys a gauge symmetry under which the gauge fields $B_i$ and $A_i$ transform as

$$B_i \mapsto B_i - d\lambda_i, \qquad A_i \mapsto A_i + \sum_j \frac{\mathfrak{a}_{ij} N_j}{\gcd(N_i, N_j)} \lambda_j. \qquad (C.7)$$

Given the continuum formulation, it is easy to consider the dimensional reduction of the anomaly theory on a circle. That is, we consider $\mathfrak{M}_4 = \mathfrak{M}_3 \times S^1$ with $\partial \mathfrak{M}_3 = \Sigma_g$. The gauge fields $B_i$ and $A_i$ can be decomposed as:

$$B_i \to B_i + \eta_i \wedge C_i, \qquad A_i \to A_i + \eta_i \phi_i, \qquad (C.8)$$

where $\eta_i \in H^1(S^1, \mathbb{Z})$, $B_i \in H^2(\mathfrak{M}_3, U(1))$, $C_i \in H^1(\mathfrak{M}_3, U(1))$, and $\phi_i \in H^0(\mathfrak{M}_3, U(1))$. Here $B_i$ and $C_i$ are the continuum versions of the 3d fields $B$ and $C$ for the anomaly theory of the two-dimensional theory on $\Sigma$. Plugging this back into (C.6) and dimensionally reducing, we find that the 3d anomaly theory is given by:

$$S_{3d}^{\text{anom}} = \int_{\mathfrak{M}_3} \sum_{i,j} \frac{\mathfrak{a}_{ij} N_i N_j}{4\pi \gcd(N_i, N_j)} (C_i \wedge B_j + B_i \wedge C_j) + \sum_i \frac{N_i}{2\pi} (C_i \wedge dA_i + B_i \wedge d\phi_i). \qquad (C.9)$$

This is the continuum version of the mixed 't Hooft anomaly (70) between $\Gamma^{(1)}$ and $\Gamma^{(0)}$ discussed in the main text.

# D  Proofs for $SU(N)_K$ CS theory

In this appendix, we discuss proofs and extended derivations of various results we use in section 3 for calculating expectation values of topological operators in the $SU(N)_K$ theory, for the discrete gauging procedure. In section D.1, we find the fixed points under the action of the full $\mathbb{Z}_N$ 0-form symmetry, in the absence of an anomaly. In section D.2, we extend this analysis to the subgroups of $\mathbb{Z}_N$, which covers all possible anomalies for $SU(N)_K$. Section D.3 discusses the sum over the mixed correlators by carefully studying the mixed gravitational anomaly. In section D.4, we extend Jordan's totient function to accommodate a $\theta$-angle for the gauged 1-form symmetry.

## D.1  Fixed points under the full group

In this section, we complete the proof that there is a unique fixed point under the zero-form symmetry $\mathbb{Z}_N^{(0)}$ in the $SU(N)_K$ theory, for any $N$, whenever $K$ a multiple of $N$. In section 3, we have shown this for $N$ odd. Let us now consider the case $N$ even. In (170), we found the general expression for the fixed point

$$(l_a; \ell) = \left( s, s + \kappa, s + 2\kappa, \dots, s + (N-1)\kappa; Ns + \kappa \frac{N(N-1)}{2} \mod \kappa N \right), \qquad (D.1)$$

where $\kappa = \frac{K}{N} \in \mathbb{Z}$. As discussed in section 3, for $N$ odd the fixed point has $\ell = 0$.

---

[44]It is worth noting that, on general four-manifolds, the 4d TQFT (C.6) would have the periodicities $\mathfrak{a}_{ii} \sim \mathfrak{a}_{ii} + 2N_i$ (and $\mathfrak{a}_{ij} \sim \mathfrak{a}_{ij} + \gcd(N_i, N_j)$ for $i \neq j$), and would only be well-defined if $\mathfrak{a}_{ii} N_i$ were even [42]. We consider spin manifolds only because we are studying supersymmetric field theories, in which case the anomalies coefficients $\mathfrak{a}_{ij}$ are indeed valued in $\mathbb{Z}_{\gcd(N_i, N_j)}$ for every $i, j$.

**Existence and uniqueness**   Let us now prove that the fixed point (D.1) is unique also if $N$ is even. First, we prove that $s = \lceil \frac{\kappa}{2} \rceil$ gives a fixed point, and then we show that it is unique. Consider $\ell = Ns + \kappa \frac{N(N-1)}{2} \mod \kappa N$ for $\kappa$ even. With $s = \frac{\kappa}{2}$ we have $\ell = \frac{\kappa}{2}N^2$, which is $\equiv 0 \mod \kappa N$, since $N$ is even and thus $\ell = \frac{N}{2}\kappa N \equiv 0 \mod \kappa N$. For $\kappa$ odd, on the other hand, with $s = \frac{\kappa+1}{2}$ we have $\ell = \frac{N}{2} + \frac{\kappa}{2}N^2$, where the second term is equivalent to 0 modulo $\kappa N$, for the same reason as above. Since $\frac{N}{2} \in \mathbb{N}$ and is $< N$, we thus have $\ell = \frac{N}{2}$ in this case. We can summarise all cases as follows:

$$e^{-2\pi i \frac{\ell}{N}} = (-1)^{\kappa(N-1)}. \tag{D.2}$$

We have shown that for both $\kappa$ even and odd, $s = \lceil \frac{\kappa}{2} \rceil$ in (170) gives a fixed point. In order to show uniqueness, we add to $s$ an integer $b$ in a valid range and show that it vanishes. For $\kappa$ even, consider therefore $s = \frac{\kappa}{2} + b$ with $b \in \mathbb{Z}$, which since $0 \le s < \kappa$ is between $-\frac{\kappa}{2} \le b < \frac{\kappa}{2}$. For this value of $s$, we find $\ell = Nb$. Since $|b| \le \frac{\kappa}{2} < \kappa$, the remainder of $\ell$ is $Nb \mod N\kappa = Nb$. We now impose $\ell < N$, from which it is clear that $b = 0$. Similarly if $\kappa$ is odd, then for $s = \frac{\kappa+1}{2} + b$ we find $\ell = Nb \mod N\kappa \equiv Nb < N$ and thus $b = 0$.

**Fixed Bethe vacua**   We have proven the existence and uniqueness of the fixed point (170) for all $N$ and all multiples $K$ of $N$. In fact, the fixed Bethe vacuum is independent of the value of $K$ and takes a simple form: if $N$ is odd, then $l_a = (a-1)\kappa$ and $\ell = 0$. Thus if we denote by $\hat{u}_{\text{fixed},a}$ the components of the fixed point solution $\hat{u}_{\text{fixed}}$, we find

$$\hat{u}_{\text{fixed},a} = \frac{1}{N}(a-1), \qquad N \text{ odd}. \tag{D.3}$$

When $N$ is even, we need to distinguish the two cases $\kappa$ even and odd. If $\kappa$ is even, then $l_a = \kappa(a - \frac{1}{2})$, while $\ell = 0$. Therefore, we find $\hat{u}_{\text{fixed},a} = \frac{1}{N}(a - \frac{1}{2})$. If $\kappa$ is odd rather, we have $l_a = \frac{1}{2} + \kappa(a - \frac{1}{2})$, while $\ell = \frac{N}{2}$, such that

$$\hat{u}_{\text{fixed},a} = \frac{1}{N}\left(a - \frac{1}{2}\right), \qquad N \text{ even}, \tag{D.4}$$

which is precisely the same as for $\kappa$ even. Since the index $a$ runs from $1, \ldots, N$, we find that $2N\hat{u}_{\text{fixed}}$ is the list of even numbers $0, \ldots, 2N - 2$ if $N$ is odd, and it is the list of odd numbers $1, \ldots, 2N - 1$ if $N$ is even. Note that for both cases $N$ even and odd, we have that

$$l_a - l_b = (a - b)\kappa. \tag{D.5}$$

This identity is important when evaluating the handle-gluing operator (138) on the fixed point, which we use in (173).

## D.2   Fixed points under subgroups

When $N$ is prime, every nontrivial element of $\mathbb{Z}_N$ is a generator of the full group, and the result in the previous subsection applies to any $\gamma \ne 0$. If $N$ has nontrivial divisors instead, particular elements of $\mathbb{Z}_N$ can generate proper subgroups, as discussed in detail in section 3.1.2.

**Enumeration of fixed points**   Let us now study the fixed points under a general subgroup $\mathbb{Z}_d$ of $\mathbb{Z}_N$, where of course $d$ is a divisor of $N$ and $K$. Above, we proved that under the action of the $\mathbb{Z}_N$ subgroup there is precisely one fixed point. From the shift (166) of a Bethe vacuum under the generator $\gamma_0$, we find that $\ell \to \ell$ and $l_a \to l_a + \frac{K}{N} \mod K$. Applying this map $n$ times, the action under $n\gamma_0$ is therefore $l_a \to l_a + n\frac{K}{N} \mod K$. Since the $l_a \in \{0, \ldots, K\}$, and we are free to reorder the $l_a$'s, this means that two particular values of $l_a$ are related by an action of $n\gamma_0$ if they differ by $\gcd(n, N)\frac{K}{N}$. On the other hand, all fixed points under $n\gamma_0$ are

fixed points under the subgroup $\langle n\gamma_0 \rangle \cong \mathbb{Z}_d$, where $d = N/\gcd(n,N)$. This shows that the fixed points under any $\mathbb{Z}_d$ subgroup have $l_a$ values whose differences are multiples of $\frac{K}{d}$. This generalises the result (170), where $d = N$ for the generator $\langle \gamma_0 \rangle \cong \mathbb{Z}_N$, and we found that the values of $l_a$ differed by $\frac{K}{N}$.

The above analysis proves that we can decompose any fixed point $(l_1, \ldots, l_N; \ell)$ under the action of a $\mathbb{Z}_d$ subgroup into $d$ orbits of tuples $(\tilde{l}_1, \ldots, \tilde{l}_{\frac{N}{d}})$, which is determined from $(l_1, \ldots, l_N; \ell)$ by collapsing the whole list modulo $\frac{K}{d}$ (i.e. $\tilde{l}_a \in \mathbb{Z}_{\frac{K}{d}}$), and adding the orbits[45]

$$(l_1, \ldots, l_N) = \bigcup_{j=0}^{d-1} \left( \tilde{l}_1 + j\frac{K}{d}, \ldots, \tilde{l}_{\frac{N}{d}} + j\frac{K}{d} \right). \tag{D.6}$$

This allows us to count the number of fixed points under a $\mathbb{Z}_d$ subgroup. Since the $d$ full orbits do not have any degrees of freedom, we merely need to enumerate the set of integers $(\tilde{l}_1, \ldots, \tilde{l}_{\frac{N}{d}})$ where each $\tilde{l}_a \in \mathbb{Z}_{\frac{K}{d}}$. This number is simply $\binom{\frac{K}{d}}{\frac{N}{d}}$. Since the 'last' value $\ell$ is not affected by the $\mathbb{Z}_d$ action, we have $N$ initial possibilities for it. As discussed in section 3.1, only every $K$-th candidate tuple satisfies the traclessness condition, which leaves us with

$$\frac{N}{K}\binom{\frac{K}{d}}{\frac{N}{d}} = \binom{\frac{K}{d}-1}{\frac{N}{d}-1}, \tag{D.7}$$

fixed points. This concludes our counting exercise on fixed points, which we use in (175).

## D.3 Sum over mixed correlators

In section 3.4, we consider the gauging of the 0-form symmetry on $T^2$. Due to the particular structure (183) of the fixed points, this amount to summing $\langle \mathcal{U}^{\gcd(m,n)\gamma_0}(\mathcal{C}) \rangle_{T^2}$ over all $m, n \in \mathbb{Z}_N$, and results in a counting exercise of mutual fixed points. In order to study the full 3d one-form gauging, we need to consider mixed correlators such as

$$\left\langle \Pi^{m\gamma_0} \mathcal{U}^{n\gamma_0}(\mathcal{C}) \right\rangle_{T^2} = (-1)^{mn(\frac{K}{N}+1)(N-1)} \left\langle \mathcal{U}^{\gcd(m,n)\gamma_0}(\mathcal{C}) \right\rangle_{T^2}, \tag{D.8}$$

which we studied in section 3.2, or the 'maximal' insertion $\langle \Pi^{m\gamma_0} \mathcal{U}^{n\gamma_0}(\mathcal{C}) \, \mathcal{U}^{n'\gamma_0}(\widetilde{\mathcal{C}}) \rangle_{T^2}$. Due to the modular anomaly on $T^3$, the sign (D.8) makes the sum more elaborate. Before deriving the sum over the maximal insertions required in section 3.4, let us consider the simpler sum over the smaller mixed correlators (D.8) first.

As explained in the main text, when $N$ is odd, the relative sign is equal to one, and the summation proceeds as in (214). Let us thus consider the case $N$ even in the following.

**Case 1: No relative sign**  There are various cases that we can study, in order to find a general formula for the sum over $\mathbb{Z}_N$. Clearly, it depends on the value of $\frac{K}{N}$ for which $n, m$ the sign (D.8) is $+1$ or $-1$. For instance, if $N|K$ we can have the cases where $\frac{K}{N}+1$ is even or odd. If it is even, then there is no relative sign between the two correlators, and we can proceed in summing over them as described above,

$$N|K \text{ and } \frac{K}{N} \text{ odd}: \quad \sum_{m,n=0}^{N-1} \langle \Pi^{m\gamma_0} \mathcal{U}^{n\gamma_0}(\mathcal{C}) \rangle_{T^2} = \sum_{d \, | \, \gcd(N,K)} J_2(d) \binom{\frac{K}{d}-1}{\frac{N}{d}-1}. \tag{D.9}$$

---

[45]We abuse notation by using $\cup$ as the concatenation of tuples.

**Case 2: Alternating sign**   Let us thus consider the next simplest case, where $N|K$ but $\frac{K}{N}+1$ is odd. In that case, the relative sign is merely $(-1)^{mn}$. We can now attempt to enumerate the contributions from all the $m, n$ with these alternating signs. Rather than partitioning $N^2$ into the divisors of $N$, we first partition the $N^2$ tuples $(m, n)$ into two sets where $(-1)^{mn}$ is $+1$ and $-1$. Since $(-1)^{mn} = 1$ if either $m$ or $n$ or both are even, while $(-1)^{mn} = -1$ only if both $m, n$ are odd, this gives a partition of the sort

$$\sum_{m,n=0}^{N-1} (-1)^{mn} = 3\left(\frac{N}{2}\right)^2 - \left(\frac{N}{2}\right)^2 = \frac{N^2}{2}\,. \tag{D.10}$$

Indeed, if $N$ is even, then there are $\frac{N}{2}$ even and $\frac{N}{2}$ odd numbers in the set $\{0, \dots, N-1\}$. This counting exercise allows us to find a general rule for which divisors $d$ of $N$ have contributions from both signs. Collapsing the sum over the $(m, n)$ to a sum over divisors $d$, we identify $d \gcd(m, n, N) = N$. The sign $(-1)^{mn}$ is a minus sign if and only if both $m, n$ are odd. Here it becomes apparent that the condition becomes dependent on the nature of $N$: if $N$ has no odd divisors (*i.e.* it is $2^n$), then this gcd is always equal to 1. In general, with $\gcd(m, n, N)$ for $m, n$ odd we can probe in this way all *odd* divisors of $N$. Thus the sum is alternating precisely if $\frac{N}{d}$ is odd. For any such odd $d|N$ we count now the number of values of $(-1)^{mn} \langle U_{\gcd(m,n)\gamma_0}(B)\rangle_{T^2}$ with plus and minus signs. Using a similar analysis as above (D.10), one can show that for any fixed odd value of $\frac{N}{d}$, there are twice as many $+$ signs as $-$ signs, schematically $n_+ = 2n_-$. Since Jordan's totient $J_2 = n_+ + n_-$ enumerates those regardless of signs, we find that the difference is $n_+ - n_- = \frac{1}{3}J_2$. On the other hand if $\frac{N}{d}$ is even, then $\gcd(m, n, N)$ must be even, in which case $(-1)^{mn} = 1$. We have thus shown

$$N|K \text{ and } \frac{K}{N} \text{ even:} \quad \sum_{m,n=0}^{N-1} \langle \Pi^{m\gamma_0}\mathcal{U}^{n\gamma_0}(\mathcal{C})\rangle_{T^2} = \sum_{d|\gcd(N,K)} \widetilde{\mathscr{J}}_2^N(d)\binom{\frac{K}{d}-1}{\frac{N}{d}-1}, \tag{D.11}$$

where we temporarily introduce the symbol

$$\widetilde{\mathscr{J}}_2^N(d) = \begin{cases} \frac{1}{3}J_2(d), & \frac{N}{d} \text{ odd}, \\ J_2(d), & \frac{N}{d} \text{ even}, \end{cases} \tag{D.12}$$

which we will generalise below.

**Case 3: $N$ even**   The case remains where $N \nmid K$. Of course, for most integers $m$ and $n$, the values $\langle \Pi^{m\gamma_0}\mathcal{U}^{n\gamma_0}(\mathcal{C})\rangle_{T^2}$ vanish due to the mixed anomaly (187). As shown above, in this situation $\frac{K}{N}$ is not necessarily an integer, however for all $m$ and $n$ such that the correlator does not vanish, $\frac{mnK}{N}$ is an integer, and the phase in (D.8) is merely a sign.

Let us study this sign in detail. We focus on the case where the sign is a $-$, that is, $mn(\frac{K}{N}+1)$ is odd. The tricky part is to identify all those $m, n \in \{0, \dots, N-1\}$ for which above combination is odd. This is however not necessary, since as we showed above, it depends only on the fixed value of $\gcd(m, n, N)$ if such correlators are alternating. Since we translate $d = N/\gcd(m, n, N)$ to divisors $d$ of $N$, we can pick the representatives $m = n = \frac{N}{d}$ for any $d|N$. This number can be either odd or even. Indeed, $\gcd(\frac{N}{d}, \frac{N}{d}, N) = \frac{N}{d}$, since of course $\frac{N}{d}$ is a divisor of $N$. For these representatives, we can study the parity of

$$mn\left(\frac{K}{N}+1\right) = \frac{N}{d}\left(\frac{K}{d}+\frac{N}{d}\right). \tag{D.13}$$

If $\frac{N}{d}$ is even, then this number is even and does not give rise to a sign. If $\frac{N}{d}$ is odd, then this number if odd only if $\frac{K}{d}$ is even.

We have shown that the contribution from a given pair $(m, n)$ can be odd only if $\frac{N}{d}$ is odd and at the same time $\frac{K}{d}$ is even. This allows us to generalise (D.12) to arbitrary $K$,

$$\widetilde{\mathscr{I}}_2^{N,K}(d) = \begin{cases} \frac{1}{3}J_2(d), & \frac{N}{d} \text{ odd and } \frac{K}{d} \text{ even}, \\ J_2(d), & \text{otherwise}. \end{cases} \tag{D.14}$$

From this we find

$$N \text{ even:} \quad \sum_{m,n=0}^{N-1} \langle \Pi^{m\gamma_0} \mathcal{U}^{n\gamma_0}(\mathcal{C}) \rangle_{T^2} = \sum_{d \mid \gcd(N,K)} \widetilde{\mathscr{I}}_2^{N,K}(d) \binom{\frac{K}{d}-1}{\frac{N}{d}-1}, \tag{D.15}$$

which holds for $N$ even but arbitary $K$. Let us confirm that it is compatible with the above cases where $N|K$. In either cases $\frac{N}{d}$ even/odd, there exists only a sign issue if $\frac{N}{d}$ is odd. Let now $\frac{K}{N}$ be odd, then $\frac{K}{d}$ is an odd multiple of the odd number $\frac{N}{d}$, which means that $\frac{K}{N}$ is odd. In that case $\widetilde{\mathscr{I}}_2^{N,K}(d) = J_2(d)$, and we derive (D.9). Let now $\frac{K}{N}$ be even, then $\frac{K}{d}$ is an even multiple of the odd number $\frac{N}{d}$ and therefore even. In that case $\widetilde{\mathscr{I}}_2^{N,K}(d) = \widetilde{\mathscr{I}}_2^N(d)$ just returns to the temporary definition (D.12), and indeed we rederive (D.11).

**Arbitrary $N$ and $K$**   Let us now combine all cases, with $N$ and $K$ arbitrary integers. From (D.8) and (214) it is clear that for $N$ odd the correct enumeration is $J_2(d)$ for all divisors $d$ of both $N$ and $K$. This means we can slightly modify $\widetilde{\mathscr{I}}_2^{N,K}(d)$ to include all cases,

$$\mathscr{I}_2^{N,K}(d) = \begin{cases} \frac{1}{3}J_2(d), & N \text{ even, } \frac{N}{d} \text{ odd, } \frac{K}{d} \text{ even}, \\ J_2(d), & \text{otherwise}. \end{cases} \tag{D.16}$$

After all this numerical gymnastics, we arrive at the general expression:

$$N, K \in \mathbb{N}: \quad \sum_{m,n=0}^{N-1} \langle \Pi^{m\gamma_0} \mathcal{U}^{n\gamma_0}(\mathcal{C}) \rangle_{T^2} = \sum_{d \mid \gcd(N,K)} \mathscr{I}_2^{N,K}(d) \binom{\frac{K}{d}-1}{\frac{N}{d}-1}. \tag{D.17}$$

One important check concerns the divisibility of $J_2(d)$ by 3. Since $\mathscr{I}_2^{N,K}(d) = \frac{1}{3}J_2(d)$ only if $N$ is even and $\frac{N}{d}$ odd, this case only occurs if $d$ is even. We can show that $J_2(d)$ for $d$ even is always divisible by 3: For this, let $d = 2m$. If $m$ is odd, then we can show using the definition (215) that $J_2(2m) = 3J_2(m)$. Similarly if $m$ is even, then $J_2(2m) = 4J_2(m)$. Thus if $2m$ contains an odd divisor, we can apply this recursion and find a divisor $3|J_2(2m)$. If $2m$ does not contain any odd divisors, we have $2m = 2^n$, for which $J_2(2^n) = 3 \times 2^{2n-2} \in 3\mathbb{N}$, such that $3|J_2(2m)$ as well. We have thus shown that for any even integer $d$, $J_2(d)$ is divisible by 3, and $\mathscr{I}_2^{N,K} : \mathbb{N} \to \mathbb{N}$ is a well-defined integral map for all integers $N$ and $K$.

**Other mixed correlators**   On $T^3$, there is only one other mixed correlator, which is the maximal insertion $\langle \Pi^{m\gamma_0} \mathcal{U}^{n\gamma_0}(\mathcal{C}) \mathcal{U}^{n'\gamma_0}(\widetilde{\mathcal{C}}) \rangle_{T^2}$. Due to (218), the sum over $m, n, n' \in \mathbb{Z}_N$ only slightly generalises the summation exercise above. Clearly, the phase is a $\pm$ sign at most, and the counting of the number $n_\pm$ of $\pm$ contributions proceeds by relating $3n_+ = 4n_-$ for any divisor $d$ giving rise to minus signs. Then $n_+ + n_- = J_3(d)$, while we are interested in the number $n_+ - n_- = \frac{1}{7}J_3(d)$ after the cancellation of the minus signs. This justifies the definition $\mathscr{I}_3^{N,K}$ in (219) and proves (220) for all values of $N$ and $K$.

### D.4 Theta angle for general $N$

In this section, we elaborate on the calculation of the Witten index of the $PSU(N)_K^{\theta_s}$ theory with angle $\theta_s$, as defined in (226). More generally, for any value of $N$ and $K$ we aim to evaluate

$$\sum_{m,n,n'\in\mathbb{Z}_N} e^{2\pi i \frac{ms}{N}} \langle \Pi^{m\gamma_0}\mathcal{U}^{n\gamma_0}(\mathcal{C})\,\mathcal{U}^{n'\gamma_0}(\widetilde{\mathcal{C}})\rangle_{T^2}\,, \tag{D.18}$$

regardless of the precise form of the mixed anomaly. This sum is rather involved to evaluate explicitly, since the phases for nontrivial theta angle interfere with the enumeration of signs originating from the gravitational anomaly (218). For vanishing theta angle, this resulted in the refinement (219) of Jordan's totient function, as discussed in the previous subsection.

In section 3.4, we argue that the sum (D.18) should be analysed in three steps of increasing difficulty, which is due to the two distinct intricacies: The $\theta$-angle probes the theory in a different fashion depending on whether $N$ is square-free or not [15, 16].[46] The second obstruction is the gravitational anomaly discussed in section 3.2, which we can omit if $K$ rather than $N$ is square-free. Let us thus study these three steps in detail.

**Both $N$ and $K$ square-free**    First, let us assume that both $N$ and $K$ are square-free, that is, no prime factor appears more than once in their prime factor decomposition. This assumption simplifies the discussion for the following reason. Summing over exponentials as in (D.18) gives rise to sums of geometric series of the form

$$\sum_{n=0}^{d-1} e^{2\pi i \frac{ns}{d}} = d\,\delta_{s\,\mathrm{mod}\,d,0}\,, \tag{D.19}$$

where $d$ is any divisor of $N$. Evaluating the triple sum (226) involves products of several such geometric-type series, which are difficult to bring to a simple form. However, we can express

$$\delta_{s\,\mathrm{mod}\,d,0} = \prod_{p|d} \delta_{s\,\mathrm{mod}\,p,0}\,, \tag{D.20}$$

as a product over all prime divisors $p$ of $d$, if and only if every divisor $d$ is square-free – that is, if $N$ is square-free. This is clearly not the case if $N$ is not square-free, for instance $\delta_{s\,\mathrm{mod}\,4,0}$ can not be written as a product over prime divisors of 4.[47] We will return to this point below.

These elaborations lead us to a conjecture on the general form of the Witten index of the $PSU(N)_K$ theory with arbitrary theta angle. For this, we need to introduce an integer-valued 'square-free' refinement of Jordan's totient function,

$$J_k^{\mathrm{sf}}(d,s) \equiv d^k \prod_{p|d}\left(\delta_{s\,\mathrm{mod}\,p,0} - \frac{1}{p^k}\right)\,, \tag{D.21}$$

where the product is again over prime divisors of $d$. Clearly, $J_k^{\mathrm{sf}}(d,0) = J_k(d)$ is the ordinary Jordan totient. Using (D.20), one can show that

$$\sum_{d|N} J_k^{\mathrm{sf}}(d,s) = N^k \delta_{s\,\mathrm{mod}\,N,0}\,, \tag{D.22}$$

---

[46] Recall that the property of a number to be square-free is important in the study of $S$-duality orbits of $\mathcal{N} = 4$ SYM with gauge algebra $\mathfrak{su}(N)$ [15]. If $N$ is square-free, then there is a single orbit relating the $SU(N)/\mathbb{Z}_p$ theories with $p|N$.

[47] If $d$ is not square-free, the lhs of (D.20) would be replaced by $\delta_{m\,\mathrm{mod}\,\mathrm{rad}(d)}$, where the *radical* $\mathrm{rad}(d)$ is the product of the distinct prime numbers dividing $d$.

for all $k \in \mathbb{N}$ and all square-free integers $N$, generalising the relation (216). Then we find for both $N$ and $K$ square-free:

$$\sum_{m,n,n' \in \mathbb{Z}_N} e^{2\pi i \frac{ms}{N}} \langle \Pi^{m\gamma_0} \mathcal{U}^{n\gamma_0}(\mathcal{C}) \mathcal{U}^{n'\gamma_0}(\widetilde{\mathcal{C}}) \rangle_{T^2} = \sum_{d \mid \gcd(N,K)} J_3^{\text{sf}}(d,s) \binom{\frac{K}{d}-1}{\frac{N}{d}-1}. \tag{D.23}$$

**Arbitrary $N$ and square-free $K$**   We can generalise these results to arbitrary $N$ with square-free $K$. As describe above, the result will hold more generally for all values of $N$ and $K$ such that the conditions (201) are false for all divisors $r \mid \gcd(N,K)$. For this, we define for any integer $d$ the *radical* $\text{rad}(d)$ as the product of the distinct prime numbers dividing $d$. Then we refine (D.21) as [50, 52, 53, 131]

$$J_k(d,s) \equiv d^k \delta_{s \bmod \frac{d}{\text{rad}(d)},0} \prod_{p \mid d} \left( \delta_{s \bmod p^{e_d(p)},0} - \frac{1}{p^k} \right), \tag{D.24}$$

where for any prime divisor $p$ of $d$, $e_d(p)$ is the maximal exponent of which $p$ appears in the prime factor decomposition of $d$, that is, we can write $d = \prod_{p \mid d} p^{e_d(p)}$. The number-theoretic interpretation is that $J_k(d,s)$ is the contribution of the partition of $N^k$ into a sum over divisors $d$ of any integer $N$,

$$\sum_{d \mid N} J_k(d,s) = N^k \delta_{s \bmod N,0}, \tag{D.25}$$

generalising (D.22) to arbitrary integers $N$. Then for any value of $N$ and any square-free integer $K$ we find

$$\sum_{m,n,n' \in \mathbb{Z}_N} e^{2\pi i \frac{ms}{N}} \langle \Pi^{m\gamma_0} \mathcal{U}^{n\gamma_0}(\mathcal{C}) \mathcal{U}^{n'\gamma_0}(\widetilde{\mathcal{C}}) \rangle_{T^2} = \sum_{d \mid \gcd(N,K)} J_3(d,s) \binom{\frac{K}{d}-1}{\frac{N}{d}-1}. \tag{D.26}$$

**Arbitrary $N$ and $K$**   Finally, let us comment on the case where both $N$ and $K$ are arbitrary. In these cases, rather than the geometric series (D.19), we get alternating geometric series of the form

$$d \text{ even:} \quad \sum_{n=0}^{d-1} (-1)^n e^{2\pi i \frac{ns}{d}} = d \, \delta_{s-\frac{d}{2} \bmod d,0}. \tag{D.27}$$

Indeed, consider the example of $N = 4$ and $K = 8$, such that $N$ has a divisor $d = 4$ with $N/d$ odd and $K/d = 2$ even. By direct calculation of the sum (D.18), we indeed get a term $\delta_{s-2 \bmod 4,0}$,

$$4^2 \mathbf{I}_{\text{W}}\left[ PSU(4)_8^{\theta_s} \right] = 32 + 40\delta_{s-2 \bmod 4,0} + 24\delta_{s \bmod 2,0} + 8\delta_{s \bmod 4,0}. \tag{D.28}$$

A yet simpler example is $N = 2$ with $K = 4$. Here,

$$4\mathbf{I}_{\text{W}}\left[ SO(3)_4^{\theta_s} \right] = 3 + \delta_{s \bmod 2,0} + 5\delta_{s-1 \bmod 2,0}, \tag{D.29}$$

while $J_3(3,s) = 8\delta_{s \bmod 2,0} - 1$. We leave it for future work to find a suitable modification of (D.24) that works for all integers $N$ and $K$ and generalises (D.26).

# E   Number theory tidbits

In this appendix, we collect some number theoretic identities that are used in the body of the paper, and we review properties of the classical Jordan totient function.

## E.1 Glaisher's theorem

Following section 3.3 and in particular (206), want to prove that there exists an integer $M_{N,K}$, such that

$$\binom{K-1}{N-1} = 1 + M_{N,K} N^2\,, \tag{E.1}$$

assuming that $N \geq 3$ is prime and $N|K$. Let us bring this to a slightly more standard form:

**Proposition 1.** *Let $p \geq 3$ be a prime and $n \in \mathbb{N}$ an integer. Then*

$$\binom{np-1}{p-1} \equiv 1 \mod p^2\,. \tag{E.2}$$

The proof follows directly from the following

**Lemma 1** (Lemma 19 of [132]). *Let $p$ be prime and $k, n \in \mathbb{N}$. Then*

$$\binom{np^l}{kp^l} \equiv \binom{np^{l-1}}{kp^{l-1}} \mod p^{3l-1}\,. \tag{E.3}$$

*Proof.* Let us prove Proposition 1. Set $k = l = 1$ in Lemma 1. Thus $\binom{np}{p} \equiv n \mod p^2$. But $\binom{np}{p} = n\binom{np-1}{p-1}$, and the claim follows. $\qquad\square$

Similar statements are known in the literature (see *e.g.* [133]). Corollaries of Lemma 1 are Wolstenholme's theorem [90] and Babbage's theorem [89], while Proposition 1 is somewhat inconsistently called Glaisher's theorem, being consequences of Glaisher's congruence [91, 134]. See [135] for an excellent review. Let us also note also that excluding further the prime $p = 3$, we can divide by another factor of $p$:

**Proposition 2.** *For any prime $p \geq 5$ and any integer $n$, we have*

$$\binom{np-1}{p-1} \equiv 1 \mod p^3\,. \tag{E.4}$$

## E.2 Totient functions

Totient functions are arithmetic functions which are associated with divisors of a given integer. In this appendix, we list important definitions and properties that are used in the body of the paper, and refer the reader to [136–140] for comprehensive treatments.

**Euler's totient**    The simplest example of a totient function is Euler's totient $\varphi$, which counts the positive integers up to a given integer $n$ that are relatively prime to $n$. In other words, $\varphi(n)$ is the number of integers $k$ in the range $k = 1, \ldots, n$ for which the greatest common divisor $\gcd(n, k)$ is equal to 1. The integers $k$ of this form are sometimes referred to as *totatives* of $n$, giving the function its name. Clearly, $\varphi(1) = 1$ and $\varphi(p) = p - 1$ for prime numbers $p$.

An important property is that every integer $n$ can be partitioned into the Euler totients of its divisors,

$$\sum_{d|n} \varphi(d) = n\,. \tag{E.5}$$

This identity is tightly linked to cyclic groups: For any integer $d$, $\varphi(d)$ is the number of possible generators of the cyclic group $\mathbb{Z}_d$. Indeed, if $\mathbb{Z}_d$ is generated by some element $g$ with $g^d = 1$, then $g^k$ is another generator if and only if $k$ is coprime to $d$. Since every element of $\mathbb{Z}_n$ generates a cyclic subgroup, and all subgroups $\mathbb{Z}_d \subset \mathbb{Z}_n$ are generated by precisely $\varphi(d)$ elements of $\mathbb{Z}_n$, the above formula holds.

The Euler totient can be calculated in several ways. Euler's product formula states that

$$\varphi(n) = n \prod_{p|n} \left(1 - \frac{1}{p}\right), \tag{E.6}$$

where the product is over all prime numbers $p$ dividing $n$.

**Jordan's totient**   A generalisation of Euler's totient function is Jordan's totient function, denoted by $J_k(n)$.[48] For both $k$ and $n$ positive integers, $J_k(n)$ equals the number of $k$-tuples of positive integers that are less than or equal to $n$ and that together with $n$ form a coprime set of $k+1$ integers,

$$J_k(n) = |\{(m_1,\ldots,m_k) \in \mathbb{N} \,|\, 1 \le m_i \le n,\, \gcd(m_1,\ldots,m_k,n) = 1\}|. \tag{E.7}$$

The function $J_k$ has a group-theoretical interpretation as well. Indeed, $J_k(n)$ counts the number of sequences $(g_1,\ldots,g_k)$ of elements in $\mathbb{Z}_n$ such that, if $G_i$ is the subgroup generated by $\{g_1,\ldots,g_i\}$, then we have the subgroup sequence

$$\{0\} \le G_1 \le G_2 \le \ldots \le G_k = \mathbb{Z}_n. \tag{E.8}$$

By an inclusion–exclusion principle it can be shown that Jordan's totient function equals

$$J_k(n) = n^k \prod_{p|n} \left(1 - \frac{1}{p^k}\right), \tag{E.9}$$

where $p$ again ranges through all prime divisors of $n$. Clearly, Jordan's totient function is a generalisation of $\varphi$, since $\varphi = J_1$. An important property is

$$\sum_{d|n} J_k(d) = n^k. \tag{E.10}$$

For any $p$ prime, we have $J_k(p) = p^k - 1$. Clearly, $J_k(1) = 1$. Jordan's totient is a *multiplicative* function, meaning that whenever $m$ and $n$ are coprime (that is, $\gcd(m,n) = 1$), then

$$J_k(m)J_k(n) = J_k(mn). \tag{E.11}$$

Furthermore, we have Gegenbauer's identity

$$J_{k+l}(n) = \sum_{d|n} d^l J_k(d) J_l\left(\tfrac{n}{d}\right), \tag{E.12}$$

for $n, k, l \in \mathbb{N}$ [140]. This gives a relation between $J_3$ and $J_1 = \varphi$,

$$J_3(n) = \sum_{d|n} d\, J_2(d)\, \varphi\left(\tfrac{n}{d}\right). \tag{E.13}$$

Jordan's totient function has various interesting divisibility properties. For instance, it is straightforward to prove that for even integers $2n$, we have

$$2^k - 1 \,\big|\, J_k(2n), \tag{E.14}$$

for all $k, n \in \mathbb{N}$. Finally, we note that there are various other generalisations of Euler's totient $\varphi$. One interesting generalisation of Jordan's totient $J_k$ is the extension to general finite groups [110].

---

[48]Another notation used in the literature is $\varphi_k(n)$.

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
