# Peer review of "One-form symmetries and the 3d $\mathcal{N}=2$ $A$-model: Topologically twisted indices and CS theories"

_SciPost Physics, doi:SciPost Phys. 18, 066 (2025)_

## Round 2 · Referee Report · Anonymous (Referee 1) · 2024-8-23

Report

The manuscript investigates one-form symmetry in 3d N=2 supersymmetric Chern-Simons theory on a circle with topological twist in 2d. The manuscript studies the action of one-form symmetry on the Hilbert space, and computes the 't Hooft anomaly of the one-form symmetry and use it to constrain the vacuum structure.

Before I can recommend the manuscript for publication, here are some minor comments

- In the discussion of the mixed anomaly between gravity and one-form symmetry, this comes from the generator of the one-form symmetry being an emergent fermion. However, such anomaly is only nontrivial in a bosonic theory. Since the theory is supersymmetric, there is also physical fermion, so the theory is fermionic, and the anomaly becomes trivial (e.g. one can tensor the generator with physical fermion to obtain a boson). Equivalently, the second Stiefel-Whitney class is trivial on spin 4-manifolds.

- there is no theta angle in 3d, it is the holonomy of a 0-form symmetry along S1, maybe the author can clarify it?

Recommendation

Ask for minor revision

---

## Round 2 · Referee Report · Anonymous (Referee 2) · 2024-8-26

Strengths

This paper provides an analysis of line operators in three-dimensional gauge theories with one-form symmetries, following up especially C. Closset's extensive work on three-dimensional gauge theories. It is (1) reasonably general and thorough, and certainly (2) this is a timely topic.

Weaknesses

Aside from some minor issues mentioned by the other referee, it looks fine to me, at least on a short reading.

Report

I think the journal's acceptance criteria are met. Modulo minor issues mentioned by the other referee, I recommend it for publication.

Recommendation

Publish (easily meets expectations and criteria for this Journal; among top 50%)

---

## Round 2 · Referee Report · Anonymous (Referee 3) · 2024-8-28

Strengths

1.) The paper discusses 3d QFTs with N=2 SUSY and a U(1)R symmetry. This class of theories can be usefully analyzed by compactifying on a circle (retaining all the KK modes) and studying the resulting 2d (2,2) theory, more precisely its topological A-twist, which provides a natural setting to study many BPS observables of the 3d theory. The main goal of this paper is to generalize previous results (including by Closset and collaborators) on this formalism to 3d gauge theories with more general gauge groups, e.g. simple but non-simply-connected ones, which can be obtained from simply connected ones by gauging a discrete 1-form global symmetry in 3d. The paper is thus situated at the interface between SUSY gauge theories and generalized global symmetries.

2.) The main focus is on pure SUSY Chern-Simons theories, though theories with matter are briefly discussed as well. Much of the paper is dedicated to working out some non-trivial examples in great detail, tracking their symmetries and anomalies to the 2d description, and gauging anomaly-free symmetries to obtain new theories.

3.) A nice application of this formalism is the calculation of observables such as the Witten index (which has fascinating number-theoretic properties) and S3 partition functions.

Weaknesses

As already pointed out by the first referee, the paper highlights a certain mixed anomaly between a 1-form symmetry and spacetime, with 4d anomaly inflow action w2B2, where B2 is the 1-form symmetry background field, and w2 is the 2nd Stiefel-Whittney class of the spacetime four-manifold; but this anomaly is trivial on spin manifolds, which is the setting for generic SUSY theories. This needs to be addressed and clarified by the authors. It is possible that something slightly stronger can be said in pure SUSY Chern-Simons gauge theory (with or without a Yang-Mills term), by noting that the only fermions are the gauginos, with unit R-charge, so that the U(1)R background gauge field is a Spinc connection, in terms of which a version of the anomaly may survive.

Report

Modulo the minor issues mentioned above, which should be revised by the authors, I recommend the paper for publication.

Recommendation

Ask for minor revision

---

## Round 3 · Referee Report · Anonymous (Referee 1) · 2025-1-8

Report
(See e.g. https://arxiv.org/abs/1806.09592 https://arxiv.org/abs/1602.04251
https://arxiv.org/abs/1812.04716)
For example, for this to happen, the local operators need to be bosons, or the fermionic local operators need to carry charge. This can come from the A twist in the 1+1d theory.
Recommendation
Ask for minor revision

Author: Cyril Closset on 2025-01-09 [id 5098]
(in reply to Report 1 on 2025-01-08)Dear referee,
Thank you for the insightful comment on this and on the previous version of the paper.
We do agree that this anomaly is always trivial in a supersymmetric field theory, and our modified discussion in this section states this. See the last paragraph of section 3.2.
The root of the T^3 modular anomaly is the fact that the abelian anyon we want to condense might be fermionic (h=1/2) and in this case the expectation value of the topological line depends on the choice of spin structure. This is something we didn't fully appreciate when writing the paper, but that we are addressing in mode detail in a series of papers that will appear over the coming weeks.
We added comments to that effect in the introduction and throughout the paper. Correspondingly, we removed the statement that we computed the index "for all G" from the title and abstract, as our results in this paper only give a well-defined Witten index when all the Bethe vacua are bosonic. (This is the case for essentially all the literature on the Bethe-vacua approach to 3d partition functions in the last 10 years because people focussed on simply-connected gauge groups. We hope to give a full picture of the completely general case in that work to appear.)
To answer your specific query above: The 3d theory is defined on a (closed) 3-manifold M3, which is always spin. The partition function depends on the choice of spin structure, however, and the anomaly captures this possible dependence. The 4d anomaly functional can be non-trivial when we consider two distinct spin structures on M3, call it M3_\pm, and extend each to an 4-manifold M4_\pm with boundary M3_\pm, each with their distinct spin structure. Gluing the M4_+ to M4_- gives us a closed 4-manifold which is not spin, in general. The anomaly (a sign) arises when we change spin structures on M3 in this way. As we agreed, we can always trivialise the anomaly by considering the new abelian anyon obtained by fusing with a transparent fermionic line, something we mention (citing the literature, especially https://arxiv.org/abs/1812.04716) but will only explore in detail in a work to appear.
Does this clarify this issue, or did you have something else in mind? Thank you very much for your help in improving our manuscript.

---

## Round 3 · Author Response

In the revised version, we have corrected an important point: When the infrared theory after the gauging of the 3d one-form symmetry is a spin TQFT, it becomes necessary to choose a spin structure on the base Riemann surface, and the index will depend on that choice. In such cases, the Bethe vacua of the gauged theory can become fermionic, which affects the supersymmetric index counting. Although the naive index still accurately counts the Bethe vacua or lines, respectively, it does not represent a supersymmetric index in the spin case. We have corrected the relevant statements in this version, and refer the study of fermionic vacua and the spin structure dependence to a future work.

---

## Round 3 · List of Changes

Moreover, we have adjusted the title, abstract and introduction accordingly, and included several important references.

---

## Editorial Decision

published